# The Power of Extrapolation in Federated Learning

**Hanmin Li**
GenAI Center of Excellence
KAUST, Saudi Arabia
hanmin.li@kaust.edu.sa

**Kirill Acharya**[‡]
GenAI Center of Excellence
KAUST, Saudi Arabia
acharya.kk@phystech.edu

**Peter Richtárik**
GenAI Center of Excellence
KAUST, Saudi Arabia
peter.richtarik@kaust.edu.sa

## Abstract

We propose and study several server-extrapolation strategies for enhancing the theoretical and empirical convergence properties of the popular federated learning optimizer `FedProx` [Li et al., 2020]. While it has long been known that some form of extrapolation can help in the practice of FL, only a handful of works provide any theoretical guarantees. The phenomenon seems elusive, and our current theoretical understanding remains severely incomplete. In our work, we focus on smooth convex or strongly convex problems in the interpolation regime. In particular, we propose Extrapolated `FedProx` (`FedExProx`), and study three extrapolation strategies: a constant strategy (depending on various smoothness parameters and the number of participating devices), and two smoothness-adaptive strategies; one based on the notion of gradient diversity (`FedExProx-GraDS`), and the other one based on the stochastic Polyak stepsize (`FedExProx-StoPS`). Our theory is corroborated with carefully constructed numerical experiments.

## 1 Introduction

Federated learning (FL) is a distributed training approach for machine learning models, where multiple clients collaborate under the guidance of a central server to optimize a loss function [Konečný et al., 2016, McMahan et al., 2017]. This method allows clients to contribute to model training while keeping their data private, as it avoids the need for direct data sharing. Often, federated optimization is formulated as the minimization of a finite-sum objective function,

$$\min_{x \in \mathbb{R}^d} \left\{ f(x) := \frac{1}{n} \sum_{i=1}^{n} f_i(x) \right\}, \tag{1}$$

where each $f_i : \mathbb{R}^d \mapsto \mathbb{R}$ is the empirical risk of model $x$ associated with the $i$-th client. The federated averaging method (`FedAvg`) is among the most favored strategies for addressing federated learning problems, as proposed by McMahan et al. [2017], Mangasarian and Solodov [1993]. In `FedAvg`, the server initiates an iteration by selecting a subset of clients for participation in a given round. Each chosen client then proceeds with local training, employing gradient-based techniques like gradient descent (`GD`) or stochastic gradient descent (`SGD`) with random reshuffling, as discussed by Bubeck et al. [2015], Gower et al. [2019], Moulines and Bach [2011], Sadiev et al. [2022].

---

[‡]Kirill was a student at MIPT during his internship at KAUST.

Li et al. [2020] proposed replacing the local training of each client via SGD in FedAvg with the computation of a proximal term, resulting in the FedProx algorithm.

$$x_{k+1} = \frac{1}{n} \sum_{i=1}^{n} \mathrm{prox}_{\gamma f_i}(x_k), \tag{2}$$

where $\gamma > 0$ is the step size, and the proximal operator is defined as

$$\mathrm{prox}_{\gamma f_i}(x) := \arg\min_{z \in \mathbb{R}^d} \left\{ f_i(z) + \frac{1}{2\gamma} \|z - x\|^2 \right\}.$$

Contrary to gradient-based methods like GD and SGD, algorithms based on proximal operation, such as proximal point method (PPM) [Rockafellar, 1976, Parikh et al., 2014] and stochastic proximal point methods (SPPM) [Asi and Duchi, 2019, Bertsekas, 2011, Khaled and Jin, 2022, Patrascu and Necoara, 2018, Richtárik and Takác, 2020] benefit from stability against inaccuracies in learning rate specification [Ryu and Boyd, 2014]. Indeed, for GD and SGD, a step size that is excessively large can result in divergence of the algorithm, whereas a step size that is too small can significantly deteriorate the convergence rate of the algorithm. PPM was formally introduced and popularized by the seminal paper of Rockafellar [1976] to solve the variational inequality problems. In practice, the stochastic variant SPPM is more frequently used.

It is known that the proximal operator applied to a proper, closed and convex function can be viewed as the projection to some level set of the same function depending on the value of $\gamma$. In particular, if we let each $f_i$ be the indicator function of a nonempty closed convex set $\mathcal{X}_i$, then $\mathrm{prox}_{\gamma f_i}(\cdot)$ becomes the projection $\Pi_{\mathcal{X}_i}(\cdot)$ onto the set $\mathcal{X}_i$. In this case, FedProx in (2) becomes the parallel projection method for convex feasibility problem [Censor et al., 2001, 2012, Combettes, 1997a, Necoara et al., 2019], if we additionally assume

$$\mathcal{X} := \bigcap_{i=1}^{n} \mathcal{X}_i \neq \emptyset.$$

A well known fact about the parallel projection method is that its empirical efficiency can often be improved by extrapolation [Combettes, 1997a, Necoara et al., 2019]. This involves moving further along the line that connects the last iterate $x_k$ and the average projection point, resulting in the iteration

$$x_{k+1} = x_k + \alpha_k \left( \frac{1}{n} \sum_{i=1}^{n} \Pi_{\mathcal{X}_i}(x_k) - x_k \right), \tag{3}$$

where $\alpha_k \geq 1$ defines extrapolation level. Despite the various heuristic rules proposed over the years for setting $\alpha_k$ [Bauschke et al., 2006, Censor et al., 2001, Combettes, 1997b], which have demonstrated satisfactory practical performance, it was only recently that the theoretical foundation explaining the success of extrapolation techniques for solving convex feasibility problems was unveiled by Necoara et al. [2019], where the authors considered randomized version of (3) named Randomized Projection Method (RPM). The practical success of extrapolation has spurred numerous extensions of existing algorithms. Notably, Jhunjhunwala et al. [2023] combined FedAvg with extrapolation, resulting in FedExP, leveraging insights from an effective heuristic rule [Combettes, 1997b] for setting $\alpha_k$ as follows:

$$\alpha_k = \frac{\sum_{i=1}^{n} \|x_k - \Pi_{\mathcal{X}_i}(x_k)\|^2}{\left\| \sum_{i=1}^{n} (x_k - \Pi_{\mathcal{X}_i}(x_k)) \right\|^2}. \tag{4}$$

However, the authors did not consider the case of a constant extrapolation parameter, nor did they disclose the relationship between the extrapolation parameter and the stepsize of SGD. The extrapolation parameter can be viewed as a server side stepsize in the context of federated learning, its effectiveness was discussed by Malinovsky et al. [2023].

In the field of fixed point methods, extrapolation is also known as over-relaxation [Rechardson, 1911]. It is a technique used to effectively accelerate the convergence of fixed point methods, including gradient algorithms and proximal splitting algorithms [Iutzeler and Hendrickx, 2019, Condat et al., 2023].

## 1.1 Contributions

Our paper contributes in the following ways; for the notations used please refer to Appendix A.

- Based on the insights gained from the convex feasibility problem, we extend `FedProx` to its extrapolated counterpart `FedExProx` for both convex and strongly[1] convex interpolation problems (See Table 1). By optimally setting the constant extrapolation parameter, we obtain iteration complexity $\mathcal{O}\left(\frac{L_\gamma(1+\gamma L_{\max})}{\epsilon}\right)$[2] in the convex case and $\mathcal{O}\left(\frac{L_\gamma(1+\gamma L_{\max})}{\mu}\log\left(\frac{1}{\epsilon}\right)\right)$ in the strongly convex case, when all the clients participate in the training (full participation). We reveal the dependence of the optimal extrapolation parameter on smoothness, indicating that simply averaging the iterates from local training on the server is suboptimal. Instead, extrapolation should be applied to achieve faster convergence. Specifically, compared to `FedProx` with the same step size $\gamma$, our method is always at least $2 + \frac{1}{\gamma L_{\max}} + \gamma L_{\max}$ times better in terms of iteration complexity, see Remark 5.

- Our method, `FedExProx`, improves upon the worst-case iteration complexity $\mathcal{O}\left(\frac{L_{\max}}{\epsilon}\right)$ of `FedExP` [Jhunjhunwala et al., 2023] to $\mathcal{O}\left(\frac{L_\gamma(1+\gamma L_{\max})}{\epsilon}\right)$ (See Table 2). The improvement could lead to acceleration up to a factor of $n$, see Remark 6. Furthermore, we extend `FedExProx` to client partial participation setting, showing the dependence of optimal extrapolation parameter on $\tau$ which is the number of clients participating in the training and the benefits of a larger $\tau$. In particular, we show that compared to the single client setting, with complexity $\mathcal{O}\left(\frac{L_{\max}}{\epsilon}\right)$, the full participation version enjoys a speed-up up to a factor of $n$, see Remark 7.

- Our theory uncovers the relationship between the extrapolation parameter and the step size in typical gradient-type methods, leveraging the power of the Moreau envelope. We also recover `RPM` of Necoara et al. [2019] as a special case in our analysis (see Remark 12), and show that the heuristic outlined in (4), is in fact a step size based on gradient diversity [Horváth et al., 2022, Yin et al., 2018] for the Moreau envelopes of client functions.

- Building on the insights from Horváth et al. [2022], we propose two adaptive rules for determining the extrapolation parameter: based on gradient diversity (`FedExProx-GraDS`), and the stochastic Polyak step size (`FedExProx-StoPS`) [Horváth et al., 2022, Loizou et al., 2021]. The proposed methods eliminate reliance on the unknown smoothness constant and exhibit "semi-adaptivity", meaning the algorithm converges with any local step size $\gamma$ and by selecting a sufficiently large $\gamma$, we ensure that we lose at most a factor of 2 in iteration complexity.

- We validate our theory with numerical experiments. Numerical evidence suggests that `FedExProx` achieves a $2\times$ or higher speed-up in terms of iteration complexity compared to `FedProx` and improved performance compared to `FedExP`. The framework and the plots are included in the Appendix.

## 1.2 Related work

**Stochastic gradient descent.** `SGD` [Robbins and Monro, 1951, Ghadimi and Lan, 2013, Gower et al., 2019, Gorbunov et al., 2020] stands as a cornerstone algorithm utilized across the fields of machine learning. In its simplest form, the algorithm is written as $x_{k+1} = x_k - \eta \cdot g(x_k)$, where $\eta > 0$ is a scalar step size, $g(x_k)$ represents a stochastic estimator of the true gradient $\nabla f(x_k)$. We recover `GD` when $g(x_k) = \nabla f(x_k)$. The evolution of `SGD` has been marked by significant advancements since its introduction by Robbins and Monro [1951], leading to various adaptations like stochastic batch gradient descent [Nemirovski et al., 2009] and compressed gradient descent [Alistarh et al., 2017, Khirirat et al., 2018]. Gower et al. [2019] presented a framework for analyzing `SGD` with arbitrary sampling strategies in the convex setting based on expected smoothness, which was later extended by Gorbunov et al. [2020] to the case of local `SGD`. While many methods have been crafted to leverage the stochastic nature of $g(x_k)$, substantial research efforts are also dedicated to finding a

---

[1]Strongly convex: $f$ is $\mu$-strongly convex.

[2]As we later see in Theorem 1, here $L_{\max} = \max_{i\in[n]} L_i$, where each $L_i$ is the smoothness of $f_i$, $L_\gamma$ is the smoothness constant of $M^\gamma = \frac{1}{n}\sum_{i=1}^n M_{f_i}^\gamma$.

Table 1: General comparison of `FedExP`, `RPM` and `FedExProx` in terms of conditions and convergence. Each entry indicates whether the method has the corresponding feature (✓) or not (✗). We use the sign "—" where a feature is not applicable to the corresponding method.

| Features | FedExP | RPM[a] | FedExProx |
|---|---|---|---|
| Does not require interpolation regime | ✓ | ✗ | ✗ |
| Does not require convexity[b] | ✓ | ✗ | ✗ |
| Acceleration in the strongly convex setting[c] | ✗ | ✓ | ✓ |
| Does not require smoothness[d] | ✗ | ✓ | ✓ |
| Allows for partial participation of clients[e] | ✗ | ✓ | ✓ |
| Works with constant extrapolation parameter | ✗ | ✓ | ✓ |
| Smoothness and partial participation influence extrapolation | ✗ | ✓ | ✓ |
| Semi-adaptivity[f] | ✗ | — | ✓ |

[a] `RPM` refers to the randomized projection method of Necoara et al. [2019]. Our method includes it as a special case, see Remark 12

[b] Convexity: local objective $f_i$ is convex, which is the indicator function of the convex set $\mathcal{X}_i$ in `RPM`.

[c] The strong convexity pertains to $f$, and for `RPM`, it indicates that the linear regularity condition is satisfied.

[d] Smoothness: $f_i$ is $L_i$-smooth. Our algorithm also applies in the non-smooth case; see Appendix F.2.

[e] Jhunjhunwala et al. [2023] provides no convergence guarantee for client partial participation setting.

[f] The concept of "semi-adaptivity" is explained in Remark 9.

Table 2: Comparison of convergence of `FedExP`, `FedProx`, `FedExProx`, `FedExProx-GraDS` and `FedExProx-StoPS`. The local step size of `FedExP` is set to be the largest possible value $1/6tL$ in the full batch case, where $t$ is the number of local iterations of `GD` performed. We assume the assumptions of Theorem 1 also hold here. The notations are introduced in Theorem 1 and Theorem 2. The convergence for our methods are described for arbitrary $\gamma > 0$. We use $K$ to denote the total number of iterations. For `FedExProx`, optimal constant extrapolation is used. The $\mathcal{O}(\cdot)$ notation is hidden for all complexities in this table.

| | Full Participation | | |
|---|---|---|---|
| **Method** | **General Case** | **Best Case** | **Worst Case** |
| `FedExP` | $6L_{\max}\big/\sum_{k=0}^{K-1}\alpha_{k,P}$ [a] | $6L_{\max}\big/\sum_{k=0}^{K-1}\alpha_{k,P}$ | $6L_{\max}/K$ |
| `FedProx` | $(1+\gamma L_{\max})\big/\gamma\big(2-\gamma L_\gamma\big)K$ [b] | $L_{\max}\big/n\gamma L_\gamma\big(2-\gamma L_\gamma\big)K$ | $L_{\max}\big/\gamma L_\gamma\big(2-\gamma L_\gamma\big)K$ |
| `FedExProx` (New) | $L_\gamma(1+\gamma L_{\max})\big/K$ [c] | $L_{\max}/nK$ | $L_{\max}/K$ |
| `FedExProx-GraDS` (New) | $(1+\gamma L_{\max})\big/\gamma\cdot\sum_{k=0}^{K-1}\alpha_{k,G}$ [d] | $(1+\gamma L_{\max})\big/\gamma\cdot\sum_{k=0}^{K-1}\alpha_{k,G}$ | $(1+\gamma L_{\max})\big/\gamma K$ |
| `FedExProx-StoPS` (New) | $(1+\gamma L_{\max})\big/\gamma\cdot\sum_{k=0}^{K-1}\alpha_{k,S}$ [e] | $(1+\gamma L_{\max})\big/\gamma\cdot\sum_{k=0}^{K-1}\alpha_{k,S}$ | $(1+\gamma L_{\max})\big/\gamma K$ |

[a] The $\alpha_{k,P}$ here is determined according to the theory of Jhunjhunwala et al. [2023].

[b] Notice that we always have $\gamma L_\gamma < 1$, so the complexity of `FedProx` is strictly worse than `FedExProx`.

[c] We have $L_\gamma(1+\gamma L_{\max}) \leq L_{\max}$, see Remark 6.

[d] We leave out a factor of $(1+\gamma L_{\max})/(2+\gamma L_{\max})$ which is a constant between $(\frac{1}{2}, 1)$.

[e] See Remark 11 for a lower bound of $\alpha_{k,S}$, using which we can rewrite the rate as $\frac{L_\gamma(1+\gamma L_{\max})}{K}$.

better stepsize. An illustration of this is the coordinate-wise adaptive step size `Adagrad` [Duchi et al., 2011]. Another approach involves employing matrix step size, as demonstrated by Safaryan et al. [2021], Li et al. [2023, 2024]. Our analysis builds on the theory of `SGD` mainly adapted from Gower et al. [2019] with additional consideration on the upper bound of the step size.

**Stochastic proximal point method.** `PPM` was first introduced by Rockafellar [1976] to address the problems of variational inequalities at its inception. Its transition to stochastic case, motivated by the need to efficiently solve large scale optimization problems, results in `SPPM`. It is often assumed that the proximity operator can be computed efficiently for the algorithm to be practical. Over the years, `SPPM` has been the subject of extensive research, as documented by Bertsekas [2011], Bianchi [2016],

Patrascu and Necoara [2018]. Unlike traditional gradient-based methods, SPPM is more robust to inaccuracies in learning rate specifications, as demonstrated by Ryu and Boyd [2014]. Asi and Duchi [2019] studied APROX, which includes SPPM as the special case using the full proximal model; APROX was later extended into minibatch case by Asi et al. [2020]. However, this extension was based on model averaging rather than iterate averaging. The convergence rate of SPPM has been analyzed in various contexts by Khaled and Jin [2022], Ryu and Boyd [2014], Yuan and Li [2022], revealing that its performance does not surpass that of SGD in non-convex regimes.

**Projection onto convex sets.** The projection method originated from efforts to solve systems of linear equations or linear inequalities [Kaczmarz, 1937, Von Neumann, 1949, Motzkin and Schoenberg, 1954]. Subsequently, it was generalized to address the convex feasibility problem [Combettes, 1997b]. Typically, the method involves projecting onto a set $\mathcal{X}_i$, where $i$ is determined through sampling or other strategies. A particularly relevant method to our paper is the parallel projection method, in which individual projections onto the sets are performed in parallel, and their results are averaged in order to produce the next iterate. It is well-established experimentally that the parallel projection method can be accelerated through extrapolation, with numerous successful heuristics having been proposed to adaptively set the extrapolation parameter [Bauschke et al., 2006, Pierra, 1984]. However, only recently a theory was proposed by Necoara et al. [2019] to explain this phenomenon. Necoara et al. [2019] introduced stochastic reformulations of the convex feasibility problem and revealed how the optimal extrapolation parameter depends on the smoothness of the setting and the size of the minibatch. A better result under a linear regularity condition, which is connected to strong convexity, was also obtained. However, the explanation provided by Necoara et al. [2019] was not satisfactory, as it failed to clarify why adaptive rules based on gradient diversity are effective.

**Moreau envelope.** The concept of the Moreau envelope, also known as Moreau-Yosida regularization, was first introduced by Moreau [1965] as a mathematical tool for handling non-smooth functions. A particularly relevant property of the Moreau envelope is that executing proximal minimization algorithms on the original objective is equivalent to applying gradient methods to its Moreau envelope [Ryu and Boyd, 2014]. Based on this observation, Davis and Drusvyatskiy [2019] conducted an analysis of several methods, including SPPM for weakly convex and Lipschitz functions. The properties of the Moreau envelope and its applications have been thoroughly investigated in many works including Jourani et al. [2014], Planiden and Wang [2016, 2019]. Beyond its role in proximal minimization algorithms, the Moreau envelope has been utilized in the contexts of personalized federated learning [T Dinh et al., 2020] and meta-learning [Mishchenko et al., 2023].

**Adaptive step size.** One of the most crucial hyperparameters in training machine learning models with gradient-based methods is the step size. For GD and SGD, determining the step size often depends on the smoothness parameter, which is typically unknown, posing challenges in practical step size selection. There has been a growing interest in adaptive step sizes, leading to the development of numerous adaptive methods that enable real-time computation of the step size. Examples include Adagrad [Duchi et al., 2011], RMSProp [Hinton et al.], and ADAM [Kingma and Ba, 2015]. Recently, several studies have attempted to extend the Polyak step size beyond deterministic settings, leading to the development of the stochastic Polyak step size [Richtárik and Takác, 2020, Horváth et al., 2022, Loizou et al., 2021, Orvieto et al., 2022]. Gradient diversity, first introduced by Yin et al. [2018], was subsequently analyzed theoretically by Horváth et al. [2022].

## 2 Preliminaries

We now introduce the several definitions and assumptions that are used throughout the paper.

**Definition 1** (Proximity operator)**.** *The proximity operator of an extended-real-valued function* $\phi : \mathbb{R}^d \mapsto \mathbb{R} \cup \{+\infty\}$ *with step size* $\gamma > 0$ *is defined as*

$$\text{prox}_{\gamma\phi}(x) := \arg\min_{z \in \mathbb{R}^d} \left\{ \phi(z) + \frac{1}{2\gamma} \|z - x\|^2 \right\}.$$

It is known that for a proper, closed and convex function $\phi$, the minimizer of $\phi(z) + \frac{1}{2\gamma} \|z - x\|^2$ exists and is unique. Throughout this paper, we assume that the proximal operators are evaluated exactly, with no approximation or inexactness.

**Algorithm 1** Extrapolated SPPM (FedExProx) with partial client participation

1: **Parameters:** extrapolation parameter $\alpha_k > 0$, step size for the proximity operator $\gamma > 0$, starting point $x_0 \in \mathbb{R}^d$, number of clients $n$, total number of iterations $K$, number of clients participate in the training $\tau$, for simplicity, we use $\tau$-nice sampling as an example
2: **for** $k = 0, 1, 2 \ldots K - 1$ **do**
3:     The server samples $S_k \subseteq \{1, 2, \ldots, n\}$ uniformly from all subsets of cardinality $\tau$
4:     The server computes

$$x_{k+1} = x_k + \alpha_k \left( \frac{1}{\tau} \sum_{i \in S_k} \text{prox}_{\gamma f_i}(x_k) - x_k \right). \tag{7}$$

5: **end for**

---

**Definition 2** (Moreau envelope). *The Moreau envelope of an extended-real-valued function $\phi : \mathbb{R}^d \mapsto \mathbb{R} \cup \{+\infty\}$ with step size $\gamma > 0$ is defined as*

$$M_\phi^\gamma(x) := \min_{z \in \mathbb{R}^d} \left\{ \phi(z) + \frac{1}{2\gamma} \|z - x\|^2 \right\}.$$

The following assumptions are used in our analysis. We use the notation $[n]$ for the set $\{1, \ldots, n\}$.

**Assumption 1** (Differentiability). *The function $f_i$ in (1) is differentiable for all $i \in [n]$.*

**Assumption 2** (Interpolation regime). *There exists $x_\star \in \mathbb{R}^d$ such that $\nabla f_i(x_\star) = 0$ for all $i \in [n]$.*

Note that Assumption 2 indicates that each $f_i$ and $f$ are lower bounded. In this paper, we focus on cases where the interpolation regime holds. This assumption often holds in modern deep learning which are overparameterized where the number of parameters greatly exceeds the number of data points, as justified by Arora et al. [2019], Montanari and Zhong [2022]. Our motivation for this assumption partly arises from the convex feasibility problem [Combettes, 1997a, Necoara et al., 2019], wherein the intersection $\mathcal{X}$ is presumed nonempty. This is equivalent to assuming that the interpolation regime holds when $f_i$ is the indicator function of the nonempty closed convex set $\mathcal{X}_i$. Further motivations derived from the proof for this assumption will be discussed later.

**Assumption 3** (Convexity). *The function $f_i : \mathbb{R}^d \mapsto \mathbb{R}$ is convex for all $i \in [n]$. This means that for each $f_i$,*

$$0 \le f_i(x) - f_i(y) - \langle \nabla f_i(y), x - y \rangle, \quad \forall x, y \in \mathbb{R}^d. \tag{5}$$

**Assumption 4** (Smoothness). *Function $f_i : \mathbb{R}^d \mapsto \mathbb{R}$ is $L_i$-smooth, $L_i > 0$ for all $i \in [n]$. This means that for each $f_i$,*

$$f_i(x) - f_i(y) - \langle \nabla f_i(y), x - y \rangle \le \frac{L_i}{2} \|x - y\|^2, \quad \forall x, y \in \mathbb{R}^d. \tag{6}$$

*We will use $L_{\max}$ to denote $\max_{i \in [n]} L_i$.*

It is important to note that the smoothness assumption here is not necessary to obtain a convergence result, see Appendix F.2 for the detail. We introduce this assumption to highlight how the optimal extrapolation parameter depends on smoothness if it is present. The following strong convexity assumption is introduced that, if adopted, enables us to achieve better results.

**Assumption 5** (Strong convexity). *The function $f$ is $\mu$-strongly convex, $\mu > 0$. That is*

$$f(x) - f(y) - \langle \nabla f(y), x - y \rangle \ge \frac{\mu}{2} \|x - y\|^2, \quad \forall x, y \in \mathbb{R}^d.$$

We first present our algorithm FedExProx as Algorithm 1. In the subsequent sections, we first present the theory in the stochastic setting for FedExProx with a fixed extrapolation parameter in Section 3. Then we proceed to adaptive versions of our algorithm which eliminates the dependence on the unknown smoothness constant in Section 4.

## 3 Constant extrapolation

In order to demonstrate the convergence result of our algorithm in the stochastic setting, we use $\tau$-nice sampling as the way of selecting clients for partial participation. This refers to that in each iteration, the server samples a set $S_k \subseteq \{1, 2, \ldots, n\}$ uniformly at random from all subsets of size $\tau$. We want to emphasize that the sampling strategy here is merely an example, it is possible to use other client sampling strategies.

**Theorem 1.** *Suppose Assumption 1 (Differentiability), Assumption 2 (Interpolation regime), Assumption 3 (Convexity) and Assumption 4 (Smoothness) hold. If we use a fixed extrapolation parameter $\alpha_k = \alpha \in \left(0, \frac{2}{\gamma L_{\gamma,\tau}}\right)$ and any step size $0 < \gamma < +\infty$, then the average iterate of Algorithm 1 satisfies*

$$\mathbb{E}\left[f(\bar{x}_K)\right] - \inf f \leq C\left(\gamma, \tau, \alpha\right) \cdot \frac{\|x_0 - x_\star\|^2}{K},$$

*where $K$ is the number of iteration, $\bar{x}_K$ is sampled uniformly at random from the first $K$ iterates $\{x_0, x_1, \ldots, x_{K-1}\}$, $C\left(\gamma, \tau, \alpha\right)$ is defined as*

$$C\left(\gamma, \tau, \alpha\right) := \frac{1 + \gamma L_{\max}}{\alpha\gamma\left(2 - \alpha\gamma L_{\gamma,\tau}\right)} \quad and \quad L_{\gamma,\tau} := \frac{n - \tau}{\tau(n-1)}\frac{L_{\max}}{1 + \gamma L_{\max}} + \frac{n(\tau - 1)}{\tau(n-1)}L_\gamma,$$

*where $L_{\max} = \max_i L_i$, $L_\gamma$ is the smoothness constant of $M^\gamma\left(x\right) := \frac{1}{n}\sum_{i=1}^n M_{f_i}^\gamma\left(x\right)$. If we fix $\gamma$ and $\tau$ the optimal constant extrapolation parameter is given by $\alpha_{\gamma,\tau} := \frac{1}{\gamma L_{\gamma,\tau}} > 1$, which results in the following convergence guarantee:*

$$\mathbb{E}\left[f(\bar{x}_K)\right] - \inf f \leq C(\gamma, \tau, \alpha_{\gamma,\tau}) \cdot \frac{\|x_0 - x_\star\|^2}{K} = L_{\gamma,\tau}\left(1 + \gamma L_{\max}\right) \cdot \frac{\|x_0 - x_\star\|^2}{K}.$$

The proof of this theorem relies on the reformulation of the update rule in (7), using the identity $\nabla M_{f_i}^\gamma\left(x\right) = \frac{1}{\gamma}\left(x - \operatorname{prox}_{\gamma f_i}\left(x\right)\right)$ given in Lemma 2, which holds for any $x \in \mathbb{R}^d$, into the following form:

$$x_{k+1} = x_k - \alpha_k \cdot \gamma \cdot \frac{1}{\tau}\sum_{i \in S_k} \nabla M_{f_i}^\gamma\left(x_k\right). \tag{8}$$

We can then apply our modified theory for `SGD` given in Theorem 3, which is adapted from Gower et al. [2019], to obtain function value suboptimality in terms of $M^\gamma\left(x\right)$. The results are then translated back to function value suboptimality in terms of $f$. Note that (8) unveils the connection between the step size of gradient type methods and extrapolation parameter in our case.

**Remark 1.** *Theorem 1 provides convergence guarantee for Algorithm 1 in the convex case. If in addition, we assume Assumption 5 (Strong convexity) holds, the rate can be improved and we obtain linear convergence. See Corollary 1 for the details.*

**Remark 2.** *Theorem 1 indicates convergence for any $0 < \gamma < +\infty$. Indeed, as it is proved by Lemma 7, we have $C\left(\gamma, \tau, \alpha_{\gamma,\tau}\right) = L_{\gamma,\tau}\left(1 + \gamma L_{\max}\right) \leq L_{\max}$ holds for any $0 < \gamma < +\infty$. In cases where there exists at least one $L_i < L_{\max}$, we have $C\left(\gamma, \tau, \alpha_{\gamma,\tau}\right) < L_{\max}$.*

**Remark 3.** *One may question the necessity of the interpolation regime assumption. This assumption is crucial to our analysis. Besides allowing us to revisit the convex feasibility problem setting, it also guarantees that $M^\gamma\left(x\right)$ has the same set of minimizers as $f(x)$ as illustrated by Lemma 8. It also allows us to improve the upper bound on the step size by a factor of 2 in the `SGD` theory, which is demonstrated in Theorem 3 in the Appendix.*

**Remark 4.** *From the reformulation presented in (8), we see the best extrapolation parameter is obtained when $\alpha_k\gamma$ is the best step size for `SGD` running on global objective $M^\gamma\left(x\right)$. Since the best step size is affected by the smoothness and the minibatch size, so is the best extrapolation parameter.*

We can also compare our algorithm with `FedProx` in the convex overparameterized regime.

**Remark 5.** *Our algorithm includes `FedProx` as a special case when $\alpha = 1$. To recover its result, we simply plug in $\alpha = 1$, the resulting condition number is $C(\gamma, \tau, 1) = \frac{1 + \gamma L_{\max}}{\gamma(2 - \gamma L_{\gamma,\tau})}$. Compared to `FedProx`, Algorithm 1 with the same $\gamma > 0$ demonstrates superior performance, with the acceleration factor being quantified by*

$$\frac{C(\gamma, \tau, 1)}{C\left(\gamma, \tau, \alpha_{\gamma,\tau}\right)} \geq 2 + \frac{1}{\gamma L_{\max}} + \gamma L_{\max} \geq 4.$$

*See Lemma 14 for the proof. This suggests that the approach of the server averaging all iterates following local computation is suboptimal.*

In the following paragraphs, we study some special cases,

**Full participation case**   For the full participation case ($\tau = n$), using definition from Theorem 1

$$\alpha_{\gamma,n} = \frac{1}{\gamma L_\gamma} > 1, \quad L_{\gamma,n} = L_\gamma, \quad C(\gamma, n, \alpha_{\gamma,n}) = L_\gamma(1 + \gamma L_{\max}) \leq L_{\max}. \tag{9}$$

In this case, we can compare our method with `FedExP` in the convex overparameterized setting.

**Remark 6.** *Assume the conditions in Theorem 1 hold, the worst case iteration complexity of `FedExP` is given by $\mathcal{O}\left(\frac{L_{\max}}{\epsilon}\right)$, while for Algorithm 1, it is $\mathcal{O}\left(\frac{C(\gamma,n,\alpha_{\gamma,n})}{\epsilon}\right)$. As suggested by Lemma 7, Algorithm 1 has a better iteration complexity ($C(\gamma, n, \alpha_{\gamma,n}) < L_{\max}$) whenever there exists $L_i \neq L_{\max}$ for some $i \in [n]$, and the acceleration could reach up to a factor of $n$ as suggested by Example 1. In general, the speed-up in the worst case is quantified by*

$$\frac{L_{\max}}{1 + \gamma L_{\max}} \cdot \left(\frac{1}{n}\sum_{i=1}^{n}\frac{L_i}{1 + \gamma L_i}\right)^{-1} \leq \frac{L_{\max}}{C(\gamma, n, \alpha_{\gamma,n})} \leq n \cdot \frac{L_{\max}}{1 + \gamma L_{\max}} \cdot \left(\frac{1}{n}\sum_{i=1}^{n}\frac{L_i}{1 + \gamma L_i}\right)^{-1}.$$

**Single client case**   For the single client case ($\tau = 1$), using definition from Theorem 1

$$\alpha_{\gamma,1} = 1 + \frac{1}{\gamma L_{\max}} > 1, \quad L_{\gamma,1} = \frac{L_{\max}}{1 + \gamma L_{\max}}, \quad C(\gamma, 1, \alpha_{\gamma,1}) = L_{\max}.$$

**Remark 7.** *Compared with full and partial client participation, the following relations hold for any $\tau \in [n]$,*

$$C(\gamma, n, \alpha_{\gamma,n}) \leq C(\gamma, \tau, \alpha_{\gamma,\tau}) \leq C(\gamma, 1, \alpha_{\gamma,1}) \quad \text{and} \quad \alpha_{\gamma,1} \leq \alpha_{\gamma,\tau} \leq \alpha_{\gamma,n}, \quad \forall \tau \in [n].$$

*Since the iteration complexity of `FedExProx` is given by $\mathcal{O}\left(\frac{C(\gamma,\tau,\alpha_{\gamma,\tau})}{\epsilon}\right)$, the above inequalities tell us a larger client minibatch size $\tau$ leads to a larger extrapolation and a better iteration complexity. Specifically, Lemma 7 suggests the improvement over the single client case could be as much as a factor of $n$ ($C(\gamma, n, \alpha_{\gamma,n}) = \frac{1}{n}C(\gamma, 1, \alpha_{\gamma,1})$) as suggested by Example 1.*

## 4   Adaptive extrapolation

Observe that in Theorem 1, in order to determine the optimal extrapolation, we require the knowledge of $L_{\gamma,\tau}$, which is typically unknown. Although theoretically it suggests that simply averaging the iterates may result in suboptimal performance, in practice, this implication is less significant. To address this issue, we introduced two variants of `FedExProx`, based on gradient diversity and stochastic Polyak step size, given their relation to the extrapolation parameter in our cases.

**Theorem 2.** *Suppose Assumption 1 (Differentiability), Assumption 2 (Interpolation regime), Assumption 3 (Convexity) and Assumption 4 (Smoothness) hold.*

*(i) (`FedExProx-GraDS`): If we are using $\alpha_k = \alpha_{k,G}$, where*

$$\alpha_{k,G} := \frac{\frac{1}{n}\sum_{i=1}^{n}\left\|x_k - \text{prox}_{\gamma f_i}(x_k)\right\|^2}{\left\|\frac{1}{n}\sum_{i=1}^{n}\left(x_k - \text{prox}_{\gamma f_i}(x_k)\right)\right\|^2} \geq 1, \tag{10}$$

*then the iterates of Algorithm 1 with $\tau = n$ satisfy*

$$\mathbb{E}\left[f(\bar{x}_K)\right] - \inf f \leq \frac{1 + \gamma L_{\max}}{2 + \gamma L_{\max}} \cdot \left(\frac{1}{\gamma} + L_{\max}\right) \cdot \frac{\|x_0 - x_\star\|^2}{\sum_{k=0}^{K-1}\alpha_{k,G}},$$

*where $\bar{x}_K$ is chosen randomly from the first $K$ iterates $\{x_0, x_1, ..., x_{K-1}\}$ with probabilities $p_k = \alpha_{k,G}/\sum_{k=0}^{K-1}\alpha_{k,G}$.*

*(ii) (`FedExProx-StoPS`): If we are using $\alpha_k = \alpha_{k,S}$, where,*

$$\alpha_{k,S} := \frac{\frac{1}{n} \sum_{i=1}^n \left( M_{f_i}^\gamma (x_k) - \inf M_{f_i}^\gamma \right)}{\gamma \left\| \frac{1}{n} \sum_{i=1}^n \nabla M_{f_i}^\gamma (x_k) \right\|^2} \geq \frac{1}{2\gamma L_\gamma}, \tag{11}$$

*then the iterates of Algorithm 1 with $\tau = n$ satisfy*

$$\mathbb{E}\left[ f(\bar{x}_K) \right] - \inf f \leq \left( \frac{1}{\gamma} + L_{\max} \right) \cdot \frac{\|x_0 - x_\star\|^2}{\sum_{k=0}^{K-1} \alpha_{k,S}}, \tag{12}$$

*where $\bar{x}_K$ is chosen randomly from the first $K$ iterates $\{x_0, x_1, ..., x_{K-1}\}$ with probabilities $p_k = \alpha_{k,S} / \sum_{k=0}^{K-1} \alpha_{k,S}$.*

Theorem 2 describes the convergence in the full participation setting. However, we can also extend it to the stochastic setting by implementing a stochastic version of these adaptive step size rules for gradient-based methods [Horváth et al., 2022, Loizou et al., 2021]. See Theorem 5 in the Appendix for the details.

**Remark 8.** *In fact, the adaptive rule based on gradient diversity can be improved by using $\frac{L_{\max}}{1+\gamma L_{\max}}$ instead of $\frac{1}{\gamma}$ as the maximum of local smoothness constant of Moreau envelops, resulting in the extrapolation,*

$$\alpha_k = \alpha'_{k,G} := \frac{1 + \gamma L_{\max}}{\gamma L_{\max}} \cdot \frac{\frac{1}{n} \sum_{i=1}^n \left\| x_k - \mathrm{prox}_{\gamma f_i} (x_k) \right\|^2}{\left\| \frac{1}{n} \sum_{i=1}^n \left( x_k - \mathrm{prox}_{\gamma f_i} (x_k) \right) \right\|^2}. \tag{13}$$

*One can obtain a slightly better convergence guarantee than the `FedExProx-GraDS` case in Theorem 2, see Corollary 2 in the Appendix. However, the requires the knowledge of $L_{\max}$ in order to compute $\frac{1+\gamma L_{\max}}{\gamma L_{\max}}$.*

**Remark 9.** *Note that, compared to classical gradient-based methods, `FedExProx-GraDS` benefits from "semi-adaptivity". This refers to the fact that the algorithm converges for any choice of $\gamma > 0$. Although a smaller $\gamma$ hinders convergence, setting it to at least $\frac{1}{L_{\max}}$ limits the worsening of the convergence to a factor of 2.*

**Remark 10.** *Compared to `FedExProx` with the optimal constant extrapolation parameter, we gain "semi-adaptivity" here by using the gradient diversity based extrapolation. However, this results in losing the favorable dependence of convergence on $L_\gamma$ and instead establishes a dependence on $L_{\max}$.*

**Remark 11.** *For `FedExProx-StoPS`, as it is suggested by Lemma 20, the convergence depends on the favorable smoothness constant $L_\gamma$, rather than on $L_{\max}$. However, this comes at the price of having to know the minimum of each individual Moreau envelope.*

For a detailed discussion of the adaptive variants of `FedExProx`, we refer the readers to Appendix F.5. Since one of our starting points is the `RPM` by Necoara et al. [2019] to solve the convex feasibility problem with non-smooth local objectives, we have also adapted our method to non-smooth cases, as detailed in Theorem 4 in the Appendix. We also provided a discussion of our method in the non-interpolated setting and in the non-convex setting in Appendix F.

Finally, we support our findings with experiments, see Figure 1 for a simple experiment confirming that `FedExProx` indeed has a better iteration complexity than `FedProx`. For more details on the experiments, we refer the readers to Appendix I in the Appendix. Notice that in practice, each local proximity operator can be solved using different oracles. Clients may use `GD` or `SGD` to solve the local problem to a certain accuracy. The complexity of this subroutine depends on the local stepsize. If $\gamma$ is large, the local problem becomes harder to solve because we aim to minimize the local objective itself. Conversely, if it is small, the problem is easier since we do not stray far from the current iterate. As the choice of subroutine affects local computation complexity, comparing it directly with `FedExP` becomes complicated. Therefore, we compare the iteration complexity (number of communication rounds) of the two algorithms, assuming efficient local computations are carried out by the clients.

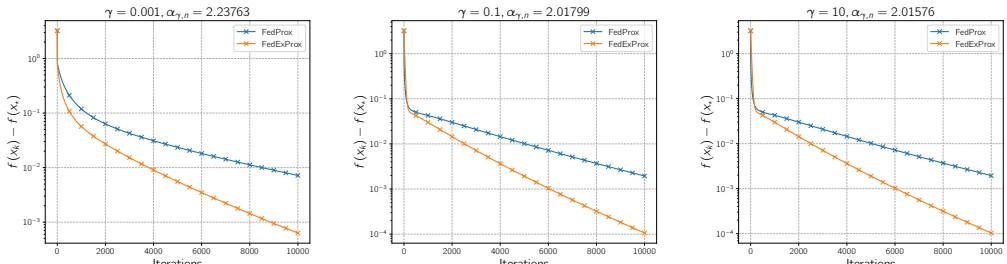

Figure 1: Comparison of `FedExProx` and `FedProx` in terms of iteration complexity in the full participation setting. The notation $\gamma$ here denotes the local step size of the proximity operator and $\alpha_{\gamma,n}$ is the corresponding optimal extrapolation parameter computed in (9) in the full participation case. In all cases, our proposed algorithm outperforms `FedProx`, suggesting that the practice of simply averaging the iterates is suboptimal.

## 5 Conclusion

### 5.1 Limitations

Our analysis of `FedExProx` serves as an initial step in adding extrapolation to `FedProx`, which currently relies on the suboptimal practice of the server merely averaging the iterates. While we discuss the behavior of our algorithm in non-interpolated and non-convex scenarios, our analysis only validates the effectiveness of extrapolation under the interpolation regime assumption.

### 5.2 Future Work

As we have just mentioned, extending our method and analysis beyond interpolation and convex regime is intriguing. In this case, new techniques may be needed for variance reduction. It is also interesting to investigate whether extrapolation can be applied together with client-specific personalization.

## Acknowledgement

The research reported in this publication was supported by funding from King Abdullah University of Science and Technology (KAUST): i) KAUST Baseline Research Scheme, ii) Center of Excellence for Generative AI, under award number 5940, iii) SDAIA-KAUST Center of Excellence in Artificial Intelligence and Data Science. The work was done during Kirill Acharya's internship at KAUST. Kirill Acharya was also affiliated with MIPT, ISP RAS, Russia. The work of Kirill Acharya was also supported by Grant App. No. 2 to Agreement No. 075-03-2024-214.

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

# Contents

## A   Notations

Throughout the paper, we use the notation $\|\cdot\|$ to denote the standard Euclidean norm defined on $\mathbb{R}^d$ and $\langle\cdot,\cdot\rangle$ to denote the standard Euclidean inner product. Given a differentiable function $f : \mathbb{R}^d \mapsto \mathbb{R}$, its gradient is denoted as $\nabla f(x)$. For a convex function $f : \mathbb{R}^d \mapsto \mathbb{R}$, we use $\partial f(x)$ to denote its subdifferential at $x$. We use the notation $D_f(x, y)$ to denote the Bregman divergence associated with a function $f : \mathbb{R}^d \mapsto \mathbb{R}$ between $x$ and $y$. The notation $\inf f$ is used to denote the minimum of a function $f : \mathbb{R}^d \mapsto \mathbb{R}$. We use $\mathrm{prox}_{\gamma f}(x)$ to denote the proximity operator of function $f : \mathbb{R}^d \mapsto \mathbb{R}$ with $\gamma > 0$ at $x \in \mathbb{R}^d$, and $M_f^\gamma(x)$ to denote the corresponding Moreau Envelope. The notation $\square$ is used for the infimal convolution of two proper functions. We denote the average of the Moreau envelope of each local objective $f_i$ by the notation $M^\gamma : \mathbb{R}^d \mapsto \mathbb{R}$. Specifically, we define $M^\gamma(x) = \frac{1}{n}\sum_{i=1}^n M_f^\gamma(x)$. Note that $M^\gamma(x)$ has an implicit dependence on $\gamma$, its smoothness constant is denoted by $L_\gamma$. We say an extended real-valued function $f : \mathbb{R}^d \mapsto \mathbb{R} \cup \{+\infty\}$ is proper if there exists $x \in \mathbb{R}^d$ such that $f(x) < +\infty$. We say an extended real-valued function $f : \mathbb{R}^d \mapsto \mathbb{R} \cup \{+\infty\}$ is closed if its epigraph is a closed set. The following Table 3 summarizes the commonly used notations and quantities appeared in this paper.

## B   Basic Facts

**Fact 1** (First prox theorem). *[Beck, 2017, Theorem 6.3] Let $f : \mathbb{R}^d \mapsto \mathbb{R}$ be a proper, closed and convex function. Then $\mathrm{prox}_f(x)$ is a singleton for any $x \in \mathbb{R}^d$.*

**Fact 2** (Second prox theorem). *[Beck, 2017, Theorem 6.39] Let $f : \mathbb{R}^d \mapsto \mathbb{R} \cup \{+\infty\}$ be a proper, closed and convex function. Then for any $x, u \in \mathbb{R}^d$, the following three claims are equivalent:*

*(i)  $u = \mathrm{prox}_f(x)$.*

*(ii)  $x - u \in \partial f(u)$.*

*(iii)  $\langle x - u, y - u\rangle \leq f(y) - f(u)$ for any $y \in \mathbb{R}^d$.*

**Fact 3** (Bregman divergence). *The Bregman divergence associated with a function $f$ between $x, y \in \mathbb{R}^d$ is defined as,*

$$D_f(x, y) := f(x) - f(y) - \langle \nabla f(y), x - y\rangle. \tag{14}$$

*If $f$ is convex, then for any $x, y \in \mathbb{R}^d$*

$$D_f(x, y) \geq 0. \tag{15}$$

Table 3: Summary of frequently used notations and quantities in this paper.

| Notations | Explanation |
|---|---|
| $n$ | The total number of clients. |
| $d$ | The dimension of the model. |
| $x$ | The model which belongs to $\mathbb{R}^d$. |
| $K$ | The total number of iterations. |
| $x_k$ | The model at $k$-th iteration. |
| $\alpha_k$ | The extrapolation parameter at iteration $k$. |
| $f_i(x)$ | Each local objective function. |
| $\gamma$ | The step size in the proximity operator. |
| $f(x)$ | The global objective $f$. |
| $\mathrm{prox}_{\gamma f_i}(x)$ | The proximity operator associated with $f_i$ and $\gamma > 0$ at point $x \in \mathbb{R}^d$. |
| $M_{f_i}^\gamma(x)$ | The Moreau envelope associated with $f_i$ and $\gamma > 0$ at point $x \in \mathbb{R}^d$. |
| $M^\gamma(x)$ | The average of $M_{f_i}^\gamma(x)$. |
| $L_i$ | The smoothness constant of $f_i$. |
| $L$ | The smoothness constant of $f$. |
| $\mu$ | The strong convexity constant of $f$. |
| $L_i/(1+\gamma L_i)$ | The smoothness constant of $M_{f_i}^\gamma$ |
| $L_{\max}$ | The maximum of all $L_i$, for $i \in [n]$. |
| $L_{\max}/(1+\gamma L_{\max})$ | The maximum of the smoothness constant of each $M_{f_i}^\gamma$ for $i \in [n]$. |
| $L_\gamma$ | The smoothness constant of $M^\gamma$. |
| $L_{\gamma,\tau}$ | The interpolation between the $L_\gamma$ and $L_{\max}/(1+\gamma L_{\max})$ induced by $\tau$-nice sampling. |
| $\alpha_{\gamma,\tau}$ | The optimal extrapolation parameter of `FedExProx` under $\tau$-nice sampling. |
| $C(\gamma,\tau,\alpha)$ | The convergence rate of `FedExProx` with $\tau$-nice sampling in the convex case. |
| $\alpha_{k,G}$ | The gradient diversity extrapolation in the $k$-th iteration defined in Theorem 2. |
| $\alpha_{k,S}$ | The stochastic Polyak extrapolation in the $k$-th iteration defined in Theorem 2. |
| $\alpha_{k,G}'$ | The improved gradient diversity based extrapolation used in Corollary 2. |
| $D_f(x,y)$ | The Bregman divergence associated with $f$ between two points $x, y \in \mathbb{R}^d$. |
| $S_k$ | The set of indices server sampled in the $k$-th iteration. |
| $\alpha_{\tau,k,G}$ | The gradient diversity based extrapolation in the $k$-th iteration for `FedExProx-GraDS-PP`. |
| $\alpha_{\tau,k,S}$ | The stochastic Polyak based extrapolation in the $k$-th iteration for `FedExProx-StoPS-PP`. |

*If $f$ is convex, $L$-smooth and differentiable, the following inequalities hold for any $x, y \in \mathbb{R}^d$,*

$$\frac{1}{L}\left\|\nabla f(x) - \nabla f(y)\right\|^2 \le D_f(x,y) + D_f(y,x) \le L\left\|x - y\right\|^2,$$

$$\frac{1}{L}\left\|\nabla f(x) - \nabla f(y)\right\|^2 \le 2D_f(x,y) \le L\left\|x - y\right\|^2. \tag{16}$$

**Fact 4** (Increasing function). *Let $f(x) = \frac{x}{1+\gamma x}$, where $\gamma > 0$. Then $f(x)$ is monotone increasing when $x > 0$.*

## C  Properties of Moreau envelope

In this section, we explore the properties of the Moreau envelope of individual functions $f_i$, and the global objective $M^\gamma = \frac{1}{n}\sum_{i=1}^n M_{f_i}^\gamma$. Before that, we present the definition of infimal convolution

**Definition 3** (Infimal convolution). *The infimal convolution of two proper functions $f, g : \mathbb{R}^d \mapsto \mathbb{R} \cup \{+\infty\}$ is defined via the following formula*

$$(f\square g)(x) = \min_{z \in \mathbb{R}^d}\{f(z) + g(x - z)\}.$$

One key observation is that $M_f^\gamma$ can be viewed as the infimal convolution of the proper, closed and convex function $f$ and the real-valued convex function $\frac{1}{2\gamma}\left\|\cdot\right\|^2$. This observation enables us to infer the convexity and smoothness of the Moreau envelope from the properties of the original function.

First, we present two lemmas about basic properties of Moreau envelope.

**Lemma 1** (Real-valuedness). *Let $f : \mathbb{R}^d \mapsto \mathbb{R} \cup \{+\infty\}$ be a proper, closed and convex function. Then its Moreau envelope $M_f^\gamma$ for any $\gamma > 0$ is a real-valued function. In particular, the following identity holds for $x \in \mathbb{R}^d$ according to the definition of Moreau envelope,*

$$M_f^\gamma(x) = f\left(\mathrm{prox}_{\gamma f}(x)\right) + \frac{1}{2\gamma}\left\|x - \mathrm{prox}_{\gamma f}(x)\right\|^2.$$

**Lemma 2** (Differentiability of Moreau envelope). *[Beck, 2017, Theorem 6.60] Let $f : \mathbb{R}^d \mapsto \mathbb{R} \cup \{+\infty\}$ be a proper, closed and convex function. Then its Moreau envelope $M_f^\gamma$ for any $\gamma > 0$ is $\frac{1}{\gamma}$-smooth, and for any $x \in \mathbb{R}^d$, we have*

$$\nabla M_f^\gamma(x) = \frac{1}{\gamma}\left(x - \mathrm{prox}_{\gamma f}(x)\right).$$

We then focus on the relation between individual $f_i$ and $M_{f_i}^\gamma$. The following lemma suggests that the convexity of individual $f_i$ guarantees the convexity of $M_{f_i}^\gamma$.

**Lemma 3** (Convexity of Moreau envelope). *[Beck, 2017, Theorem 6.55] Let $f : \mathbb{R}^d \mapsto \mathbb{R} \cup \{+\infty\}$ be a proper and convex function. Then $M_f^\gamma$ is a convex function.*

It is also true that the smoothness of individual $f_i$ indicates the smoothness of $M_{f_i}^\gamma$.

**Lemma 4** (Smoothness of Moreau envelope). *Let $f : \mathbb{R}^d \mapsto \mathbb{R}$ be a convex and $L$-smooth function. Then $M_f^\gamma$ is $\frac{L}{1+\gamma L}$-smooth.*

One notable fact is that $f_i$ and $M_{f_i}^\gamma$ have the same set of minimizers.

**Lemma 5** (Minimizer equivalence). *Let $f : \mathbb{R}^d \mapsto \mathbb{R} \cup \{+\infty\}$ be a proper, closed and convex function. Then for any $\gamma > 0$, $f$ and $M_f^\gamma$ has the same set of minimizers.*

In addition, $M_f^\gamma$ is a global lower bound of $f$.

**Lemma 6** (Individual lower bound). *Let $f : \mathbb{R}^d \mapsto \mathbb{R} \cup \{+\infty\}$ be a proper, closed and convex function. Then the Moreau envelope $M_f^\gamma$ satisfies $M_f^\gamma(x) \le f(x)$ for all $x \in \mathbb{R}^d$.*

Next, we focus on the global objective $M^\gamma(x)$. The following lemma bounds its smoothness constant from both above and below.

**Lemma 7** (Global convexity and smoothness). *Let each $f_i$ be proper, closed convex and $L_i$-smooth. Then $M$ is convex and $L_\gamma$-smooth with*

$$\frac{1}{n^2}\sum_{i=1}^n \frac{L_i}{1+\gamma L_i} \le L_\gamma \le \frac{1}{n}\sum_{i=1}^n \frac{L_i}{1+\gamma L_i}.$$

*As a result of the above inequalities, we have the following inequality on the condition number defined in Theorem 1 which holds for any $\tau \in [n]$,*

$$L_\gamma(1 + \gamma L_{\max}) = C(\gamma, n, \alpha_{\gamma,n}) \le C(\gamma, \tau, \alpha_{\gamma,\tau}) \le C(\gamma, 1, \alpha_{\gamma,1}) = L_{\max}.$$

*When there exists at least one $L_i < L_{\max}$, we have $C(\gamma, n, \alpha_{\gamma,n}) < C(\gamma, \tau, \alpha_{\gamma,\tau}) < L_{\max} = C(\gamma, 1, \alpha_{\gamma,1})$. Even $L_i = L_{\max}$ holds for all $i \in [n]$, there are cases (See Example 1 in the proof.) that $C(\gamma, n, \alpha_{\gamma,n}) = \frac{1}{n}C(\gamma, 1, \alpha_{\gamma,1}) = \frac{1}{n}L_{\max}$.*

A key observation in this case is the generalization of Lemma 5 into the finite-sum setting under the interpolation regime.

**Lemma 8** (Minimizer equivalence). *If we let every $f_i : \mathbb{R}^d \mapsto \mathbb{R} \cup \{+\infty\}$ be proper, closed and convex, then $f(x) = \frac{1}{n}\sum_{i=1}^n f_i(x)$ has the same set of minimizers and minimum as*

$$M^\gamma(x) = \frac{1}{n}\sum_{i=1}^n M_{f_i}^\gamma(x),$$

*if we are in the interpolation regime and $0 < \gamma < \infty$.*

The following lemma generalizes Lemma 6 into the finite-sum setting.

**Lemma 9** (Global lower bound)**.** *Let each $f_i : \mathbb{R}^d \mapsto \mathbb{R} \cup \{+\infty\}$ be proper, closed and convex. Then the following inequality holds for any $x \in \mathbb{R}^d$ and $\gamma > 0$,*

$$M^\gamma (x) \leq M_f^\gamma (x) \leq f(x).$$

*In addition, if we assume we are in the interpolation regime, then $M^\gamma$, $M_f^\gamma$ and $f$ have the same set of minimizers, for any $x_\star$ in this set of minimizers, the following identity holds,*

$$M^\gamma (x_\star) = M_f^\gamma (x_\star) = f(x_\star).$$

Besides the global lower bound provided above, there is also a relation between the function value suboptimality of $M^\gamma$ and $f$.

**Lemma 10** (Suboptimality bound)**.** *Suppose Assumption 1 (Differentiability), 2 (Interpolation Regime), 3 (Convexity) and 4 (Smoothness) hold, for any minimizer $x_\star$ of $M^\gamma (x)$, all $x \in \mathbb{R}^d$, the following inequality holds for each client objective,*

$$M_{f_i}^\gamma (x) - M_{f_i}^\gamma (x_\star) \geq \frac{1}{1 + \gamma L_i} \left( f_i(x) - f_i(x_\star) \right). \tag{17}$$

*Furthermore, this suggests*

$$M^\gamma (x) - M^\gamma (x_\star) \geq \frac{1}{1 + \gamma L_{\max}} \left( f_i(x) - f_i(x_\star) \right). \tag{18}$$

A direct consequence of the above function suboptimality bound is the star strong convexity of $M^\gamma$ from the strong convexity of $f$.

**Lemma 11.** *(Star strong convexity) Assume Assumption 1 (Differentiability), Assumption 2 (Interpolation Regime), Assumption 3 (Convexity), Assumption 4 (Smoothness) and Assumption 5 (Strong convexity) hold, then the convex function $M^\gamma (x)$ satisfies the following inequality,*

$$M^\gamma (x) - M^\gamma (x_\star) \geq \frac{\mu}{1 + \gamma L_{\max}} \cdot \frac{1}{2} \| x - x_\star \|^2 ,$$

*for any $x \in \mathbb{R}^d$ and a minimizer $x_\star$ of $M^\gamma (x)$.*

The star strong convexity property of $M^\gamma$ allows us to improve the sublinear convergence in the convex regime into linear convergence.

## D   Technical lemmas

**Lemma 12.** *Let $f : \mathbb{R}^d \mapsto \mathbb{R}$ be a proper, closed and convex function. Then $x$ is a minimizer of $f$ if and only if $x = \mathrm{prox}_{\gamma f} (x)$.*

**Lemma 13.** *Assume we are working with the finite-sum problem $f = \frac{1}{n} \sum_{i=1}^n f_i$, where each $f_i$ is convex and $L_i$-smooth, $f$ is convex and $L$-smooth. Then the smoothness of $L$ satisfies*

$$\frac{1}{n^2} \sum_{i=1}^n L_i \leq L \leq \frac{1}{n} \sum_{i=1}^n L_i,$$

*where both bounds are attainable.*

**Lemma 14.** *Assume that all the conditions mentioned in Theorem 1 hold, then the condition number $C(\gamma, \tau, 1)$ of `FedProx` and the condition number $C(\gamma, \tau, \alpha_{\gamma,\tau})$ of the optimal constant extrapolation parameter $\alpha_\star = \frac{1}{\gamma L_{\gamma,\tau}}$ satisfy the following inequality,*

$$\frac{C(\gamma, \tau, 1)}{C(\gamma, \tau, \alpha_{\gamma,\tau})} \geq 2 + \frac{1}{\gamma L_{\max}} + \gamma L_{\max} \geq 4 \quad \forall \tau \in [n].$$

**Lemma 15.** *Assume that all the conditions mentioned in Theorem 1 hold, then the following inequalities hold,*

$$C(\gamma, n, \alpha_{\gamma,n}) \leq C(\gamma, \tau, \alpha_{\gamma,\tau}) \leq C(\gamma, 1, \alpha_{\gamma,1}), \quad \forall \tau \in [n],$$

*and*

$$\alpha_{\gamma,1} \geq \alpha_{\gamma,\tau} \geq \alpha_{\gamma,n}, \quad \forall \tau \in [n].$$

# E    Theory of SGD

In order to prove our main theorem, we partly rely on the theory of SGD. The following theorem on the convergence of SGD with $\tau$-nice sampling is adapted from Gower et al. [2019]. We introduce modifications to the proof technique and tailor the theorem specifically to the interpolation regime. In this context, the upper bound on the step size is increased by a factor of 2. We first formulate the algorithm as follows for completeness.

---

**Algorithm 2** SGD with $\tau$-nice sampling

---

1: **Parameters:** learning rate $\eta > 0$, starting point $x_0 \in \mathbb{R}^d$, minibatch size $\tau \in \{1, 2, \ldots, n\}$
2: **for** $k = 0, 1, 2, \ldots$ **do**
3:    The server samples $S_k \subseteq \{1, 2, \ldots, n\}$ uniformly from all subsets of cardinality $\tau$
4:    The server performs one gradient step

$$x_{k+1} = x_k - \eta \cdot \frac{1}{\tau} \sum_{\xi_i \in S_k} \nabla f_{\xi_i}(x_k).$$

5: **end for**

---

**Theorem 3.** *Assume Assumption 1 (Differentiability), 2 (Interpolation regime), 3 (Convexity), 4 (Smoothness) hold. Additionally, assume $f$ is $L$-smooth where $L \leq \frac{1}{n} \sum_{i=1}^{n} L_i$.[3] If we are running* SGD *with $\tau$-nice sampling using step size $\eta$ that satisfies $0 < \eta < \frac{2}{A_\tau}$, where*

$$A_\tau := \frac{n-\tau}{\tau(n-1)} L_{\max} + \frac{n(\tau-1)}{\tau(n-1)} L, \qquad and \qquad L_{\max} := \max_i L_i,$$

*then the iterates of Algorithm 2 satisfy*

$$\mathbb{E}\left[f(\bar{x}_K)\right] - \inf f \leq \frac{1}{\eta(2 - \eta A_\tau)} \cdot \frac{\|x_0 - x_\star\|^2}{K},$$

*where $K$ is the total number of iterations, $\bar{x}_K$ is chosen uniformly at random from the first $K$ iterates $\{x_0, x_1, \ldots, x_{K-1}\}$. If, additionally, we assume the following property (which we will refer to as "star strong convexity") holds, then the iterates of Algorithm 2 satisfy*

$$\mathbb{E}\left[\|x_K - x_\star\|^2\right] \leq \left(1 - \eta(2 - \eta A_\tau) \cdot \frac{\mu}{2}\right)^K \|x_0 - x_\star\|^2.$$

# F    Additional analysis on FedExProx

In this section, we provide some additional details on the analysis of FedExProx and its adaptive variants.

## F.1    FedExProx in the strongly convex case

The following corollary summarizes the convergence guarantee in the strongly convex case.

**Corollary 1.** *Suppose the assumptions in Theorem 1 hold, and assume in addition that Assumption 5 (Strong Convexity) holds, then we achieve linear convergence for the final iterate of Algorithm 1, which satisfies*

$$\mathbb{E}\left[\|x_K - x_\star\|^2\right] \leq \left(1 - \alpha\gamma(2 - \alpha\gamma L_{\gamma,\tau}) \cdot \frac{\mu}{2\left(1 + \gamma L_{\max}\right)}\right)^K \|x_0 - x_\star\|^2,$$

*where the definition of $L_{\gamma,\tau}$ is given in Theorem 1. Fixing the choice of $\gamma$ and $\tau$, the optimal extrapolation parameter that minimizes the convergence rate is given by $\alpha_{\gamma,\tau} = \frac{1}{\gamma L_{\gamma,\tau}} > 1$, which results in the following convergence in the strongly convex case:*

$$\mathbb{E}\left[\|x_K - x_\star\|^2\right] \leq \left(1 - \frac{\mu}{2 L_{\gamma,\tau}\left(1 + \gamma L_{\max}\right)}\right)^K \|x_0 - x_\star\|^2.$$

---

[3]This is justified by Lemma 13.

As one can observe, by additionally assuming $\mu$ strong convexity of the original function $f$, we improve the sublinear convergence in the convex case into linear convergence.

### F.2 `FedExProx` in the non-smooth case

Our analysis also adapts to the non-smooth cases. This is based on the observation that even if we only assume Assumption 1 (differentiability), Assumption 2 (Interpolation Regime) and Assumption 3 (Convexity) hold and do not have additional assumptions on smoothness, still each $M_{f_i}^{\gamma}$ is $\frac{1}{\gamma}$-smooth because of Lemma 2. Thus, the theory of `SGD` in the convex smooth case still applies. However, there are some differences from the smooth case. For the sake of simplicity, we will mainly focus on the convex non-smooth case with a constant extrapolation parameter, the results in the strongly convex regime and with adaptive extrapolation can be obtained similarly as in the proof of Theorem 1 and Theorem 2.

**Theorem 4.** *Assume Assumption 1 (Differentiability), 2 (Interpolation Regime) and 3 (Convexity) hold. If we choose a constant extrapolation parameter $\alpha_k = \alpha$ satisfying*

$$0 < \alpha < \frac{2}{\gamma L_{\gamma,\tau}},$$

*where $L_\gamma$ is the smoothness constant of $M^\gamma(x) = \frac{1}{n} \sum_{i=1}^n M_{f_i}^\gamma(x)$, $L_{\gamma,\tau}$ is given by*

$$L_{\gamma,\tau} = \frac{n-\tau}{\tau(n-1)} \cdot \frac{1}{\gamma} + \frac{n(\tau-1)}{\tau(n-1)} \cdot L_\gamma.$$

*Then the iterates of Algorithm 1 satisfy*

$$\gamma M^\gamma(\bar{x}_K) - \inf \gamma M^\gamma \leq \frac{1}{\alpha(2 - \alpha\gamma L_{\gamma,\tau})} \cdot \frac{\|x_0 - x_\star\|^2}{K},$$

*where $\bar{x}_K$ is chosen uniformly from the first $K$ iterates $\{x_0, x_1, \ldots, x_{K-1}\}$. It is easy to see that the best $\alpha$ is given by*

$$\alpha_\star = \frac{1}{\gamma L_{\gamma,\tau}} \geq 1,$$

*where the corresponding convergence is given by*

$$\gamma M^\gamma(\bar{x}_K) - \inf \gamma M^\gamma \leq \left( \frac{n-\tau}{\tau(n-1)} + \frac{n(\tau-1)}{\tau(n-1)} \gamma L_\gamma \right) \cdot \frac{\|x_0 - x_\star\|^2}{K}.$$

**Remark 12.** *Notice that in this case we recover the convergence result of `RPM` presented in Necoara et al. [2019] in the convex case. Indeed, if each $f_i(x) = \mathbb{I}_{\mathcal{X}_i}(x)$, then we have*

$$\mathrm{prox}_{\gamma f_i}(x) = \Pi_{\mathcal{X}_i}(x), \forall x \in \mathbb{R}^d,$$

*and*

$$\gamma M_{f_i}^\gamma(x) = \frac{1}{2} \|x - \Pi_{\mathcal{X}_i}(x)\|^2, \quad \text{and} \quad \gamma M^\gamma(x) = \frac{1}{2} \cdot \frac{1}{n} \sum_{i=1}^n \|x - \Pi_{\mathcal{X}_i}(x)\|^2.$$

*Since we are in the interpolation regime, $\inf \gamma M^\gamma = 0$, and the convergence result becomes*

$$\frac{1}{2} \cdot \frac{1}{n} \sum_{i=1}^n \|x_K - \Pi_{\mathcal{X}_i}(x_K)\|^2 \leq \left( \frac{n-\tau}{\tau(n-1)} + \frac{n(\tau-1)}{\tau(n-1)} \gamma L_\gamma \right) \cdot \frac{\|x_0 - x_\star\|^2}{K}.$$

*Notice that here $\gamma L_\gamma \leq 1$ is the smoothness constant associated with each distance function $\frac{1}{2} \|x - \Pi_{\mathcal{X}_i}(x)\|^2$. The difference in the coefficients on the left-hand side from the original results presented in Necoara et al. [2019] results from different sampling strategies employed.*

A key difference in the non-smooth setting is that extrapolation in some cases may not be beneficiary, as illustrated by the following remark.

**Remark 13.** *In the non-smooth case, it is possible that $\gamma L_\gamma = 1$, where the optimal $\alpha_\star = 1$, in this case, extrapolation will not generate any benefits. However, as it is mentioned by Necoara et al. [2019], there are many examples where $\gamma L_\gamma < 1$ and extrapolation indeed accelerates the algorithm. This is different from the smooth case, where extrapolation always helps.*

**Remark 14.** *Since we do not assume smoothness, Lemma 10 no longer applies. Therefore, the convergence result is stated in terms of the function value suboptimality of Moreau envelope instead of the original objective $f$ which is used in the smooth case.*

Using a similar approach, it is also possible to obtain a convergence guarantee for `FedExProx` in the strongly convex non-smooth regime, assuming in addition that $M^\gamma(x)$ is $\mu_\gamma$-strongly convex, where we recover the convergence result of `RPM` in Necoara et al. [2019] in cases where the smooth and linear regularity conditions are both satisfied. The following Table 4 confirms that our analysis of `FedExProx` recovers the theory of `RPM` as a special case.

Table 4: Comparison of iteration complexity of `RPM` from Necoara et al. [2019] obtained using our theory and the original theory. In both cases, the optimal extrapolation parameter is used. The notation $\mathcal{O}(\cdot)$ is hidden. $\varepsilon$ is the error level reached by function value suboptimality for convex case, squared distance to the solution for strongly convex case.

| Setting | Original Theory | Our Theory |
|---|---|---|
| Convex + smooth case[1] | $\gamma L_{\gamma,\tau} \cdot \frac{\|x_0 - x_\star\|^2}{\varepsilon}$ | $\gamma L_{\gamma,\tau} \cdot \frac{\|x_0 - x_\star\|^2}{\varepsilon}$ |
| Strongly convex + smooth case[2] | $\frac{L_{\gamma,\tau}}{\mu_\gamma} \cdot \log\left(\frac{\|x_0 - x_\star\|^2}{\varepsilon}\right)$ | $\frac{L_{\gamma,\tau}}{\mu_\gamma} \cdot \log\left(\frac{\|x_0 - x_\star\|^2}{\varepsilon}\right)$ |

[1] The smoothness here does not refer to each $f_i$ being $L_i$-smooth, but $\gamma M^\gamma$ being $\gamma L_\gamma$-smooth. This corresponds to the smooth regularity condition presented in Necoara et al. [2019].

[2] Here the strongly convex setting meaning that the linear regularity condition in Necoara et al. [2019] is satisfied. In our theory, it refers to $M^\gamma(x)$ being $\mu_\gamma$-strongly convex with $\mu_\gamma < L_\gamma$.

### F.3 Discussion on the non-interpolation case

For the non-interpolation regime cases, we assume that Assumption 1 (Differentiability), Assumption 3 (Convexity) and Assumption 4 (Smoothness) hold. The differences are listed as follows

(i) Although $f_i$ and $M_{f_i}^\gamma$ have the same set of minimizers, $f$ and $M^\gamma$ does not necessarily have the same set of minimizers. This will lead to the convergence of `FedExProx` to the minimizer $x'_{\star,\gamma}$ of $M^\gamma(x)$ instead of $x_\star$ of $f$. As a result, we will only converge to a neighborhood of the $x_\star$ depending on the specific setting.

(ii) Since we are not in the interpolation regime, the upper bound on the step size of `SGD` with sampling is reduced by a factor of 2. Thus, the optimal extrapolation parameter $\alpha'_\star$ in the non-interpolated cases is also halved, $\alpha'_\star = \frac{1}{2}\alpha_\star$. As a result, it is possible that $\alpha'_\star \leq 1$. The same phenomenon is also observed in `FedExP` of Jhunjhunwala et al. [2023], where their heuristic in determining the extrapolation parameter adaptively is also reduced by a factor of 2 in non overparameterized cases.

Observe that all of the above results in both smooth/non-smooth, interpolated/non-interpolated cases suggests that the practice of server simply averaging the iterates it obtained from local training is suboptimal.

### F.4 Discussion on the non-convex case

In the non-convex case, we assume Assumption 1 (Differentiability) holds, and we need the following additional assumptions on $f : \mathbb{R}^d \mapsto \mathbb{R}$ and $f_i : \mathbb{R}^d \mapsto \mathbb{R}$:

**Assumption 6** (Lower boundedness). *Function $f_i$ is lower bounded by $\inf f_i$.*

**Assumption 7** (Weak convexity). *Function $f_i$ is $\rho > 0$ weakly convex, this means that $f_i + \frac{\rho}{2}\|\cdot\|^2$ is convex.*

We have the following lemma under the above assumptions:

**Lemma 16.** *[Böhm and Wright, 2021, Lemma 3.1] Let $f$ be a proper, closed, $\rho$-weakly convex function and let $\gamma < \frac{1}{\rho}$. Then the Moreau envelope $M_f^\gamma$ is continuously differentiable on $\mathbb{R}^d$ with*

$$\nabla M_f^\gamma(x) = \frac{1}{\gamma}\left(x - \text{prox}_{\gamma f}(x)\right).$$

*In addition, the Moreau envelope is* $\max\left\{\frac{1}{\gamma}, \frac{\rho}{1-\gamma\rho}\right\}$*-smooth. We will thereby denote the smoothness constant as* $L_{\gamma,\rho}$.

Indeed, if the stepsize $\gamma$ in this case is chosen properly such that $\frac{1}{\gamma} > \rho$, then it is straight forward to see the function within the proximity operator $\mathrm{prox}_{\gamma f_i}$ given by $f_i + \frac{1}{2} \cdot \frac{1}{\gamma} \left\|\cdot\right\|^2$ is strongly convex. Thus the proximity operator still results in a singleton. Lemma 16 allows us to again reformulate the original algorithm using the gradient of Moreau envelope. The only difference from the convex regime is that the Moreau envelope $M_{f_i}^{\gamma}$ is not necessarily convex. The following lemmas illustrate the connection between $M_{f_i}^{\gamma}$ and $f_i$:

**Lemma 17.** *[Yu et al., 2015, Proposition 7] Let $\gamma > 0$, $f$ be a closed, proper function that is lower bounded. Then $M_f^{\gamma} \leq f$, $\inf M_f^{\gamma} = \inf f$, $\arg\min_x M_f^{\gamma}(x) = \arg\min_x f(x) \subseteq \left\{x : x \in \mathrm{prox}_{\gamma f}(x)\right\}$.*

**Lemma 18.** *Let $f : \mathbb{R}^d \mapsto \mathbb{R}$ be $\rho$-weakly convex with $\rho > 0$ and differentiable. If we take $0 < \gamma < \frac{1}{\rho}$, then $M_{f_i}^{\gamma}$ has the same set of stationary points as $f_i$.*

For the sake of simplicity, we will consider only the full participation case with a constant extrapolation parameter $\alpha_k = \alpha$. The following lemma describes the convergence of GD in the non-convex case, which is adapted from the theory of Khaled and Richtárik [2023].

**Lemma 19.** *Assume function $f$ is $L$-smooth and lower bounded. If we are running GD with a constant stepsize $\eta$ satisfying $0 < \eta < \frac{1}{L}$. Then for any $K \geq 1$, the iterates $x_k$ of GD satisfy*

$$\min_{0 \leq k \leq K-1} \mathbb{E}\left[\left\|\nabla f(x_k)\right\|^2\right] \leq \frac{2\left(f(x_0) - \inf f\right)}{\eta K}.$$

Now we directly apply Lemma 19 in our case,

1. Since each $M_{f_i}^{\gamma}$ is $L_{\gamma,\rho}$-smooth, $M^{\gamma}$ is $L_{\gamma}$-smooth with $L_{\gamma} \leq L_{\gamma,\rho}$, which result in the following bound on the extrapolation parameter

$$0 < \alpha < \frac{1}{\gamma L_{\gamma}}.$$

   Notice that in this case we have the following estimation of $\gamma L_{\gamma}$,

$$\frac{1}{\gamma L_{\gamma}} \geq \frac{1}{\gamma L_{\gamma,\rho}} = \min\left\{1, \frac{1-\gamma\rho}{\gamma\rho}\right\}.$$

   This suggests that extrapolation may not be much beneficiary in the non-convex case.

2. The following convergence guarantee can be obtained.

$$\min_{0 \leq k \leq K-1} \mathbb{E}\left[\left\|\nabla M^{\gamma}(x_k)\right\|^2\right] \leq \frac{2\left(M^{\gamma}(x_0) - \inf M^{\gamma}\right)}{\alpha\gamma K}.$$

   Notice that by Lemma 17, we know that $M_{f_i}^{\gamma}(x_0) \leq f_i(x_0)$. We also have $\inf M^{\gamma} \geq \frac{1}{n}\sum_{i=1}^{n} \inf M_{f_i}^{\gamma} = \frac{1}{n}\sum_{i=1}^{n} \inf f_i$ since $\inf M_{f_i}^{\gamma} = \inf f_i$ is true for each client by Lemma 17. Thus, we have

$$M^{\gamma}(x_0) - \inf M^{\gamma} \leq f(x_0) - \inf f + \inf f - \frac{1}{n}\sum_{i=1}^{n} \inf f_i.$$

   We can relax the above convergence guarantee and obtain

$$\min_{0 \leq k \leq K-1} \mathbb{E}\left[\left\|\nabla M^{\gamma}(x_k)\right\|^2\right] \leq \frac{2\left(f(x_0) - \inf f\right)}{\alpha\gamma K} + \frac{2\left(\inf f - \frac{1}{n}\sum_{i=1}^{n} \inf f_i\right)}{\alpha\gamma K}.$$

   The above convergence guarantee indicates that the algorithm converges to some stationary points of $M^{\gamma}(x)$ in the non-convex case.

3. In the non-convex case, we did not assume anything similar to the interpolation regime in the convex case. As a result, we did not know the relation between the set of stationary points of $M^\gamma(x)$ and $f(x)$, denoted as $\mathcal{Y}'$ and $\mathcal{Y}$, respectively. However, if we assume, in addition, that each stationary point $y' \in \mathcal{Y}'$ of $M^\gamma$ is also a stationary point of each $M^\gamma_{f_i}$, then $y'$ is also a stationary point of $f_i$ according to Lemma 18. Thus, $\nabla f(y') = \frac{1}{n}\sum_{i=1}^{n} \nabla f_i(y') = 0$, which indicates $y' \in \mathcal{Y}$. As a result, we have $\mathcal{Y}' \subseteq \mathcal{Y}$. This means that under this additional assumption, the algorithm converges to a stationary point of $f$.

### F.5 Additional notes on adaptive variants

**Notes on gradient diversity variant.** In general, the gradient diversity step size $\eta_k$ used in SGD to solve the finite sum minimization problem

$$\min_{x\in\mathbb{R}^d}\left\{ f(x) = \frac{1}{n}\sum_{i=1}^{n} f_i(x) \right\},$$

can be written as

$$\eta_k := \frac{1}{L_{\max}} \cdot \frac{\frac{1}{n}\sum_{i=1}^{n}\left\|\nabla f_i(x_k)\right\|^2}{\left\|\frac{1}{n}\sum_{i=1}^{n}\nabla f_i(x_k)\right\|^2},$$

where $L_{\max}$ is the maximum of local smoothness constants. In our case, since each local Moreau envelope is $\frac{L_i}{1+\gamma L_i}$-smooth and $\frac{1}{\gamma}$-smooth[4], we can use both $\frac{L_{\max}}{1+\gamma L_{\max}}$ (here in Corollary 2, if we know $L_{\max}$) and $\frac{1}{\gamma}$ (in original Theorem 2, if we do not know $L_{\max}$) as the maximum of local smoothness. We present the convergence result of Algorithm 1 with the following rule given in (13),

$$\alpha'_{k,G} = \frac{1+\gamma L_{\max}}{\gamma L_{\max}} \cdot \frac{\frac{1}{n}\sum_{i=1}^{n}\left\|x_k - \mathrm{prox}_{\gamma f_i}(x_k)\right\|^2}{\left\|\frac{1}{n}\sum_{i=1}^{n}\left(x_k - \mathrm{prox}_{\gamma f_i}(x_k)\right)\right\|^2}.$$

**Corollary 2.** *Suppose all the assumptions mentioned in Theorem 2 hold, if we are using (13) to determine $\alpha'_{k,G}$ in each iteration for Algorithm 1 with $\tau = n$, then the iterates satisfy*

$$\mathbb{E}\left[f(\bar{x}_K)\right] - f^{\inf} \leq \left(\frac{1}{\gamma} + L_{\max}\right) \cdot \frac{\left\|x_0 - x_\star\right\|^2}{\sum_{k=0}^{K-1}\alpha'_{k,G}}.$$

*where $\bar{x}_K$ is chosen randomly from the first $K$ iterates $\{x_0, x_1, ..., x_{K-1}\}$ with probabilities $p_k = \alpha'_{k,G}/\sum_{k=0}^{K-1}\alpha'_{k,G}$.*

Notice that compared to the case of FedExProx-GraDS in Theorem 2, the convergence rate given in Corollary 2 is indeed better. This can be seen by comparing them directly, for FedExProx-GraDS, we have

$$\mathbb{E}\left[f(\bar{x}_K)\right] - \inf f \leq \frac{1+\gamma L_{\max}}{2+\gamma L_{\max}} \cdot \left(\frac{1}{\gamma} + L_{\max}\right) \cdot \frac{\left\|x_0 - x_\star\right\|^2}{\sum_{k=0}^{K-1}\alpha_{k,G}},$$

and for Algorithm 1 with $\alpha'_{k,G}$ given in (13), we have

$$\mathbb{E}\left[f(\bar{x}_K)\right] - f^{\inf} \leq \left(\frac{1}{\gamma} + L_{\max}\right) \cdot \frac{\left\|x_0 - x_\star\right\|^2}{\sum_{k=0}^{K-1}\alpha'_{k,G}}$$

$$= \frac{\gamma L_{\max}}{1+\gamma L_{\max}} \cdot \left(\frac{1}{\gamma} + L_{\max}\right) \cdot \frac{\left\|x_0 - x_\star\right\|^2}{\sum_{k=0}^{K-1}\alpha_{k,G}}.$$

Since

$$\frac{\gamma L_{\max}}{1+\gamma L_{\max}} \leq \frac{1+\gamma L_{\max}}{2+\gamma L_{\max}}, \quad \forall \gamma > 0,$$

the convergence of Algorithm 1 in the full participation case with (13) given in Corollary 2 is indeed better than FedExProx-GraDS. However, this adaptive rule is only practical when we have the knowledge of local smoothness.

---

[4]Note that $\frac{L_i}{1+\gamma L_i} < \frac{1}{\gamma}$ for any $\gamma > 0$.

**Notes on stochastic Polyak variant.** In this paragraph, we further elaborate on the convergence of `FedExProx-StoPS`. We start by providing a lower bound on the adaptive extrapolation parameter.

**Lemma 20.** *Suppose that all assumptions mentioned in Theorem 2 hold, then the following inequalities hold for any $x \in \mathbb{R}^d$ and $x_\star$ that is a minimizer of $f$,*

$$\frac{\frac{1}{n} \sum_{i=1}^n \left( M_{f_i}^\gamma(x) - M_{f_i}^\gamma(x_\star) \right)}{\gamma \cdot \left\| \frac{1}{n} \sum_{i=1}^n \nabla M_{f_i}^\gamma(x) \right\|^2} \geq \frac{1}{2\gamma L_\gamma}.$$

*Using the above lower bound, we can further write the convergence of `FedExProx-StoPS` as*

$$\mathbb{E}\left[ f(\bar{x}^K) \right] - \inf f \leq 2L_\gamma \left( 1 + 2\gamma L_{\max} \right) \cdot \frac{\|x_0 - x_\star\|^2}{K}.$$

Observe that we recover the favorable dependence of convergence on the smoothness of $M^\gamma$. However, this comes at the price of having to know each $M_{f_i}^\gamma(x_\star)$ or, equivalently in the interpolation regime, knowing $M^\gamma(x_\star)$.

### F.6 Extension of adaptive variants into client partial participation (PP) setting

In this subsection, we extend the adaptive variants of `FedExProx` into the stochastic setting. We will refer to them as `FedExProx-GraDS-PP` and, `FedExProx-StoPS-PP` respectively. Specifically, we consider that the server chooses the client using the $\tau$-nice sampling strategy we have introduced before in Algorithm 1. The following theorem summarizes the convergence guarantee of `FedExProx-GraDS-PP` and `FedExProx-StoPS-PP` in the convex case. Its extension to the strongly convex case where we additionally assume Assumption 5 (Strong convexity) is straight forward.

**Theorem 5.** *Suppose Assumption 1 (Differentiability), Assumption 2 (Interpolation regime), Assumption 3 (Convexity) and Assumption 4 (Smoothness) hold. Assume we are running `FedExProx` with $\tau$-nice client sampling.*

(i) *(`FedExProx-GraDS-PP`): If we are using $\alpha_k = \alpha_{\tau,k,G}(x_k, S_k)$, where*

$$\alpha_{\tau,k,G}(x_k, S_k) = \frac{\frac{1}{\tau} \sum_{i \in S_k} \left\| x_k - \text{prox}_{\gamma f_i}(x_k) \right\|^2}{\left\| \frac{1}{\tau} \sum_{i \in S_k} \left( x_k - \text{prox}_{\gamma f_i}(x_k) \right) \right\|^2}. \tag{19}$$

*Then the iterates of Algorithm 1 satisfy*

$$\mathbb{E}\left[ f(\bar{x}_K) \right] - \inf f \leq \left( \frac{1 + \gamma L_{\max}}{2 + \gamma L_{\max}} \right) \cdot \left( \frac{1}{\gamma} + L_{\max} \right) \cdot \frac{\|x_0 - x_\star\|^2}{\inf \alpha_{\tau,k,G} \cdot K}, \tag{20}$$

*where $K$ is the total number of iteration, $\bar{x}_K$ is samples uniformly at random from the first $K$ iterates $\{x_0, x_1, \ldots, x_{K-1}\}$, $\inf \alpha_{\tau,k,G}$ is defined as*

$$\inf \alpha_{\tau,k,G} := \inf_{x \in \mathbb{R}^d, S \subseteq [n], |S| = \tau} \alpha_{\tau,k,G}(x, S).$$

*satisfying*

$$\alpha_{\tau,k,G}(x_k, S_k) \geq \inf \alpha_{\tau,k,G} \geq 1.$$

(ii) *(`FedExProx-StoPS-PP`): If we are using $\alpha_k = \alpha_{\tau,k,S}(x_k, S_k)$, where*

$$\alpha_{\tau,k,S}(x_k, S_k) = \frac{\frac{1}{\tau} \sum_{i=1}^\tau \left( M_{f_i}^\gamma(x_k) - \inf M_{f_i}^\gamma \right)}{\gamma \left\| \frac{1}{\tau} \sum_{i=1}^\tau \nabla M_{f_i}^\gamma(x_k) \right\|^2}. \tag{21}$$

*Then the iterates of Algorithm 1 satisfy*

$$\mathbb{E}\left[ f(\bar{x}_K) \right] - \inf f \leq \left( \frac{1}{\gamma} + L_{\max} \right) \cdot \frac{\|x_0 - x_\star\|^2}{\inf \alpha_{\tau,k,S} \cdot K}, \tag{22}$$

*where $K$ is the total number of iteration, $\bar{x}_K$ is sampled uniformly at random from the first $K$ iterates $\{x_0, x_1, \ldots, x_{K-1}\}$, $\inf \alpha_{\tau,k,G}$ is defined as*

$$\inf \alpha_{\tau,k,S} := \inf_{x \in \mathbb{R}^d, S \subseteq [n], |S| = \tau} \alpha_{\tau,k,S}(x, S).$$

*satisfying*

$$\alpha_{\tau,k,S}(x_k, S_k) \geq \inf \alpha_{\tau,k,S} \geq \frac{1}{2}\left(1 + \frac{1}{\gamma L_{\max}}\right).$$

**Remark 15.** *For* `FedExProx-GraDS-PP`*, different from the full participation setting, the denominator of the sublinear term on the right-hand side of* (20) *is replaced by $K \cdot \inf \alpha_{\tau,k,G}$.*

(i) *In the single client case ($\tau = 1$), we have*

$$\alpha_{1,k,G} = \inf \alpha_{1,k,G} = 1.$$

(ii) *In the partial participation case ($1 < \tau < n$), it is possible that*

$$\inf \alpha_{\tau,k,G} > 1,$$

*resulting in acceleration compared to single client case.*

(iii) *For the full participation case ($\tau = n$), we have*

$$\alpha_{k,G} = \alpha_{n,k,G},$$

*and*

$$\sum_{k=0}^{K-1} \alpha_{k,G} \geq K \cdot \inf \alpha_{n,k,G},$$

*thus the convergence guarantee here is a relaxed version of that presented in Theorem* 2.

A similar discussion also applies to `FedExProx-StoPS-PP` in the client partial participation setting.

**Remark 16.** *For* `FedExProx-StoPS-PP`*, different from the full participation setting, the denominator of the sublinear term on the right-hand side of* (22) *is replaced by $K \cdot \inf \alpha_{\tau,k,S}$.*

(i) *In the single client case ($\tau = 1$), we have*

$$\alpha_{1,k,S} \geq \inf \alpha_{1,k,G} = \frac{1}{2}\left(1 + \frac{1}{\gamma L_{\max}}\right).$$

(ii) *In the partial participation case ($1 < \tau < n$), it is possible that*

$$\inf \alpha_{\tau,k,S} > \frac{1}{2}\left(1 + \frac{1}{\gamma L_{\max}}\right),$$

*resulting in acceleration compared to single client case.*

(iii) *For the full participation case ($\tau = n$), we have*

$$\alpha_{k,S} = \alpha_{n,k,S},$$

*and*

$$\sum_{k=0}^{K-1} \alpha_{k,S} \geq K \cdot \inf \alpha_{n,k,S},$$

*thus the convergence guarantee here is a relaxed version of that presented in Theorem* 2.

The following Table 5 summarizes the convergence of new algorithms and their variants appeared in our paper.

# G   Missing proofs of theorems and corollaries

## G.1   Proof of Theorem 1

The proof of this theorem can be divided into three parts.

Table 5: Summary of convergence of new algorithms appeared in our paper in the convex setting. The $\mathcal{O}(\cdot)$ notation is hidden for all complexities in this table. For convergence in the full client participation case, results of Theorem 1 and Theorem 2 are used where the relevant notations are defined. For convergence in the partial participation, the results of Theorem 5 are used.

| Method | Full Participation | Partial Participation | Single Client |
|---|---|---|---|
| FedExProx | $L_\gamma(1+\gamma L_{\max})/K$ | $L_{\gamma,\tau}(1+\gamma L_{\max})/K$ | $L_{\max}/K$ |
| FedExProx-GraDS | $(1+\gamma L_{\max})/\gamma \cdot \sum_{k=0}^{K-1}\alpha_{k,G}$ | $(1+\gamma L_{\max})/(\gamma K \cdot \inf \alpha_{\tau,k,G})$ | $(1+\gamma L_{\max})/(\gamma K)$ |
| FedExProx-StoPS | $(1+\gamma L_{\max})/\gamma \cdot \sum_{k=0}^{K-1}\alpha_{k,S}$ | $(1+\gamma L_{\max})/(\gamma K \cdot \inf \alpha_{\tau,k,S})$ | $(1+\gamma L_{\max})/(\gamma K \cdot \inf \alpha_{1,k,S})$ |

**Step 1: Reformulate the algorithm using Moreau envelope.** We know from Lemma 2 that for any $x \in \mathbb{R}^d$.

$$\nabla M_{f_i}^\gamma(x) = \frac{1}{\gamma}\left(x - \mathrm{prox}_{\gamma f_i}(x)\right).$$

Using the above identity, we can rewrite the update rule given in (7) in the following form,

$$x_{k+1} = x_k - \alpha_k \gamma \cdot \frac{1}{\tau}\sum_{i \in S_k}\nabla M_{f_i}^\gamma(x_k). \tag{23}$$

The above reformulation suggests that running FedExProx with $\tau$-nice sampling strategy is equivalent to running SGD with $\tau$-nice sampling to the global objective $M^\gamma(x) = \frac{1}{n}\sum_{i=1}^n M_{f_i}^\gamma(x)$ with step size $\alpha_k \gamma$. Now, it seems natural to apply the theory of SGD adapted in Theorem 3. However, before proceeding, we list the properties we know about the global objective $M^\gamma$ and each local objective $M_{f_i}^\gamma$.

1. Each $M_{f_i}^\gamma(x)$ is convex. This is a consequence of a direct application of Lemma 3 to each $f_i$. Since $M^\gamma$ is the average of convex functions $M_{f_i}^\gamma$, we conclude that $M^\gamma(x)$ is also convex.

2. Each $M_{f_i}^\gamma(x)$ is $\frac{L_i}{1+\gamma L_i}$-smooth, where $L_i$ is the smoothness constant of $f_i$. This is proved by applying Lemma 4 to each $f_i$. Drawing on Lemma 13 for justification, it is reasonable to assume $M^\gamma(x)$ is $L_\gamma$-smooth with $L_\gamma \leq \frac{1}{n}\sum_{i=1}^n \frac{L_i}{1+\gamma L_i}$-smooth.

3. Each $M_{f_i}^\gamma(x)$ has the same set of minimizers and minimum as $f_i$. This result arises from applying Lemma 5 to each function $f_i$.

4. Furthermore, if Assumption 2 (Interpolation Regime) holds, $M^\gamma(x)$ and $f(x)$ have the same set of minimizers and minimum. This is demonstrated in Lemma 8.

**Step 2: Applying the theory of gradient type methods.** Notice that here $M_{f_i}^\gamma(x)$ is $\frac{L_i}{1+\gamma L_i}$-smooth and convex, $M^\gamma(x)$ is convex and $L_\gamma$-smooth. Furthermore, due to the assumption of interpolation regime, $M^\gamma(x)$ and $f(x)$ have the same set of minimizers. Applying the theory of SGD with $\tau$-nice sampling in this case, where

$$A_\tau = L_{\gamma,\tau} = \frac{n-\tau}{\tau(n-1)}\cdot \max_{i\in[n]}\left(\frac{L_i}{1+\gamma L_i}\right) + \frac{n(\tau-1)}{\tau(n-1)}L_\gamma.$$

Notice that using Fact 4, we know that

$$\max_{i\in[n]}\left(\frac{L_i}{1+\gamma L_i}\right) \overset{\text{Fact 4}}{=} \frac{L_{\max}}{1+\gamma L_{\max}},$$

thus $L_\gamma$ can be simplified and written as

$$L_{\gamma,\tau} = \frac{n-\tau}{\tau(n-1)}\cdot \frac{L_{\max}}{1+\gamma L_{\max}} + \frac{n(\tau-1)}{\tau(n-1)}L_\gamma,$$

where $L_{\max} = \max_i L_i$. We obtain the following result given that $0 < \alpha\gamma < \frac{2}{L_{\gamma,\tau}}$ in the convex setting,

$$\mathbb{E}\left[M^\gamma\left(\bar{x}_K\right)\right] - M^\gamma\left(x_\star\right) \overset{\text{Theorem 3}}{\leq} \frac{1}{\alpha\gamma(2 - \alpha\gamma L_{\gamma,\tau})} \cdot \frac{\|x_0 - x_\star\|^2}{K},$$

where $\bar{x}_K$ is sampled uniformly at random from the first $K$ iterates $\{x_0, x_1, \ldots, x_{K-1}\}$. However, the convergence mentioned pertains to $M^\gamma(x)$. Given our objective is to solve (1), it is necessary to reinterpret this outcome in terms of $f$.

**Step 3: Translate the result into function values of $f$.** This step is only needed in the convex setting. We use the lower bound in Lemma 10,

$$M^\gamma\left(\bar{x}_K\right) - M^\gamma\left(x_\star\right) \overset{(18)}{\geq} \frac{1}{1 + \gamma L_{\max}}\left(f(\bar{x}_K) - f(x_\star)\right),$$

to obtain the following result

$$\mathbb{E}\left[f(\bar{x}_K)\right] - f(x_\star) \leq \frac{1 + \gamma L_{\max}}{\alpha\gamma\left(2 - \alpha\gamma L_{\gamma,\tau}\right)} \cdot \frac{\|x_0 - x_\star\|^2}{K}.$$

Observe that we have

$$C\left(\gamma, \tau, \alpha\right) = \frac{1 + \gamma L_{\max}}{\alpha\gamma\left(2 - \alpha\gamma L_{\gamma,\tau}\right)},$$

and its numerator does not depend on $\alpha$. If we fix the choice of $\gamma$ and $\tau$, then the denominator is maximized when $\alpha\gamma L_{\gamma,\tau} = 1$. This yields the optimal constant extrapolation parameter $\alpha_{\gamma,\tau} = \frac{1}{\gamma L_{\gamma,\tau}}$ and the following convergence corresponding to it

$$\mathbb{E}\left[f(\bar{x}_K)\right] - f(x_\star) \leq L_{\gamma,\tau}\left(1 + \gamma L_{\max}\right) \cdot \frac{\|x_0 - x_\star\|^2}{K}.$$

Finally, notice that

$$\gamma L_\gamma \overset{\text{Lemma 13}}{\leq} \frac{1}{n}\sum_{i=1}^n \frac{\gamma L_i}{1 + \gamma L_i} < 1,$$

for any $\gamma > 0$. This suggests that,

$$\gamma L_{\gamma,\tau} = \frac{n - \tau}{\tau(n-1)} \cdot \frac{\gamma L_{\max}}{1 + \gamma L_{\max}} + \frac{n(\tau - 1)}{\tau(n-1)}\gamma L_\gamma$$

$$< \frac{n - \tau}{\tau(n-1)} + \frac{n(\tau - 1)}{\tau(n-1)} = 1,$$

which in turn tells us $\alpha_{\gamma,\tau} = \frac{1}{\gamma L_{\gamma,\tau}} > 1$. This concludes the proof.

### G.2 Proof of Theorem 2

We start with the following decomposition,

$$\|x_{k+1} - x_\star\|^2 = \|x_k - \alpha_k\gamma\nabla M^\gamma\left(x_k\right) - x_\star\|^2$$

$$= \|x_k - x_\star\|^2 - 2\alpha_k\gamma\left\langle\nabla M^\gamma\left(x_k\right), x_k - x_\star\right\rangle + \alpha_k^2\gamma^2\|\nabla M^\gamma\left(x\right)\|^2. \tag{24}$$

**Case 1: FedExProx-GraDS** For gradient diversity based $\alpha_k$, we have

$$\alpha_k = \alpha_{k,G} = \frac{\frac{1}{n}\sum_{i=1}^n\left\|\gamma\nabla M_{f_i}^\gamma\left(x_k\right)\right\|^2}{\|\gamma\nabla M^\gamma\left(x_k\right)\|^2} = \frac{\frac{1}{n}\sum_{i=1}^n\left\|\nabla M_{f_i}^\gamma\left(x_k\right)\right\|^2}{\|\nabla M^\gamma\left(x_k\right)\|^2}.$$

For the last term of (24),

$$\alpha_{k,G}^2\gamma^2\|\nabla M^\gamma\left(x_k\right)\|^2 = \alpha_{k,G}\gamma^2 \cdot \frac{1}{n}\sum_{i=1}^n\left\|\nabla M_{f_i}^\gamma\left(x_k\right)\right\|^2$$

$$= \alpha_{k,G}\gamma^2 \cdot \frac{1}{n}\sum_{i=1}^n\left\|\nabla M_{f_i}^\gamma\left(x_k\right) - \nabla M_{f_i}^\gamma\left(x_\star\right)\right\|^2$$

$$\overset{(16)}{\leq} \alpha_{k,G}\gamma^2 \cdot \frac{1}{n}\sum_{i=1}^n \frac{L_i}{1 + \gamma L_i}\left(D_{M_{f_i}^\gamma}\left(x_k, x_\star\right) + D_{M_{f_i}^\gamma}\left(x_\star, x_k\right)\right),$$

where the last inequality follows from the $\frac{L_i}{1+\gamma L_i}$-smoothness of $M_{f_i}^\gamma$ given in Lemma 4. We further obtain using Fact 4 that

$$\alpha_{k,G}^2\gamma^2\left\|\nabla M^\gamma\left(x_k\right)\right\|^2 \overset{\text{Fact 4}}{\leq} \alpha_{k,G}\gamma^2\cdot\frac{L_{\max}}{1+\gamma L_{\max}}\cdot\left(D_{M^\gamma}\left(x_k,x_\star\right)+D_{M^\gamma}\left(x_\star,x_k\right)\right)$$

$$= \alpha_{k,G}\gamma\cdot\frac{\gamma L_{\max}}{1+\gamma L_{\max}}\left(D_{M^\gamma}\left(x_k,x_\star\right)+D_{M^\gamma}\left(x_\star,x_k\right)\right). \tag{25}$$

For the second term of (24), we have

$$-2\alpha_{k,G}\gamma\left\langle\nabla M^\gamma\left(x_k\right),x_k-x_\star\right\rangle = 2\alpha_{k,G}\gamma\left\langle\nabla M^\gamma\left(x_k\right)-\nabla M^\gamma\left(x_\star\right),x_\star-x_k\right\rangle$$

$$= -2\alpha_{k,G}\gamma\left(D_{M^\gamma}\left(x_k,x_\star\right)+D_{M^\gamma}\left(x_\star,x_k\right)\right). \tag{26}$$

Plugging (26) and (25) into (24), we have

$$\left\|x_{k+1}-x_\star\right\|^2 \leq \left\|x_k-x_\star\right\|^2 - \alpha_{k,G}\gamma\left(2-\frac{\gamma L_{\max}}{1+\gamma L_{\max}}\right)\left(D_{M^\gamma}\left(x_k,x_\star\right)+D_{M^\gamma}\left(x_\star,x_k\right)\right).$$

Notice that we know that

$$D_{M^\gamma}\left(x_k,x_\star\right)\overset{(14)}{=}M^\gamma\left(x_k\right)-M^\gamma\left(x_\star\right),\quad D_{M^\gamma}\left(x_\star,x_k\right)\overset{(15)}{\geq}0.$$

As a result, we obtain

$$\left\|x_{k+1}-x_\star\right\|^2 \leq \left\|x_k-x_\star\right\|^2 - \alpha_{k,G}\gamma\left(2-\frac{\gamma L_{\max}}{1+\gamma L_{\max}}\right)\left(M^\gamma\left(x_k\right)-M^\gamma\left(x_\star\right)\right).$$

Summing up the above recursion for $k=0,1,...,K-1$, we notice that many of them will telescope and $M^\gamma\left(x_\star\right)=\inf M^\gamma$ due to interpolation regime as it is proved by Lemma 8. Thus, we obtain

$$\gamma\left(2-\frac{\gamma L_{\max}}{1+\gamma L_{\max}}\right)\sum_{k=0}^{K-1}\alpha_{k,G}\left(M^\gamma\left(x_k\right)-\inf M^\gamma\right)\leq\left\|x_0-x_\star\right\|^2.$$

Denote $p_k = \alpha_{k,G}/\sum_{k=0}^{K-1}\alpha_{k,G}$ for $k=0,1,...,K-1$. If we pick $\bar{x}_K$ randomly according to probabilities $p_k$ from the first $K$ iterates $\{x_0,x_1,\ldots,x_{K-1}\}$, then we can further write the above recursion as

$$\mathbb{E}\left[M^\gamma\left(\bar{x}^K\right)\right]-\inf M^\gamma\leq\frac{1+\gamma L_{\max}}{2+\gamma L_{\max}}\cdot\frac{1}{\gamma}\cdot\frac{\left\|x_0-x_\star\right\|^2}{\sum_{k=0}^{K-1}\alpha_{k,G}}.$$

Utilizing Lemma 10, we further obtain,

$$\mathbb{E}\left[f(\bar{x}_K)\right]-\inf f\leq\frac{1+\gamma L_{\max}}{2+\gamma L_{\max}}\cdot\left(\frac{1}{\gamma}+L_{\max}\right)\cdot\frac{\left\|x_0-x_\star\right\|^2}{\sum_{k=0}^{K-1}\alpha_{k,G}}.$$

The above inequality indicates convergence. Indeed, by convexity of standard Euclidean norm, we have

$$\alpha_{k,G}\geq\frac{\left\|\frac{1}{n}\sum_{i=1}^n\left(x_k-\text{prox}_{\gamma f_i}\left(x_k\right)\right)\right\|^2}{\left\|\frac{1}{n}\sum_{i=1}^n\left(x_k-\text{prox}_{\gamma f_i}\left(x_k\right)\right)\right\|^2}=1.$$

This tells us that

$$\sum_{k=0}^{K-1}\alpha_{k,G}\geq K.$$

**Case 2: FedExProx-StoPS** For stochastic Polyak step size based $\alpha_{k,S}$, since we are in the interpolation regime, by Lemma 9, we have

$$M^\gamma\left(x_\star\right)=\inf M^\gamma=\frac{1}{n}\sum_{i=1}^n\inf M_{f_i}^\gamma.$$

As a result,

$$\alpha_k=\alpha_{k,S}=\frac{\frac{1}{n}\sum_{i=1}^n\left(M_{f_i}^\gamma\left(x_k\right)-\inf M_{f_i}^\gamma\right)}{\gamma\left\|\frac{1}{n}\sum_{i=1}^n\nabla M_{f_i}^\gamma\left(x_k\right)\right\|^2}=\frac{M^\gamma\left(x_k\right)-M^\gamma\left(x_\star\right)}{\gamma\left\|\nabla M^\gamma\left(x_k\right)\right\|^2}.$$

We have for the last term of (24),

$$\alpha_{k,S}^2 \gamma^2 \|\nabla M^\gamma (x_k)\|^2 = \alpha_{k,S}\gamma \left(M^\gamma (x_k) - M^\gamma (x_\star)\right). \tag{27}$$

For the second term of (24), we have

$$
\begin{aligned}
-2\alpha_{k,S}\gamma \left\langle \nabla M^\gamma (x_k), x_k - x_\star \right\rangle &= 2\alpha_{k,S}\gamma \left\langle \nabla M^\gamma (x_k), x_\star - x_k \right\rangle \\
&\overset{(5)}{\leq} 2\alpha_{k,S}\gamma \left(M^\gamma (x_\star) - M^\gamma (x_k)\right) \\
&= -2\alpha_{k,S}\gamma \left(M^\gamma (x_k) - M^\gamma (x_\star)\right),
\end{aligned} \tag{28}
$$

where the inequality is due to the convexity of $M^\gamma$. Plugging (28) and (27) into (24), we obtain

$$\|x_{k+1} - x_\star\|^2 \leq \|x_k - x_\star\|^2 - \alpha_{k,S}\gamma \left(M^\gamma (x_k) - M^\gamma (x_\star)\right).$$

Summing up the above recursion for $k = 0, 1, ..., K - 1$, we notice that many of them will telescope. Thus, we obtain

$$\gamma \sum_{k=0}^{K-1} \alpha_{k,S} \left(M^\gamma (x_k) - \inf M^\gamma\right) \leq \|x_0 - x_\star\|^2.$$

Denote $p_k = \alpha_{k,S}/\sum_{k=0}^{K-1} \alpha_{k,S}$ for $k = 0, 1, ..., K - 1$. If we sample $\bar{x}^K$ randomly according to probabilities $p_k$ from the first $K$ iterates $\{x_0, x_1, \ldots, x_{K-1}\}$, we can further write the above recursion as

$$\mathbb{E}\left[M^\gamma \left(\bar{x}^K\right)\right] - \inf M^\gamma \leq \frac{1}{\gamma} \cdot \frac{\|x_0 - x_\star\|^2}{\sum_{k=0}^{K-1} \alpha_{k,S}}.$$

Utilizing the local bound in Lemma 10, we further obtain,

$$\mathbb{E}\left[f(\bar{x}^K)\right] - \inf f \overset{(17)}{\leq} \left(\frac{1}{\gamma} + L_{\max}\right) \cdot \frac{\|x_0 - x_\star\|^2}{\sum_{k=0}^{K-1} \alpha_{k,S}}. \tag{29}$$

Notice that the above inequality indeed indicates convergence, since

$$\sum_{k=0}^{K-1} \alpha_{k,S} = \sum_{k=0}^{K-1} \frac{M^\gamma (x_k) - M^\gamma (x_\star)}{\gamma \|\nabla M^\gamma (x_k)\|^2} \geq \frac{1}{2\gamma L_\gamma},$$

where the inequality follows from Lemma 20. The above upper bounds allow us to further write the convergence in (29) as

$$\mathbb{E}\left[f(\bar{x}^K)\right] - \inf f \leq 2L_\gamma \left(1 + 2\gamma L_{\max}\right) \cdot \frac{\|x_0 - x_\star\|^2}{K}.$$

This concludes the proof.

### G.3 Proof of Theorem 3

We start from the decomposition,

$$\|x_{k+1} - x_\star\|^2 = \|x_k - x_\star\|^2 - 2\eta \left\langle x_k - x_\star, \frac{1}{\tau} \sum_{i \in S_k} \nabla f_i(x_k) \right\rangle + \eta^2 \left\| \frac{1}{\tau} \sum_{i \in S_k} \nabla f_i(x_k) \right\|^2,$$

where $S_k$ is the set sampled at iteration $k$. Taking expectation conditioned on $x_k$, we have

$$
\mathbb{E}_{S_k}\left[\|x_{k+1} - x_\star\|^2\right]
$$

$$
= \|x_k - x_\star\|^2 - 2\eta \left\langle x_k - x_\star, \nabla f(x_k) - \nabla f(x_\star) \right\rangle + \eta^2 \mathbb{E}_{S_k}\left[\left\| \frac{1}{\tau} \sum_{i \in S_k} \nabla f_i(x_k) \right\|^2\right].
$$

We can write the second inner product term as

$$\left\langle x_k - x_\star, \nabla f(x_k) - \nabla f(x_\star) \right\rangle \overset{(14)}{=} D_f (x_k, x_\star) + D_f (x_\star, x_k), \tag{30}$$

where $D_f(x_k, x_\star)$ denotes the Bregman divergence associated with $f$ between $x_k$ and $x_\star$. For the last squared norm term, we first define the indicator random variable $\chi_{k,i}$ as

$$\chi_{k,i} = \begin{cases} 1, & \text{when } i \in S_k, \\ 0, & \text{when } i \notin S_k. \end{cases}$$

Since we are in the interpolation regime, we have

$$\mathbb{E}_{S_k}\left[\left\|\frac{1}{\tau}\sum_{i \in S_k} \nabla f_i(x_k)\right\|^2\right] = \mathbb{E}_{S_k}\left[\left\|\frac{1}{\tau}\sum_{i=1}^{n} \chi_{k,i}\left(\nabla f_i(x_k) - \nabla f_i(x_\star)\right)\right\|^2\right].$$

Denote $a_{k,i} = \nabla f_i(x_k) - \nabla f_i(x_\star)$,

$$\mathbb{E}_{S_k}\left[\left\|\frac{1}{\tau}\sum_{i=1}^{n} \chi_{k,i}\left(\nabla f_i(x_k) - \nabla f_i(x_\star)\right)\right\|^2\right]$$

$$= \mathbb{E}_{S_k}\left[\left\|\frac{1}{\tau}\sum_{i=1}^{n} \chi_{k,i} a_{k,i}\right\|^2\right]$$

$$= \frac{1}{\tau^2}\mathbb{E}_{S_k}\left[\sum_{i=1}^{n} \chi_{k,i}^2 \|a_{k,i}\|^2 + \sum_{1 \le i \ne j \le n} \chi_{k,i}\chi_{j,k}\langle a_{k,i}, a_{k,j}\rangle\right]$$

$$= \frac{1}{\tau^2}\sum_{i=1}^{n} \mathbb{E}_{S_i^k}\left[\chi_{k,i}^2\right]\|a_{k,i}\|^2 + \sum_{1 \le i \ne j \le n} \mathbb{E}_{S_i^k}\left[\chi_{k,i}\chi_{j,k}\right]\langle a_{k,i}, a_{k,j}\rangle$$

$$= \frac{1}{n\tau}\sum_{i=1}^{n} \|a_{k,i}\|^2 + \frac{\tau-1}{n\tau(n-1)}\left(\left\|\sum_{i=1}^{n} a_{k,i}\right\|^2 - \sum_{i=1}^{n} \|a_{k,i}\|^2\right)$$

$$= \frac{n-\tau}{\tau(n-1)} \cdot \frac{1}{n}\sum_{i=1}^{n} \|a_{k,i}\|^2 + \frac{n(\tau-1)}{\tau(n-1)} \cdot \left\|\frac{1}{n}\sum_{i=1}^{n} a_{k,i}\right\|^2. \tag{31}$$

For the first term above in (31), due to the smoothness and convexity of each $f_i$, we have

$$\frac{1}{n}\sum_{i=1}^{n} \|a_{k,i}\|^2 = \frac{1}{n}\sum_{i=1}^{n} \|\nabla f_i(x_k) - \nabla f_i(x_\star)\|^2$$

$$\le \frac{1}{n}\sum_{i=1}^{n} L_i\left(D_{f_i}(x_\star, x_k) + D_{f_i}(x_k, x_\star)\right)$$

$$\le L_{\max}\frac{1}{n}\sum_{i=1}^{n}\left(D_{f_i}(x_\star, x_k) + D_{f_i}(x_k, x_\star)\right)$$

$$= L_{\max}\left(D_f(x_\star, x_k) + D_f(x_k, x_\star)\right),$$

where the first inequality is obtained as a result of Fact 3. For the second term, we have due to the smoothness and convexity of $f$,

$$\left\|\frac{1}{n}\sum_{i=1}^{n} a_{k,i}\right\|^2 = \left\|\frac{1}{n}\sum_{i=1}^{n}\left(\nabla f_i(x_k) - \nabla f_i(x_\star)\right)\right\|^2$$

$$= \|\nabla f(x_k) - \nabla f(x_\star)\|^2$$

$$\le L\left(D_f(x_\star, x_k) + D_f(x_k, x_\star)\right),$$

where the inequality is obtained using Fact 3. Combining the above two inequalities and plugging them into (31), we obtain

$$\mathbb{E}_{S_k}\left[\left\|\frac{1}{\tau}\sum_{i \in S_k} \nabla f_i(x_k)\right\|^2\right] \le \left(\frac{n-\tau}{\tau(n-1)} \cdot L_{\max} + \frac{n(\tau-1)}{\tau(n-1)} \cdot L\right)\left(D_f(x_\star, x_k) + D_f(x_k, x_\star)\right).$$

$$\tag{32}$$

Notice that we already defined $A_\tau$ as

$$A_\tau = \frac{n - \tau}{\tau(n-1)} \cdot L_{\max} + \frac{n(\tau - 1)}{\tau(n-1)} \cdot L.$$

Combining (30) and (32), we have

$$\mathbb{E}_{S_k}\left[\|x_{k+1} - x_\star\|^2\right] \leq \|x_k - x_\star\|^2 - \eta(2 - \eta A_\tau)\left(D_f\left(x_\star, x_k\right) + D_f\left(x_k, x_\star\right)\right).$$

If we require $0 < \eta < \frac{2}{A_\tau}$, we have $\eta(2 - \eta A_\tau) \geq 0$.

**Convex regime.** It remains to notice that $D_f\left(x_k, x_\star\right) + D_f\left(x_\star, x_k\right) \geq D_f\left(x_k, x_\star\right) = f(x_k) - f(x_\star) \geq 0$, and we have

$$\mathbb{E}_{S_k}\left[\|x_{k+1} - x_\star\|^2\right] \leq \|x_k - x_\star\|^2 - \eta(2 - \eta A_\tau)\left(f(x_k) - f(x_\star)\right).$$

Taking expectation again and using tower property, we get

$$\mathbb{E}\left[\|x_{k+1} - x_\star\|^2\right] \leq \mathbb{E}\left[\|x_k - x_\star\|^2\right] - \eta(2 - \eta A_\tau)\left(\mathbb{E}\left[f(x_k)\right] - \inf f\right).$$

Unrolling this recurrence, we get

$$\mathbb{E}\left[f(\bar{x}_K)\right] - \inf f \leq \frac{1}{\eta(2 - \eta A_\tau)} \cdot \frac{\|x_0 - x_\star\|^2}{K},$$

where $K$ is the total number of iterations, $\bar{x}_K$ is selected uniformly at random from the first $K$ iterates $\{x_0, x_1, \ldots, x_{K-1}\}$.

**Star strongly convex regime.** Due to star strong convexity of $f$, we further lower bound the Bregman divergence

$$D_f\left(x_k, x_\star\right) = f(x_k) - f(x_\star) \geq \frac{\mu}{2}\|x_k - x_\star\|^2.$$

and we have

$$\mathbb{E}_{S_k}\left[\|x_{k+1} - x_\star\|^2\right] \leq \left(1 - \eta(2 - \eta A_\tau) \cdot \frac{\mu}{2}\right)\|x_k - x_\star\|^2.$$

Taking expectation again, using tower property we get

$$\mathbb{E}\left[\|x_{k+1} - x_\star\|^2\right] \leq \left(1 - \eta(2 - \eta A_\tau) \cdot \frac{\mu}{2}\right)\mathbb{E}\left[\|x_k - x_\star\|^2\right].$$

Unrolling the recurrence, we get

$$\mathbb{E}\left[\|x_K - x_\star\|^2\right] \leq \left(1 - \eta(2 - \eta A_\tau) \cdot \frac{\mu}{2}\right)^K\|x_0 - x_\star\|^2.$$

This concludes the proof.

### G.4 Proof of Theorem 4

Since each $f_i$ is proper, closed and convex, by Lemma 2, we know that each $M_{f_i}^\gamma$ is $\frac{1}{\gamma}$-smooth. Therefore, it is reasonable to assume that $M^\gamma = \frac{1}{n}\sum_{i=1}^n M_{f_i}^\gamma$ is $L_\gamma$-smooth, with $L_\gamma \leq \frac{1}{\gamma}$. Applying Theorem 3 in this case, we obtain,

$$M^\gamma\left(\bar{x}_K\right) - \inf M^\gamma \overset{\text{Theorem 3}}{\leq} \frac{1}{\alpha\gamma\left(2 - \alpha\gamma L_{\gamma,\tau}\right)} \cdot \frac{\|x_0 - x_\star\|^2}{K},$$

where $\bar{x}_K$ is chosen uniformly at random from the first $K$ iterates $\{x_0, x_1, \ldots, x_{K-1}\}$, and

$$L_{\gamma,\tau} = \frac{n - \tau}{\tau(n-1)} \cdot \frac{1}{\gamma} + \frac{n(\tau - 1)}{\tau(n-1)} \cdot L_\gamma.$$

Multiplying both sides by $\gamma$, we obtain

$$\gamma M^\gamma(\bar{x}_K) - \inf \gamma M^\gamma \le \frac{1}{\alpha(2 - \alpha\gamma L_{\gamma,\tau})} \cdot \frac{\|x_0 - x_\star\|^2}{K}.$$

It is easy to see that the coefficient on the right-hand side is minimized when $\alpha = \frac{1}{\gamma L_{\gamma,\tau}}$, and the convergence is given by

$$\gamma M^\gamma(\bar{x}_K) - \inf \gamma M^\gamma \le \left( \frac{n - \tau}{\tau(n-1)} + \frac{n(\tau-1)}{\tau(n-1)} \cdot \gamma L_\gamma \right) \cdot \frac{\|x_0 - x_\star\|^2}{K}.$$

Notice that $L_\gamma \le \frac{1}{\gamma}$. As a result,

$$\alpha_\star = \frac{1}{\gamma L_\gamma} \ge 1.$$

### G.5    Proof of Theorem 5

**Case of `FedExProx-GraDS-PP`.**    We start with the following identity

$$\|x_{k+1} - x_\star\|^2 = \|x_k - x_\star\|^2 - 2\alpha_{\tau,k,G} \cdot \gamma \left\langle \frac{1}{\tau} \sum_{i \in S_k} \nabla M_{f_i}^\gamma(x_k), x_k - x_\star \right\rangle$$

$$+ \alpha_{\tau,k,G}^2 \cdot \gamma^2 \cdot \left\| \frac{1}{\tau} \sum_{i \in S_k} \nabla M_{f_i}^\gamma(x_k) \right\|^2. \tag{33}$$

For the last term, we have

$$\alpha_{\tau,k,G}^2 \cdot \gamma^2 \cdot \left\| \frac{1}{\tau} \sum_{i \in S_k} \nabla M_{f_i}^\gamma(x_k) \right\|^2 = \alpha_{\tau,k,G} \cdot \gamma^2 \cdot \frac{1}{\tau} \sum_{i \in S_k} \left\| \nabla M_{f_i}^\gamma(x_k) \right\|^2$$

$$= \alpha_{\tau,k,G} \cdot \gamma^2 \cdot \frac{1}{\tau} \sum_{i \in S_k} \left\| \nabla M_{f_i}^\gamma(x_k) - \nabla M_{f_i}^\gamma(x_\star) \right\|^2,$$

where the last step is due to the assumption that we are in the interpolation regime. Using Fact 3, we can further upper bound the above expression,

$$\alpha_{\tau,k,G}^2 \cdot \gamma^2 \cdot \left\| \frac{1}{\tau} \sum_{i \in S_k} \nabla M_{f_i}^\gamma(x_k) \right\|^2$$

$$\le \alpha_{\tau,k,G} \cdot \gamma^2 \cdot \frac{1}{\tau} \sum_{i \in S_k} \frac{L_i}{1 + \gamma L_i} \left( D_{M_{f_i}^\gamma}(x_k, x_\star) + D_{M_{f_i}^\gamma}(x_\star, x_k) \right)$$

$$\le \alpha_{\tau,k,G} \cdot \gamma \cdot \frac{\gamma L_{\max}}{1 + \gamma L_{\max}} \cdot \frac{1}{\tau} \sum_{i \in S_k} \left( D_{M_{f_i}^\gamma}(x_k, x_\star) + D_{M_{f_i}^\gamma}(x_\star, x_k) \right), \tag{34}$$

where the last inequality is due to Fact 4. Now we look at the second term in Equation (33).

$$-2\alpha_{\tau,k,G} \cdot \gamma \left\langle \frac{1}{\tau} \sum_{i \in S_k} \nabla M_{f_i}^\gamma(x_k), x_k - x_\star \right\rangle$$

$$= -2\alpha_{\tau,k,G} \cdot \gamma \left\langle \frac{1}{\tau} \sum_{i \in S_k} \left( \nabla M_{f_i}^\gamma(x_k) - M_{f_i}^\gamma(x_\star) \right), x_k - x_\star \right\rangle$$

$$= -2\alpha_{\tau,k,G} \cdot \gamma \cdot \frac{1}{\tau} \sum_{i \in S_k} \left( D_{M_{f_i}^\gamma}(x_k, x_\star) + D_{M_{f_i}^\gamma}(x_\star, x_k) \right). \tag{35}$$

Plugging (34) and (35) into (33), we obtain,

$$\|x_{k+1} - x_\star\|^2$$

$$\leq \|x_k - x_\star\|^2 - \alpha_{\tau,k,G} \cdot \gamma \left(2 - \frac{\gamma L_{\max}}{1 + \gamma L_{\max}}\right) \cdot \frac{1}{\tau} \sum_{i \in S_k} \left(D_{M_{f_i}^\gamma}(x_k, x_\star) + D_{M_{f_i}^\gamma}(x_\star, x_k)\right)$$

$$\leq \|x_k - x_\star\|^2 - \alpha_{\tau,k,G} \cdot \gamma \left(\frac{2 + \gamma L_{\max}}{1 + \gamma L_{\max}}\right) \cdot \frac{1}{\tau} \sum_{i \in S_k} \left(M_{f_i}^\gamma(x_k) - M_{f_i}^\gamma(x_\star)\right), \tag{36}$$

where the last inequality is due to

$$D_{M_{f_i}^\gamma}(x_k, x_\star) \stackrel{(14)}{=} M_{f_i}^\gamma(x_k) - M_{f_i}^\gamma(x_\star), \quad \text{and} \quad D_{M_{f_i}^\gamma}(x_\star, x_k) \stackrel{(15)}{\geq} 0.$$

Now we want to lower bound $\alpha_{\tau,k,G}$, notice that it can be viewed as a function of the iterate $x$ and the sampled set $S$. Therefore, we use the notation

$$\inf \alpha_{\tau,k,G} = \inf_{x \in \mathbb{R}^d, S \subseteq [n], |S| = \tau} \alpha_{\tau,k,G}(x, S).$$

As a result, we have

$$\alpha_{\tau,k,G} \geq \inf \alpha_{\tau,k,G} \geq 1,$$

where the second inequality comes from the convexity of standard Euclidean norm. Plugging this lower bound into (36), we obtain

$$\|x_{k+1} - x_\star\|^2$$

$$\leq \|x_k - x_\star\|^2 - \inf \alpha_{\tau,k,G} \cdot \gamma \left(\frac{2 + \gamma L_{\max}}{1 + \gamma L_{\max}}\right) \cdot \frac{1}{\tau} \sum_{i \in S_k} \left(M_{f_i}^\gamma(x_k) - M_{f_i}^\gamma(x_\star)\right).$$

Taking expectation conditioned on $x_k$, we have

$$\mathbb{E}_{S_k}\left[\|x_{k+1} - x_\star\|^2\right]$$

$$\leq \|x_k - x_\star\|^2 - \inf \alpha_{\tau,k,G} \cdot \gamma \left(\frac{2 + \gamma L_{\max}}{1 + \gamma L_{\max}}\right) \cdot \frac{1}{n} \sum_{i=1}^n \left(M_{f_i}^\gamma(x_k) - M_{f_i}^\gamma(x_\star)\right)$$

$$= \|x_k - x_\star\|^2 - \inf \alpha_{\tau,k,G} \cdot \gamma \left(\frac{2 + \gamma L_{\max}}{1 + \gamma L_{\max}}\right) \cdot (M^\gamma(x_k) - \inf M),$$

where the last identity is due to the fact that we are in the interpolation regime. Using Lemma 10, we have

$$\mathbb{E}_{S_k}\left[\|x_{k+1} - x_\star\|^2\right]$$

$$\leq \|x_k - x_\star\|^2 - \inf \alpha_{\tau,k,G} \cdot \gamma \left(\frac{2 + \gamma L_{\max}}{1 + \gamma L_{\max}}\right) \cdot \frac{1}{1 + \gamma L_{\max}} (f(x_k) - \inf f).$$

Taking expectation again and using tower property, we obtain

$$\mathbb{E}\left[\|x_{k+1} - x_\star\|^2\right]$$

$$\leq \mathbb{E}\left[\|x_k - x_\star\|^2\right] - \inf \alpha_{\tau,k,G} \cdot \gamma \left(\frac{2 + \gamma L_{\max}}{1 + \gamma L_{\max}}\right) \cdot \frac{1}{1 + \gamma L_{\max}} \mathbb{E}\left[f(x_k) - \inf f\right].$$

Following the same step as Theorem 1, we can unroll the above recurrence and obtain

$$\mathbb{E}\left[f(\bar{x}_K)\right] - \inf f \leq \left(\frac{1 + \gamma L_{\max}}{2 + \gamma L_{\max}}\right) \cdot \left(\frac{1}{\gamma} + L_{\max}\right) \cdot \frac{\|x_0 - x_\star\|^2}{\inf \alpha_{\tau,k,G} \cdot K},$$

where $K$ is the total number of iterations, $\bar{x}_K$ is sampled uniformly at random from the first $K$-iterates $\{x_0, x_1, \ldots, x_{K-1}\}$.

**Case of `FedExProx-StoPS-PP`.** We start with the following identity

$$\|x_{k+1} - x_\star\|^2 = \|x_k - x_\star\|^2 - 2\alpha_{\tau,k,S} \cdot \gamma \left\langle \frac{1}{\tau} \sum_{i \in S_k} \nabla M_{f_i}^\gamma(x_k), x_k - x_\star \right\rangle$$

$$+ \alpha_{\tau,k,S}^2 \cdot \gamma^2 \cdot \left\| \frac{1}{\tau} \sum_{i \in S_k} \nabla M_{f_i}^\gamma(x_k) \right\|^2. \tag{37}$$

For the last term of Equation (37), we have

$$\alpha_{\tau,k,S}^2 \cdot \gamma^2 \cdot \left\| \frac{1}{\tau} \sum_{i \in S_k} \nabla M_{f_i}^\gamma(x_k) \right\|^2 = \alpha_{\tau,k,S} \cdot \gamma \cdot \frac{1}{\tau} \sum_{i \in S_k} \left( M_{f_i}^\gamma(x_k) - \inf M_{f_i}^\gamma \right)$$

$$= \alpha_{\tau,k,S} \cdot \gamma \cdot \frac{1}{\tau} \sum_{i \in S_k} \left( D_{M_{f_i}^\gamma}(x_k, x_\star) \right). \tag{38}$$

While for the second term we have

$$- 2\alpha_{\tau,k,S} \cdot \gamma \left\langle \frac{1}{\tau} \sum_{i \in S_k} \nabla M_{f_i}^\gamma(x_k), x_k - x_\star \right\rangle$$

$$= -2\alpha_{\tau,k,S} \cdot \gamma \cdot \frac{1}{\tau} \sum_{i \in S_k} \left( D_{M_{f_i}^\gamma}(x_k, x_\star) + D_{M_{f_i}^\gamma}(x_\star, x_k) \right)$$

$$\overset{(15)}{\leq} -2\alpha_{\tau,k,S} \cdot \gamma \cdot \frac{1}{\tau} \sum_{i \in S_k} D_{M_{f_i}^\gamma}(x_k, x_\star). \tag{39}$$

Plugging (38) and (39) into (37), we obtain

$$\|x_{k+1} - x_\star\|^2 \leq \|x_k - x_\star\|^2 - \alpha_{\tau,k,S} \cdot \gamma \cdot \frac{1}{\tau} \sum_{i \in S_k} \left( M_{f_i}^\gamma(x_k) - \inf M_{f_i}^\gamma \right). \tag{40}$$

Now we want to lower bound $\alpha_{\tau,k,S}$, notice that it can be viewed as a function of the iterate $x$ and the sampled set $S$. Therefore, we use the notation

$$\inf \alpha_{\tau,k,S} = \inf_{x \in \mathbb{R}^d, S \subseteq [n], |S| = \tau} \alpha_{\tau,k,S}(x, S).$$

As a result, we have

$$\alpha_{\tau,k,S} \geq \inf \alpha_{\tau,k,S}.$$

Notice that since each $M_{f_i}^\gamma$ is $\frac{L_i}{1+\gamma L_i}$-smooth, we conclude that the function $\frac{1}{\tau} \sum_{i \in S_k} M_{f_i}^\gamma$ is at least $\frac{L_{\max}}{1+\gamma L_{\max}}$-smooth[5]. Using the smoothness of the mentioned function and Fact 3, a lower bound on $\inf \alpha_{\tau,k,S}$ is obvious,

$$\inf \alpha_{k,\tau,S} \geq \frac{1}{2 \cdot \frac{L_{\max}}{1+\gamma L_{\max}} \gamma} = \frac{1}{2} \left( 1 + \frac{1}{\gamma L_{\max}} \right).$$

This means that we have

$$\alpha_{\tau,k,S} \geq \inf \alpha_{\tau,k,S} \geq \frac{1}{2} \left( 1 + \frac{1}{\gamma L_{\max}} \right).$$

Using the above lower bound in (40), we have

$$\|x_{k+1} - x_\star\|^2 \leq \|x_k - x_\star\|^2 - \inf \alpha_{\tau,k,S} \cdot \gamma \cdot \frac{1}{\tau} \sum_{i \in S_k} \left( M_{f_i}^\gamma(x_k) - \inf M_{f_i}^\gamma \right).$$

Taking expectation conditioned on $x_k$, and noticing that we are in the interpolation regime, we obtain

$$\mathbb{E}_{S_k} \left[ \|x_{k+1} - x_\star\|^2 \right] \leq \|x_k - x_\star\|^2 - \inf \alpha_{\tau,k,S} \cdot \gamma \cdot (M^\gamma(x_k) - \inf M).$$

---

[5]Same as $M^\gamma(x)$, its smoothness constant can be much better.

Using Lemma 10, we have

$$\mathbb{E}_{S_k}\left[\|x_{k+1} - x_\star\|^2\right] \overset{\text{Lemma 10}}{\leq} \|x_k - x_\star\|^2 - \inf \alpha_{\tau,k,S} \cdot \frac{\gamma}{1 + \gamma L_{\max}} \cdot (f(x_k) - \inf f).$$

Now, following the exact same steps as in the previous case of `FedExProx-GraDS`, we result in

$$\mathbb{E}\left[f(\bar{x}_K)\right] - \inf f \leq \left(\frac{1}{\gamma} + L_{\max}\right) \cdot \frac{\|x_0 - x_\star\|^2}{\inf \alpha_{\tau,k,S} \cdot K},$$

where $K$ is the total number of iterations, $\bar{x}_K$ is sampled uniformly at random from the first $K$-iterates $\{x_0, x_1, \ldots, x_{K-1}\}$.

### G.6 Proof of Corollary 1

If additionally we assume $f$ is $\mu$-strongly convex, then from Lemma 11, we know it indicates the following star strong convexity of $M^\gamma$ holds,

$$M^\gamma(x) - M^\gamma(x_\star) \geq \frac{\mu}{1 + \gamma L_{\max}} \cdot \frac{1}{2}\|x - x_\star\|^2.$$

Thus, we apply Theorem 3 with $\tau$-nice sampling in the star strong convexity case, and obtain the following result:

$$\mathbb{E}\left[\|x_K - x_\star\|^2\right] \overset{\text{Theorem 3}}{\leq} \left(1 - \alpha\gamma(2 - \alpha\gamma L_{\gamma,\tau}) \cdot \frac{\mu}{2(1 + \gamma L_{\max})}\right)^K \|x_0 - x_\star\|^2.$$

Since the convergence here is stated in terms of squared distance to the minimizer, we do not need further transformation. Notice that the convergence rate in this case,

$$1 - \alpha\gamma(2 - \alpha\gamma L_{\gamma,\tau}) \cdot \frac{\mu}{2(1 + \gamma L_{\max})},$$

is also minimized when $\alpha = \alpha_{\gamma,\tau} = \frac{1}{\gamma L_{\gamma,\tau}}$. In case of $\alpha = \alpha_{\gamma,\tau}$, the convergence is given by

$$\mathbb{E}\left[\|x_K - x_\star\|^2\right] \leq \left(1 - \frac{\mu}{2L_{\gamma,\tau}(1 + \gamma L_{\max})}\right)^K \|x_0 - x_\star\|^2.$$

This concludes the proof.

### G.7 Proof of Corollary 2

Similar to the proof of Theorem 2, we start with the following identity

$$\|x_{k+1} - x_\star\|^2 = \left\|x_k - \alpha'_{k,G}\gamma\nabla M^\gamma(x_k) - x_\star\right\|^2$$

$$= \|x_k - x_\star\|^2 - \alpha'_{k,G}\gamma\langle\nabla M^\gamma(x_k), x_k - x_\star\rangle + \left(\alpha'_{k,G}\right)^2\gamma^2\|\nabla M^\gamma(x)\|^2. \quad (41)$$

The extrapolation parameter can be rewritten as

$$\alpha'_{k,G} = \frac{1 + \gamma L_{\max}}{\gamma L_{\max}} \cdot \frac{\frac{1}{n}\sum_{i=1}^n\left\|\nabla M^\gamma_{f_i}(x_k)\right\|^2}{\|\nabla M^\gamma(x_k)\|^2}.$$

We have for the last term of (41),

$$\left(\alpha'_{k,G}\right)^2\gamma^2\|\nabla M^\gamma(x_k)\|^2$$

$$= \alpha'_{k,G}\gamma \cdot \left(\gamma + \frac{1}{L_{\max}}\right)\frac{1}{n}\sum_{i=1}^n\left\|\nabla M^\gamma_{f_i}(x_k)\right\|^2$$

$$= \alpha'_{k,G}\gamma \cdot \left(\gamma + \frac{1}{L_{\max}}\right) \cdot \frac{1}{n}\sum_{i=1}^n\left\|\nabla M^\gamma_{f_i}(x_k) - \nabla M\gamma f_i x_\star\right\|^2$$

$$\leq \alpha'_{k,G}\gamma \cdot \left(\gamma + \frac{1}{L_{\max}}\right) \cdot \frac{1}{n}\sum_{i=1}^n\frac{L_i}{1 + \gamma L_i}\left(D_{M^\gamma_{f_i}}(x_k, x_\star) + D_{M^\gamma_{f_i}}(x_\star, x_k)\right),$$

where the last inequality follows from the $\frac{L_i}{1+\gamma L_i}$-smoothness of $M_{f_i}^\gamma$. Utilizing the monotonicity of $\frac{x}{1+\gamma x}$, for $x > 0$, we further obtain

$$\left(\alpha'_{k,G}\right)^2 \gamma^2 \left\|\nabla M^\gamma\left(x_k\right)\right\|^2$$

$$\overset{\text{Fact 4}}{\leq} \alpha'_{k,G}\gamma \cdot \left(\gamma + \frac{1}{L_{\max}}\right) \cdot \frac{L_{\max}}{1+\gamma L_{\max}} \cdot \frac{1}{n} \sum_{i=1}^n \left(D_{M_{f_i}^\gamma}\left(x_k, x_\star\right) + D_{M_{f_i}^\gamma}\left(x_\star, x_k\right)\right)$$

$$= \alpha'_{k,G}\gamma \cdot \left(\gamma + \frac{1}{L_{\max}}\right) \cdot \frac{L_{\max}}{1+\gamma L_{\max}} \cdot \left(D_{M^\gamma}\left(x_k, x_\star\right) + D_{M^\gamma}\left(x_\star, x_k\right)\right)$$

$$= \alpha'_{k,G}\gamma \left(D_{M^\gamma}\left(x_k, x_\star\right) + D_{M^\gamma}\left(x_\star, x_k\right)\right). \tag{42}$$

For the second term of (41), we have

$$-2\alpha'_{k,G}\gamma \left\langle \nabla M^\gamma\left(x_k\right), x_k - x_\star\right\rangle = 2\alpha'_{k,G}\gamma \left\langle \nabla M^\gamma\left(x_k\right), x_\star - x_k\right\rangle$$

$$= 2\alpha'_{k,G}\gamma \left\langle \nabla M^\gamma\left(x_k\right) - \nabla M^\gamma\left(x_\star\right), x_\star - x_k\right\rangle$$

$$= -2\alpha'_{k,G}\gamma \left(D_{M^\gamma}\left(x_k, x_\star\right) + D_{M^\gamma}\left(x_\star, x_k\right)\right). \tag{43}$$

Plugging (43) and (42) into (41), we obtain

$$\left\|x_{k+1} - x_\star\right\|^2 \leq \left\|x_k - x_\star\right\|^2 - \alpha'_{k,G}\gamma \left(D_{M^\gamma}\left(x_k, x_\star\right) + D_{M^\gamma}\left(x_\star, x_k\right)\right).$$

Notice that we know that

$$D_{M^\gamma}\left(x_k, x_\star\right) \overset{(14)}{=} M^\gamma\left(x_k\right) - M^\gamma\left(x_\star\right), \quad D_{M^\gamma}\left(x_\star, x_k\right) \overset{(15)}{\geq} 0.$$

As a result, we have

$$\left\|x_{k+1} - x_\star\right\|^2 \leq \left\|x_k - x_\star\right\|^2 - \alpha'_{k,G}\gamma \left(M^\gamma\left(x_k\right) - M^\gamma\left(x_\star\right)\right).$$

Summing up the above recursion for $k = 0, 1, ..., K - 1$, we notice that many of them telescope, we obtain

$$\gamma \sum_{k=0}^{K-1} \alpha'_{k,G}\left(M^\gamma\left(x_k\right) - \inf M^\gamma\right) \leq \left\|x_0 - x_\star\right\|^2.$$

Denote $p_k = \alpha'_{k,G}/\sum_{k=0}^{K-1}\alpha'_{k,G}$ for $k = 0, 1, ..., K - 1$. If we sample $\bar{x}_K$ randomly according to probabilities $p_k$ from the first $K$ iterates $\{x_0, x_1, \ldots, x_{K-1}\}$, we can further write the above recursion as

$$\mathbb{E}\left[M^\gamma\left(\bar{x}_K\right)\right] - \inf M^\gamma \leq \frac{1}{\gamma} \cdot \frac{\left\|x_0 - x_\star\right\|^2}{\sum_{k=0}^{K-1}\alpha'_{k,G}}.$$

Utilizing the local bound in Lemma 10, we further obtain,

$$\mathbb{E}\left[f(\bar{x}^K)\right] - f^{\inf} \leq \left(\frac{1}{\gamma} + L_{\max}\right) \cdot \frac{\left\|x_0 - x_\star\right\|^2}{\sum_{k=0}^{K-1}\alpha'_{k,G}}.$$

This concludes the proof.

## H  Missing proofs of lemmas

### H.1  Proof of Lemma 1

Notice that since $f$ is proper, closed and convex, by Fact 1, $\text{prox}_{\gamma f}\left(x\right)$ is a singleton. We use the notation $z(x) = \text{prox}_{\gamma f}\left(x\right)$. Using the definition of $\text{prox}_{\gamma f}\left(x\right)$, we see that

$$M_f^\gamma\left(x\right) = f(z(x)) + \frac{1}{2\gamma}\left\|z(x) - x\right\|^2$$

$$= f\left(\text{prox}_{\gamma f}\left(x\right)\right) + \frac{1}{2\gamma}\left\|\text{prox}_{\gamma f}\left(x\right) - x\right\|^2.$$

Now, assume $M_f^\gamma\left(x\right) = +\infty$. We have for any $z \in \mathbb{R}^d$,

$$+\infty = M_f^\gamma\left(x\right) = f\left(z(x)\right) + \frac{1}{2\gamma}\left\|z(x) - x\right\|^2 \leq f(z) + \frac{1}{2\gamma}\left\|z - x\right\|^2,$$

which means that $z$ is also optimal, which contradicts the uniqueness $z(x) = \text{prox}_{\gamma f}\left(x\right)$. This indicates that $M_f^\gamma\left(x\right) < +\infty$, thus, it is real-valued, which concludes the proof.

## H.2 Proof of Lemma 2

Let $f^\star$ be the convex conjugate of $f$, using Corollary 6.56 in the book by Beck [2017], we have $\left(M_f^\gamma\right)^\star = f^\star + \frac{\gamma}{2}\|\cdot\|^2$. We know that the convex conjugate of a proper, closed and convex function is also proper closed and convex. As a result, $f^\star + \frac{\gamma}{2}\|\cdot\|^2$ is $\gamma$-strongly convex. This indicates that $\left(M_f^\gamma\right)^\star$ is $\gamma$-strongly convex, which implies $M_f^\gamma$ is $\frac{1}{\gamma}$-smooth. Notice that we have

$$\text{prox}_{\gamma f}(x) = \arg\min_{z \in \mathbb{R}^d} \left\{ f(z) + \frac{1}{2\gamma}\|z - x\|^2 \right\},$$

by the definition of proximity operator. Using Theorem 5.30 from Beck [2017], we have

$$\nabla M_f^\gamma(x) = \frac{1}{\gamma}\left(x - \text{prox}_{\gamma f}(x)\right).$$

This completes the proof.

## H.3 Proof of Lemma 3

To prove this lemma, we use Theorem 2.19 in the book by Beck [2017]. From the key observation that $M_f^\gamma$ is the infimal convolution of the proper, convex function $f$ and the real-valued convex function $\frac{1}{2\gamma}\|\cdot\|^2$, we deduce that $M_f^\gamma$ is convex. This completes the proof.

## H.4 Proof of Lemma 4

Let $f^\star$ be the convex conjugate of $f$. From Corollary 6.56 in the book by Beck [2017], it holds that $\left(M_f^\gamma\right)^\star = f^\star + \frac{\gamma}{2}\|\cdot\|^2$. Since $f$ is $L$-smooth, we deduce that $f^\star$ is $\frac{1}{L}$-strongly convex, and thus $\left(M_f^\gamma\right)^\star$ is $\frac{1}{L} + \gamma$-strongly convex. This suggests that $\left(M_f^\gamma\right)^\star$ is $\frac{1+\gamma L}{L}$-strongly convex, which in turn implies $M_f^\gamma$ is $\frac{L}{1+\gamma L}$-smooth. This completes the proof.

## H.5 Proof of Lemma 5

Notice that since $M_f^\gamma$ is convex and differentiable, the condition $\nabla M_f^\gamma(x) = 0$ gives its set of minimizers. This optimality condition can be written exactly as $x = \text{prox}_{\gamma f}(x)$ according to Lemma 2. Using Lemma 12, we know this condition also gives the set of minimizers of $f$, which suggests that $f$ and $M_f^\gamma$ have the same set of minimizers. Pick any $x_\star \in \mathbb{R}^d$ that is a minimizer of $f$, using Lemma 1, we have

$$
\begin{aligned}
\inf M_f^\gamma = M_f^\gamma(x_\star) \\
= f\left(\text{prox}_{\gamma f}(x_\star)\right) + \frac{1}{2\gamma}\left\|x_\star - \text{prox}_{\gamma f}(x_\star)\right\|^2 \\
= f(x_\star) = \inf f.
\end{aligned}
$$

This completes the proof.

## H.6 Proof of Lemma 6

For any $x \in \mathbb{R}^d$, we have

$$
\begin{aligned}
M_f^\gamma(x) = \min_{z \in \mathbb{R}^d} \left\{ f(z) + \frac{1}{2\gamma}\|z - x\|^2 \right\} \\
\leq f(x) + \frac{1}{2\gamma}\|x - x\|^2 \\
= f(x).
\end{aligned}
$$

This completes the proof.

## H.7 Proof of Lemma 7

From Lemma 3 and Lemma 4, we immediately obtain that each $M_{f_i}^\gamma$ is convex and $\frac{L_i}{1+\gamma L_i}$-smooth. This immediately suggests that $M = \frac{1}{n}\sum_{i=1}^n M_{f_i}^\gamma$ is convex and $L_\gamma$-smooth with

$$L_\gamma \leq \frac{1}{n}\sum_{i=1}^n \frac{L_i}{1+\gamma L_i}.$$

Then by Lemma 13, we have

$$\frac{1}{n^2}\sum_{i=1}^n \frac{L_i}{1+\gamma L_i} \overset{\text{Lemma 13}}{\leq} L_\gamma.$$

Combing the above two inequalities, we have

$$\frac{1}{n^2}\sum_{i=1}^n \frac{L_i}{1+\gamma L_i} \leq L_\gamma \leq \frac{1}{n}\sum_{i=1}^n \frac{L_i}{1+\gamma L_i}.$$

We then look at the condition number defined in Theorem 1. It is easy to verify that

$$C(\gamma, n, \alpha_{\gamma,n}) = L_\gamma(1+\gamma L_{\max}) \quad \text{and,} \quad C(\gamma, 1, \alpha_{\gamma,1}) = L_{\max}.$$

As a result,

$$C(\gamma, n, \alpha_{\gamma,n}) = L_\gamma(1+\gamma L_{\max})$$

$$\leq \frac{1}{n}\sum_{i=1}^n L_i \cdot \frac{1+\gamma L_{\max}}{1+\gamma L_i}$$

$$\leq L_{\max} = C(\gamma, n, 1),$$

Notice that we can write $C(\gamma, \tau, \alpha_{\gamma,\tau})$ as an interpolation between $C(\gamma, n, \alpha_{\gamma,n})$ and $C(\gamma, 1, \alpha_{\gamma,1})$, therefore

$$L_\gamma(1+\gamma L_{\max}) \leq C(\gamma, n, \alpha_{\gamma,n}) \leq C(\gamma, \tau, \alpha_{\gamma,\tau}) \leq C(\gamma, 1, \alpha_{\gamma,1}) = L_{\max}.$$

In cases where there exists at least one $L_i < L_{\max}$, we have

$$\frac{1}{n}\sum_{i=1}^n L_i \cdot \frac{1+\gamma L_{\max}}{1+\gamma L_i} < L_{\max}.$$

which is true for all $0 < \gamma < +\infty$. Thus, $C(\gamma, n, \alpha_{\gamma,n}) < C(\gamma, \tau, \alpha_{\gamma,\tau}) < L_{\max} = C(\gamma, 1, \alpha_{\gamma,1})$. Now we give an example that when all $L_i = L_{\max}$, still $C(\gamma, n, \alpha_{\gamma,n}) = \frac{1}{n}C(\gamma, 1, \alpha_{\gamma,1}) = \frac{1}{n}L_{\max}$.

**Example 1.** *Consider the setting where $f_i : \mathbb{R}^d \mapsto \mathbb{R}$ is defined as $f_i(x) = \frac{\theta}{2}x_i^2$ for some $\theta > 0$. Here $x_i$ denotes the $i$-th coordinate of the vector $x \in \mathbb{R}^d$, $f : \mathbb{R}^d \mapsto \mathbb{R}$ is given by $f(x) = \frac{\theta}{2n}\|x\|^2$. It is easy to show that for each $f_i$ is a convex, $\theta$-smooth function and the smoothness constant $\theta$ cannot be improved since*

$$\frac{\theta}{2}\|x\|^2 - \frac{\theta}{2}x_i^2 = \frac{\theta}{2}\sum_{j\neq i}x_j^2.$$

*For $f(x) = \frac{\theta}{2n}\|x\|^2$, apparently, it is $\frac{\theta}{n}$-smooth and convex. We have the following formula for the Moreau envelope of $f_i(x)$,*

$$M_{f_i}^\gamma(x) = \frac{1}{2}\cdot\frac{\theta}{1+\gamma\theta}\cdot x_i^2.$$

*As expected, each one of them is convex and $\frac{\theta}{1+\gamma\theta}$-smooth. For $M^\gamma(x)$, it is given by*

$$M^\gamma(x) = \frac{1}{n}\sum_{i=1}^n M_{f_i}^\gamma(x) = \frac{1}{2}\cdot\frac{\theta}{n(1+\gamma\theta)}\cdot\|x\|^2,$$

*thus, we know it is convex and $L_\gamma = \frac{\theta}{n(1+\gamma\theta)}$-smooth. In this case*

$$\frac{L_{\max}}{L_\gamma(1+\gamma L_{\max})} = \frac{\theta}{\frac{\theta}{n(1+\gamma\theta)}\cdot(1+\gamma\theta)} = n,$$

*which is*

$$L_\gamma(1+\gamma L_{\max}) = C(\gamma, n, \alpha_{\gamma,n}) = \frac{1}{n}C(\gamma, 1, \alpha_{\gamma,1}) = \frac{1}{n}L_{\max}.$$

## H.8  Proof of Lemma 8

By Lemma 5, we know that $f_i$ and $M_{f_i}^\gamma$ have the same set of minimizers and minimum. Denote the set of minimizers as $\mathcal{X}_i$, since we are in the interpolation regime, we know that the set of minimizers of $f$ is given by,

$$\mathcal{X} = \bigcap_{i=1}^n \mathcal{X}_i \neq \emptyset.$$

Now we prove that every $x$ in $\mathcal{X}$ is a minimizer of $M = \frac{1}{n}\sum_{i=1}^n M_{f_i}^\gamma$. This is true since $x \in \mathcal{X}$ minimizes each $f_i$, thus $M_{f_i}^\gamma$ at the same time. The minimum is given by

$$\inf M = \frac{1}{n}\sum_{i=1}^n \inf M_{f_i}^\gamma = \frac{1}{n}\sum_{i=1}^n \inf f_i = \inf f.$$

We then prove that every $x \notin \mathcal{X}$ is not a minimizer of $f$. If $x \notin \mathcal{X}$, there exists at least one set $\mathcal{X}_j$ such that $x \notin \mathcal{X}_j$. Thus $M_{f_j}^\gamma(x) > \inf M_{f_j}^\gamma$. This indicates that $M^\gamma(x) > \inf M$, which means, $x \notin \mathcal{X}$ is not a minimizer of $M$.

## H.9  Proof of Lemma 9

From Lemma 6, it is clear that $M_f^\gamma$ is a global lower bound of $f$ satisfying $M_f^\gamma(x) \leq f(x)$ for any $x \in \mathbb{R}^d$ and $\gamma > 0$. Notice that the definition of $M^\gamma$ indicates that

$$
\begin{aligned}
M^\gamma(x) &= \frac{1}{n}\sum_{i=1}^n M_{f_i}^\gamma(x) \\
&= \frac{1}{n}\sum_{i=1}^n \min_{z_i \in \mathbb{R}^d}\left\{ f_i(z_i) + \frac{1}{2\gamma}\|z_i - x\|^2 \right\} \\
&\leq \min_{z \in \mathbb{R}^d}\left\{ \frac{1}{n}\sum_{i=1}^n \left( f_i(z) + \frac{1}{2\gamma}\|z - x\|^2 \right) \right\} \\
&= \min_{z \in \mathbb{R}^d}\left\{ \frac{1}{n}\sum_{i=1}^n f_i(z) + \frac{1}{2\gamma}\|z - x\|^2 \right\} \\
&= M_f^\gamma(x),
\end{aligned}
$$

holds for any $x \in \mathbb{R}^d$ and $\gamma > 0$. Combining the above result, we have $M^\gamma(x) \leq M_f^\gamma(x) \leq f(x)$ for any $x \in \mathbb{R}^d$ and $\gamma > 0$. Notice that in Lemma 8, we have already shown that $M^\gamma$ and $f$ have the same set of minimizers and minimum in the interpolation regime. A direct application of Lemma 5 indicates that $M_f^\gamma$ and $f$ have the same set of minimizers and minimum. Therefore, combining the above statement, we know that $M^\gamma$, $M_f^\gamma$ and $f$ have the same set of minimizers and minimum. Thus, for any $x_\star$ belongs to the set of minimizers, we have

$$M^\gamma(x_\star) = M_f^\gamma(x_\star) = f(x_\star).$$

This completes the proof.

## H.10  Proof of Lemma 10

We start from noticing that according to Lemma 1, the following identity is true for Moreau envelope,

$$M_{f_i}^\gamma(x) = f_i(\mathrm{prox}_{\gamma f_i}(x)) + \frac{1}{2\gamma}\left\| x - \mathrm{prox}_{\gamma f_i}(x) \right\|^2. \tag{44}$$

For the second squared norm term, we have the following inequality due to the smoothness of each $f_i$ and the fact that $\nabla f_i\left(\mathrm{prox}_{\gamma f_i}(x)\right) = \frac{1}{\gamma}\left(x - \mathrm{prox}_{\gamma f_i}(x)\right)$,

$$
\begin{aligned}
\left\| x - \mathrm{prox}_{\gamma f_i}(x) \right\|^2 &= \left\langle x - \mathrm{prox}_{\gamma f_i}(x), x - \mathrm{prox}_{\gamma f_i}(x) \right\rangle \\
&= \gamma\left\langle \nabla f_i(\mathrm{prox}_{\gamma f_i}(x)), x - \mathrm{prox}_{\gamma f_i}(x) \right\rangle \\
&\geq \gamma\left(f_i(x) - f_i\left(\mathrm{prox}_{\gamma f_i}(x)\right)\right) - \frac{\gamma L_i}{2}\left\| x - \mathrm{prox}_{\gamma f_i}(x) \right\|^2,
\end{aligned}
$$

which leads to the following lower bound:

$$\left\| x - \mathrm{prox}_{\gamma f_i}(x) \right\|^2 \geq \frac{1}{\frac{1}{\gamma} + \frac{L_i}{2}} \left( f_i(x) - f_i\left( \mathrm{prox}_{\gamma f_i}(x) \right) \right).$$

Plug in the above inequality into (44) and notice that $\inf M = \frac{1}{n}\sum_{i=1}^n \inf M_{f_i}^\gamma = \frac{1}{n}\sum_{i=1}^n \inf f_i$, we obtain the following lower bound on $M_{f_i}^\gamma(x)$,

$$M_{f_i}^\gamma(x) - \inf M_{f_i}^\gamma \geq f_i\left(\mathrm{prox}_{\gamma f_i}(x)\right) + \frac{1}{2 + \gamma L_i}\left(f_i(x) - f_i\left(\mathrm{prox}_{\gamma f_i}(x)\right)\right) - \inf f_i$$

$$= \frac{1}{2 + \gamma L_i}\left(f_i(x) - \inf f_i\right) + \left(1 - \frac{1}{2 + \gamma L_i}\right)\left(f_i(\mathrm{prox}_{\gamma f_i}(x)) - \inf f_i\right).$$

(45)

Now let us look at the term $f_i\left(\mathrm{prox}_{\gamma f_i}(x)\right) - \inf f$. Using again $L_i$-smoothness of $f_i$, we have

$$f_i(x) - f_i(\mathrm{prox}_{\gamma f_i}(x)) - \left\langle \nabla f_i(\mathrm{prox}_{\gamma f_i}(x)), x - \mathrm{prox}_{\gamma f_i}(x)\right\rangle \leq \frac{L_i}{2}\left\| x - \mathrm{prox}_{\gamma f_i}(x)\right\|^2.$$

Notice that $x - \mathrm{prox}_{\gamma f_i}(x) = \gamma \nabla f_i(\mathrm{prox}_{\gamma f_i}(x))$. As a result, we have,

$$f_i(x) - \gamma \left\| \nabla f_i(\mathrm{prox}_{\gamma f_i}(x))\right\|^2 - \frac{L_i\gamma^2}{2}\left\| \nabla f_i(\mathrm{prox}_{\gamma f_i}(x))\right\|^2 \leq f_i(\mathrm{prox}_{\gamma f_i}(x)),$$

which is

$$f_i(x) - \inf f_i - \left(\gamma + \frac{\gamma^2 L_i}{2}\right)\left\| \nabla f_i(\mathrm{prox}_{\gamma f_i}(x))\right\|^2 \leq f_i\left(\mathrm{prox}_{\gamma f_i}(x)\right) - \inf f_i.$$

Using the interpolation regime assumption, we have

$$\left\| \nabla f_i\left(\mathrm{prox}_{\gamma f_i}(x)\right)\right\|^2 = \left\| \nabla f_i\left(\mathrm{prox}_{\gamma f_i}(x)\right) - \nabla f_i(x_\star)\right\|^2$$
$$\leq 2L_i D_{f_i}\left(\mathrm{prox}_{\gamma f_i}(x), x_\star\right)$$
$$= 2L_i\left(f_i(\mathrm{prox}_{\gamma f_i}(x)) - \inf f_i\right),$$

where the inequality is obtained using Fact 3. As a result, we obtain the following bound,

$$f_i\left(\mathrm{prox}_{\gamma f_i}(x)\right) - \inf f_i \geq \frac{1}{1 + \gamma L_i(2 + \gamma L_i)}\left(f_i(x) - \inf f_i\right)$$

$$= \frac{1}{(1 + \gamma L_i)^2}\left(f_i(x) - \inf f_i\right).$$

Plug the above lower bound into (45), we obtain

$$M_{f_i}^\gamma(x) - \inf M_{f_i}^\gamma \geq \frac{1}{1 + \gamma L_i}\left(f_i(x) - \inf f_i\right), \tag{46}$$

Notice that we have $M^\gamma(x) = \frac{1}{n}\sum_{i=1}^n M_{f_i}^\gamma(x)$. Since we are in the interpolation regime, from Lemma 9, we know that

$$\inf M^\gamma = M^\gamma(x_\star) = \frac{1}{n}\sum_{i=1}^n M_{f_i}^\gamma(x_\star) = \frac{1}{n}\sum_{i=1}^n \inf M_{f_i}^\gamma,$$

and

$$\inf f = f(x_\star) = \frac{1}{n}\sum_{i=1}^n f_i(x_\star) = \frac{1}{n}\sum_{i=1}^n \inf f_i.$$

We average (46) for each $i \in [n]$ and obtain

$$M^\gamma(x) - \inf M^\gamma \geq \frac{1}{n}\sum_{i=1}^n \frac{1}{1 + \gamma L_i}\left(f_i(x) - \inf f_i\right)$$

$$\geq \frac{1}{1 + \gamma L_{\max}} \cdot \frac{1}{n}\sum_{i=1}^n \left(f_i(x) - \inf f_i\right)$$

$$= \frac{1}{1 + \gamma L_{\max}}\left(f(x) - \inf f\right).$$

This concludes the proof.

## H.11 Proof of Lemma 11

We start with picking any point $x \in \mathbb{R}^d$, since we are in the interpolation regime, according to Lemma 9, we have $M^\gamma(x_\star) = f(x_\star)$. Applying Lemma 10, we get

$$M^\gamma(x) - M^\gamma(x_\star) \geq \frac{1}{1 + \gamma L_{\max}} \left( f(x) - f(x_\star) \right). \tag{47}$$

We know that from the $\mu$-strong convexity of $f$, we have for any $x \in \mathbb{R}^d$,

$$f(x) - f(x_\star) - \langle \nabla f(x_\star), x - x_\star \rangle \geq \frac{\mu}{2} \|x - x_\star\|^2.$$

Notice that since $\nabla f(x_\star) = 0$, we have

$$f(x) - f(x_\star) \geq \frac{\mu}{2} \|x - x_\star\|^2. \tag{48}$$

Combining the above two inequalities (47) and (48), we have

$$M^\gamma(x) - M^\gamma(x_\star) \geq \frac{\mu}{1 + \gamma L_{\max}} \cdot \frac{1}{2} \|x - x_\star\|^2.$$

This concludes the proof.

## H.12 Proof of Lemma 12

Notice that $x \in \mathbb{R}^d$ is a minimizer of $f$ if and only if $0 \in \partial f(x)$. This inclusion holds if and only if $0 \in \partial(\gamma f(x))$, which can be rewritten as $x - x \in \partial(\gamma f(x))$. By the equivalence of (i) and (ii) in Fact 2, the above condition is the same as $x = \operatorname{prox}_{\gamma f}(x)$.

## H.13 Proof of Lemma 13

Since each $f_i$ is $L_i$-smooth, the following function is convex for every $i \in [n]$,

$$\frac{L_i}{2} \|x\|^2 - f_i(x).$$

Thus,

$$\frac{\frac{1}{n} \sum_{i=1}^n L_i}{2} \|x\|^2 - \frac{1}{n} \sum_{i=1}^n f_i(x),$$

is also a convex function, which indicates $f(x)$ is also $\frac{1}{n} \sum_{i=1}^n L_i$-smooth. This means that

$$L \leq \frac{1}{n} \sum_{i=1}^n L_i. \tag{49}$$

Now notice that the $L$-smoothness of $f$ is equivalent to the following function being convex

$$\frac{nL}{2} \|x\|^2 - \sum_{i=1}^n f_i(x).$$

Pick any $j \in [n]$, we have

$$\frac{nL}{2} \|x\|^2 - \sum_{i=1}^n f_i(x) + \sum_{1 \leq i \neq j \leq n} f_i(x) = \frac{nL}{2} \|x\|^2 - f_j(x).$$

Since all functions are convex and the sum of convex functions is convex,

$$\frac{nL}{2} \|x\|^2 - f_j(x),$$

is convex, which indicates that $f_j(x)$ is also $nL$-smooth. As a result, for every $j \in [n]$, we have $nL \geq L_j$. Summing up the inequality for every $j \in [n]$, we have

$$\frac{1}{n^2} \sum_{j=1}^n L_j \leq L. \tag{50}$$

Combining (49) and (50), we have

$$\frac{1}{n^2}\sum_{i=1}^{n} L_i \le L \le \frac{1}{n}\sum_{i=1}^{n} L_i.$$

In order to demonstrate that both bounds are tight in the above inequality, we consider cases where they are identities.

(i): Consider the case that each function $f_i(x) = \frac{1}{2} \cdot L_i \cdot \|x\|^2$, it is easy to see that $f(x) = \frac{1}{2} \cdot \left(\frac{1}{n}\sum_{i=1}^{n} L_i\right) \cdot \|x\|^2$. In this case $L = \frac{1}{n}\sum_{i=1}^{n} L_i$, the upper bound is an identity.

(ii): Consider the case that each function $f_i(x) = \frac{1}{2} \cdot \theta \cdot x_i^2$, where $\theta > 0$ is a constant, $x_i$ is the $i$-th coordinate of $x \in \mathbb{R}^d$. In this case $f(x) = \frac{1}{2} \cdot \frac{\theta}{n} \|x\|^2$. It is easy to verify that in this case $L_i = \theta$, $L = \frac{1}{n}\theta$. Thus $\frac{1}{n^2}\sum_{i=1}^{n} L_i = L$, the lower bound is an identity.

This concludes the proof.

### H.14 Proof of Lemma 14

From the definition of $C(\gamma, \tau, 1)$ and $C(\gamma, \tau, \alpha_{\gamma,\tau})$, we know that

$$\frac{C(\gamma, \tau, 1)}{C(\gamma, \tau, \alpha_{\gamma,\tau})} = \frac{1}{\gamma L_{\gamma,\tau}(2 - \gamma L_{\gamma,\tau})}.$$

Now let $t = \gamma L_{\gamma,\tau}$, we have the following bound on $t$ according to the definition of $L_{\gamma,\tau}$ given in Theorem 1.

$$\begin{aligned} t &= \gamma L_{\gamma,\tau} \\ &= \frac{n-\tau}{\tau(n-1)} \cdot \frac{\gamma L_{\max}}{1 + \gamma L_{\max}} + \frac{n(\tau-1)}{\tau(n-1)} \cdot \gamma L_\gamma. \end{aligned}$$

Notice that in Lemma 7, we have shown that

$$L_\gamma \overset{\text{Lemma 7}}{\le} \frac{1}{n}\sum_{i=1}^{n}\frac{L_i}{1+\gamma L_i},$$

and due to Fact 4, we have

$$\frac{1}{n}\sum_{i=1}^{n}\frac{L_i}{1+\gamma L_i} \overset{\text{Fact 4}}{\le} \frac{L_{\max}}{1+\gamma L_{\max}}.$$

As a result,

$$t \le \frac{n-\tau}{\tau(n-1)} \cdot \frac{\gamma L_{\max}}{1 + \gamma L_{\max}} + \frac{n(\tau-1)}{\tau(n-1)} \cdot \frac{\gamma L_{\max}}{1 + \gamma L_{\max}} = \frac{\gamma L_{\max}}{1 + \gamma L_{\max}} < 1.$$

It is easy to show that $g(t) = \frac{1}{t(2-t)}$ is monotone decreasing when $t \in [0,1]$, thus

$$\begin{aligned} \frac{C(\gamma, \tau, 1)}{C(\gamma, \tau, \alpha_{\gamma,\tau})} &\ge \frac{1}{\frac{\gamma L_{\max}}{1+\gamma L_{\max}}\left(1 - \frac{\gamma L_{\max}}{1+\gamma L_{\max}}\right)} \\ &= 2 + \frac{1}{\gamma L_{\max}} + \gamma L_{\max} \\ &\overset{\text{AM-GM}}{\ge} 4, \end{aligned}$$

where the last inequality is due to the AM-GM inequality. This concludes the proof.

## H.15 Proof of Lemma 15

As suggested by Lemma 7, we have

$$C\left(\gamma, n, \alpha_{\gamma,n}\right) \leq C\left(\gamma, \tau, \alpha_{\gamma,\tau}\right) \leq C\left(\gamma, 1, \alpha_{\gamma,1}\right), \quad \forall \tau \in [n].$$

Notice that $\alpha_{\gamma,\tau}$ is given by

$$\alpha_{\gamma,\tau} = \frac{1}{\gamma L_{\gamma,\tau}},$$

and we know that

$$L_{\gamma,\tau} = \frac{n-\tau}{\tau(1-n)} \cdot \frac{L_{\max}}{1+\gamma L_{\max}} + \frac{n(\tau-1)}{\tau(n-1)} \cdot L_\gamma.$$

From Lemma 7 and Fact 4, we know that

$$L_\gamma \overset{\text{Lemma 7}}{\leq} \frac{1}{n} \sum_{i=1}^{n} \frac{L_i}{1+\gamma L_i} \overset{\text{Fact 4}}{\leq} \frac{L_{\max}}{1+\gamma L_{\max}}.$$

Consequently, $L_{\gamma,\tau}$ decreases as $\tau$ increases. Therefore, $\alpha_{\gamma,\tau}$ increases with the increase of $\tau$, as illustrated by the following inequality

$$\alpha_{\gamma,1} \leq \alpha_{\gamma,\tau} \leq \alpha_{\gamma,n}, \quad \forall \tau \in [n].$$

This concludes the proof.

## H.16 Proof of Lemma 16

We refer the readers to the proof of Lemma 3.1 of Böhm and Wright [2021].

## H.17 Proof of Lemma 17

We refer the readers to the proof of Proposition 7 of Yu et al. [2015].

## H.18 Proof of Lemma 18

Observe that since $0 < \gamma < \frac{1}{\rho}$, we do have $f + \frac{1}{2} \cdot \frac{1}{\gamma} \|\cdot\|^2$ being strongly convex. This indicates that $\text{prox}_{\gamma f}$ is always a singleton and therefore $M_f^\gamma$ is differentiable, as suggested by Lemma 16. Notice that $x$ is stationary point of $M_f^\gamma$ if and only if $\nabla M_f^\gamma(x) = 0$. This is equivalent to $\frac{1}{\gamma}\left(x - \text{prox}_{\gamma f}(x)\right) = 0$, which is $x = \text{prox}_{\gamma f}(x)$. In addition, $x = \text{prox}_{\gamma f}(x)$ is equivalent to

$$\nabla f(x) + \frac{1}{\gamma}(x - x) = 0,$$

which is $\nabla f(x) = 0$. Combining the above statements, we have $\nabla f(x) = 0$ if and only if $\nabla M_f^\gamma(x) = 0$. This suggests that the two functions have the same set of stationary points.

## H.19 Proof of Lemma 19

Apply Theorem 1 of Khaled and Richtárik [2023], notice that in this case GD satisfy the expected smoothness assumption given in Assumption 2 of Khaled and Richtárik [2023] with $A = 0$, $B = 1$ and $C = 0$, we obtain that when the step size $\eta$ satisfies

$$0 < \eta < \frac{1}{LB} = \frac{1}{L},$$

where $L$ is the smoothness constant of $f$, the iterates of GD satisfy

$$\min_{0 \leq k \leq K-1} \mathbb{E}\left[\|\nabla f(x_k)\|^2\right] \leq \frac{2\left(f(x_0) - \inf f\right)}{\eta K}.$$

This completes the proof.

### H.20 Proof of Lemma 20

Notice that we are in the interpolation regime, by Lemma 8, we know that $f$ and $M^\gamma$ have the same set of minimizers and minimum. As a result,

$$M^\gamma(x_\star) = \frac{1}{n} \sum_{i=1}^{n} M^\gamma_{f_i}(x_\star) \overset{\text{Lemma 8}}{=} f(x_\star). \tag{51}$$

From the above inequality, we obtain that

$$\frac{\frac{1}{n} \sum_{i=1}^{n} \left( M^\gamma_{f_i}(x) - M^\gamma_{f_i}(x_\star) \right)}{\gamma \cdot \left\| \frac{1}{n} \sum_{i=1}^{n} \nabla M^\gamma_{f_i}(x) \right\|^2} \overset{(51)}{=} \frac{M^\gamma(x) - M^\gamma(x_\star)}{\gamma \cdot \|\nabla M^\gamma(x)\|^2}.$$

Then by the smoothness of $M^\gamma$ and Fact 3, we have

$$\frac{M^\gamma(x) - M^\gamma(x_\star)}{\gamma \cdot \|\nabla M^\gamma(x)\|^2} \overset{\text{Fact 3}}{\geq} \frac{\frac{1}{2L_\gamma} \|\nabla M^\gamma(x) - \nabla M^\gamma(x_\star)\|^2}{\gamma \cdot \|\nabla M^\gamma(x)\|^2}$$

$$= \frac{1}{2\gamma L_\gamma}.$$

Thus, by combining the above inequalities, we have

$$\frac{\frac{1}{n} \sum_{i=1}^{n} \left( M^\gamma_{f_i}(x) - M^\gamma_{f_i}(x_\star) \right)}{\gamma \cdot \left\| \frac{1}{n} \sum_{i=1}^{n} \nabla M^\gamma_{f_i}(x) \right\|^2} \geq \frac{1}{2\gamma L_\gamma}.$$

Notice that from the definition of $\alpha_{k,S}$ for FedExProx-StoPS, we have

$$\alpha_{k,S} = \frac{\frac{1}{n} \sum_{i=1}^{n} \left( M^\gamma_{f_i}(x_k) - M^\gamma_{f_i}(x_\star) \right)}{\gamma \cdot \left\| \frac{1}{n} \sum_{i=1}^{n} \nabla M^\gamma_{f_i}(x_k) \right\|^2} \geq \frac{1}{2\gamma L_\gamma}.$$

Therefore, using the above lower bound, it is straight forward to further relax (12) to

$$\mathbb{E}\left[ f(\bar{x}^K) \right] - \inf f \leq 2L_\gamma \left( 1 + 2\gamma L_{\max} \right) \cdot \frac{\|x_0 - x_\star\|^2}{K}.$$

This concludes the proof.

## I  Experiments

In this section, we describe the settings and results of numerical experiments to demonstrate the effectiveness of our method.

### I.1  Experiment settings

We consider the overparameterized linear regression problem in the finite sum setting

$$\min_{x \in \mathbb{R}^d} \left\{ f(x) = \frac{1}{n} \sum_{i=1}^{n} f_i(x) \right\},$$

where $d$ is the dimension of the problem, $n$ is the total number of clients, each function $f_i$ has the following form

$$f_i(x) = \frac{1}{2} \|A_i x - b_i\|^2,$$

where $A_i \in \mathbb{R}^{n_i \times d}$, $b_i \in \mathbb{R}^{n_i}$. Here $n_i$ is the number of samples on each client. It is easy to see that for each function $f_i$, we have

$$\nabla f_i(x) = A_i^\top A_i x - A_i^\top b_i, \quad \text{and} \quad \nabla^2 f_i(x) = A_i^\top A_i \succeq O_d.$$

Thus, it follows that

$$\nabla f(x) = \frac{1}{n} \sum_{i=1}^{n} \left( \boldsymbol{A}_i^\top \boldsymbol{A}_i x - \boldsymbol{A}_i^\top b_i \right), \text{ and } \quad \nabla^2 f(x) = \frac{1}{n} \sum_{i=1}^{n} \boldsymbol{A}_i^\top \boldsymbol{A}_i \succeq \boldsymbol{O}_d.$$

The problem is therefore convex. Notice that one implicit assumption for the class of proximal point methods in practice is that the proximity operator can be computed efficiently. In the setting of linear regression, we have the following closed form formula for the proximity operator $\text{prox}_{\gamma f_i}$, which holds for any $x \in \mathbb{R}^d$,

$$\text{prox}_{\gamma f_i}(x) = \left( \boldsymbol{A}_i^\top \boldsymbol{A}_i + \frac{1}{\gamma} \boldsymbol{I}_d \right)^{-1} \cdot \left( \boldsymbol{A}_i^\top b_i + \frac{1}{\gamma} x \right). \tag{52}$$

Observe that in the linear regression problem, since we know the closed form expression of each $f_i$ and $f$, we know the corresponding smoothness constant

$$L_i = \lambda_{\max} \left( \boldsymbol{A}_i^\top \boldsymbol{A}_i \right).$$

Notice that from Lemma 1, we have

$$M_{f_i}^\gamma(x) = f_i \left( \text{prox}_\gamma(f_i) \right) + \frac{1}{2\gamma} \left\| x - \text{prox}_\gamma(f_i)(x) \right\|^2.$$

Since we know $\text{prox}_\gamma(f_i)$ in closed form using (52), we also know each local Moreau envelope in closed form, and thus the same for $M^\gamma = \frac{1}{n} \sum_{i=1}^{n} M_{f_i}^\gamma$. As a result, we can deduce $L_\gamma$ for $M^\gamma$. In our experiments, we pick $d \geq \sum_{i=1}^{n} n_i$ so that we are in the interpolation regime. Each $\boldsymbol{A}_i$ is generated randomly from a uniform distribution between $[0, 1)$, and the corresponding vector $b_i$ is also generated from the same uniform distribution. In order to find a minimizer $x_\star$, we run gradient descent for sufficient amount of iterations. All the codes for the experiments are written in Python 3.11 with NumPy and SciPy package. The code was run on a machine with AMD Ryzen 9 5900HX Radeon Graphics @ 3.3 GHz and 8 cores 16 threads. For experiment in the small dimension regime, each algorithm considers here only takes seconds to finish. For larger experiments, depending on the specific implementation, the algorithms typically take a few minutes to half an hour to finish. For `FedProx`, `FedExP` and our method `FedExProx` in the full participation case, the algorithm for a specific dataset is deterministic, while in case where client sampling is taken into account, the randomness of the algorithms comes from the specific sampling strategy used. Our code is publicly available at the following link: https://anonymous.4open.science/r/FedExProx-F262/

## I.2 Large dimension regime

In this section we provide the numerical experiments in the large dimension regime, where $n_i = 20$ for each $i \in [n]$, $n = 30$, $d = 900$.

### I.2.1 Comparison of `FedExProx` and `FedProx`

In this section, we compare the performance of `FedProx` with our method `FedExProx` in the full participation case and in the client partial participation case, demonstrating that the extrapolated counterpart outperforms `FedProx` in iteration complexity. Notice that here we are only concerned with iteration complexity, since the amount of computations is almost the same for the two algorithms. The only difference is that for `FedExProx`, instead of simply averaging the iterates obtained from each client, the server performs extrapolation. From Figure 2, it is easy to see that our proposed algorithm `FedExProx` outperforms `FedProx`, which provides numerical evidence for our theoretical findings. Notably, in order to achieve the small level of function value sub-optimality, `FedExProx` typically requires only half the number of iterations needed by `FedProx`, which indicates a factor of 2 speed up in terms of iteration complexity. Another observation is that, $\alpha_{\gamma,n}$ is decreasing as $\gamma$ increases, which suggests that when local step sizes are small, the practice of simply averaging the iterates is far from optimal.

We also compare the performance of the two algorithms in the client partial participation setting. As one can observe from Figure 3, `FedExProx` still outperforms `FedProx` in the client partial participation setting, which further corroborates our theoretical findings. Observe that $\alpha_{\gamma,\tau}$ here increases as $\tau$ becomes larger, which coincides with our predictions in Remark 7.

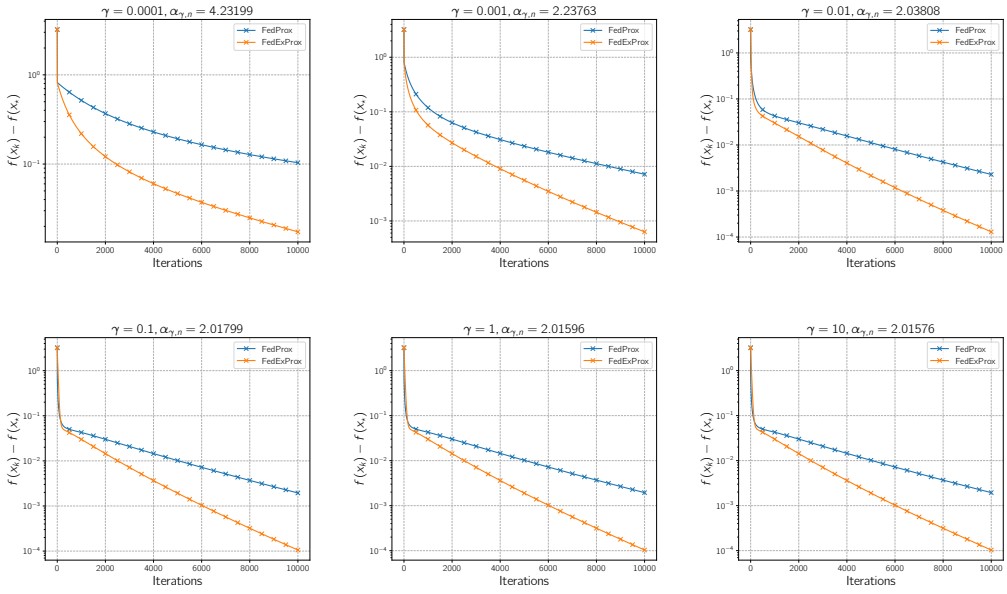

Figure 2: Comparison of convergence of `FedExProx` and `FedProx` in terms of iteration complexity in the full participation setting. For this experiment $\gamma$ is picked from the set $\{0.0001, 0.001, 0.01, 0.1, 1, 10\}$, the $\alpha_{\gamma,n}$ indicates the optimal constant extrapolation parameter as defined in Theorem 1. For each choice of $\gamma$, the two algorithms are run for $K = 10000$ iterations, respectively.

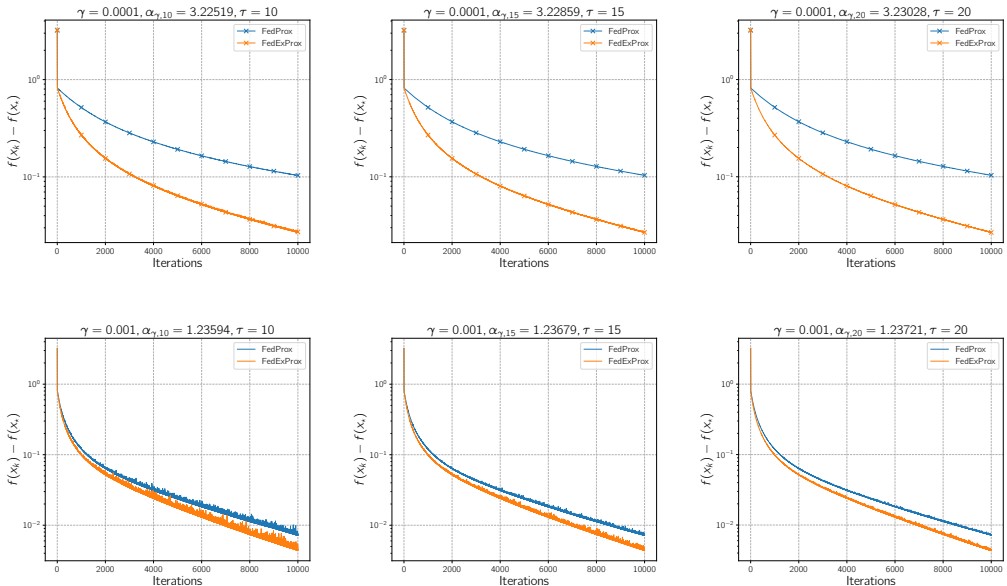

Figure 3: Comparison of convergence of `FedExProx` and `FedProx` in terms of iteration complexity in the client partial participation setting. For this experiment $\gamma$ is picked from the set $\{0.0001, 0.001\}$, the client minibatch size $\tau$ is chosen from $\{10, 15, 20\}$ and the $\alpha_{\gamma,n}$ indicates the optimal constant extrapolation parameter as defined in Theorem 1. For each choice of $\gamma$ and $\tau$, the two algorithm are run for $K = 10000$ iterations, respectively.

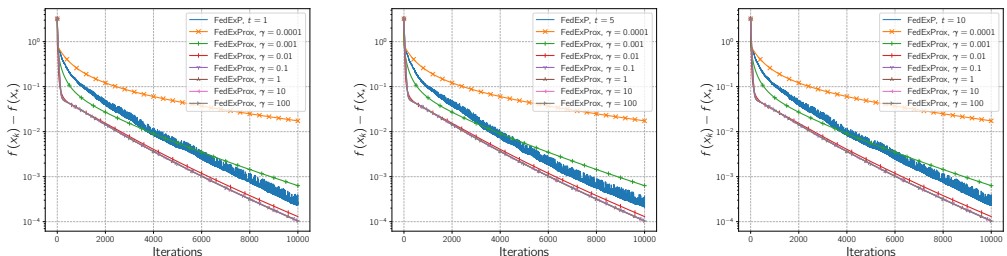

Figure 4: Comparison in terms of iteration complexity for `FedExProx` with different step sizes $\gamma$ chosen from $\{0.0001, 0.001, 0.01, 1, 10, 100\}$ in the full participation setting. In the figure, we use `FedExP` with different iterations of local training $t \in \{1, 5, 10\}$ as a benchmark in the three sub-figures. The local step size for `FedExP` is set to be the largest possible value $\frac{1}{6tL_{\max}}$, where $L_{\max} = \max_i L_i$.

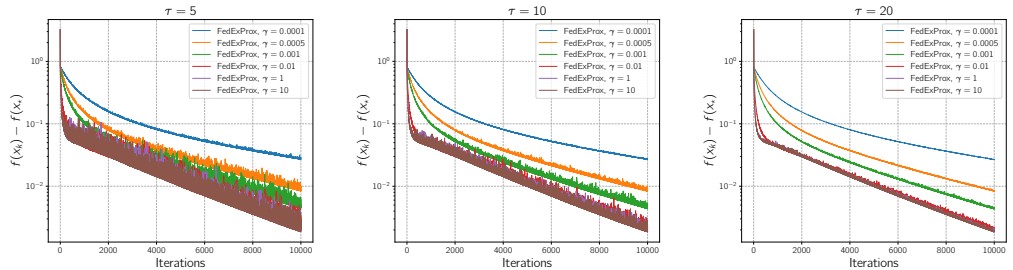

Figure 5: Comparison in terms of iteration complexity for `FedExProx` with different step sizes $\gamma$ chosen from $\{0.0001, 0.0005, 0.01, 1, 10\}$ in the client partial participation case. Different client minibatch sizes are used, the minibatch size $\tau$ is chosen from $\{5, 10, 20\}$.

### I.2.2 Comparison of `FedExProx` with different local step size

In this section, we compare the performance in terms of iteration complexity for `FedExProx` with different local step sizes. We also include `FedExP` as a reference. The local step size of `FedExP` is chosen to be $\frac{1}{6tL_{\max}}$, where $t$ is the number of gradient descent iterations performed by each client for local training, $L_{\max} = \max_i L_i$, where $L_i$ is the smoothness constant of $f_i$.

As one can observe from Figure 4, for our proposed method `FedExProx`, the larger $\gamma$ is, the faster it will converge. However, as $\gamma$ becomes larger, the improvement in iteration complexity becomes trivial at some point. Note that for different $\gamma$, the complexities required to compute the proximity operator locally varies and often larger $\gamma$ requires more computation than smaller $\gamma$. Compared to `FedExP` with the best local step size $\frac{1}{6tL_{\max}}$, `FedExProx` with a large enough $\gamma$ is better in terms of iteration complexity. In the case where the computation of proximity operator is efficient, our method has a better computation complexity as well. Notice that small $\gamma$ leads to slow down of our method, and we do not claim that the iteration complexity of `FedExProx` is always better than `FedExP`. However, it is provable that `FedExProx` indeed has a better worst case iteration complexity. We want to emphasize a key difference between `FedExP` and our method is that we do not have any constraints on the local step size $\gamma$, and our method converges for arbitrary local step size $\gamma > 0$, while for `FedExP`, a misspecified step size could lead to divergence.

We also compare `FedExProx` with different step sizes in the client sampling case, see Figure 5. However, since there is no explicit convergence guarantee for `FedExP` in this case, we did not include `FedExP` in the plot.

In the client partial participation case, the same behavior of how our proposed algorithm `FedExProx` changes according to different local step sizes $\gamma$ is observed. A small $\gamma$ leads to slow convergence of the algorithm, while for large $\gamma$, the convergence is improved. However, at some point, the improvement becomes trivial.

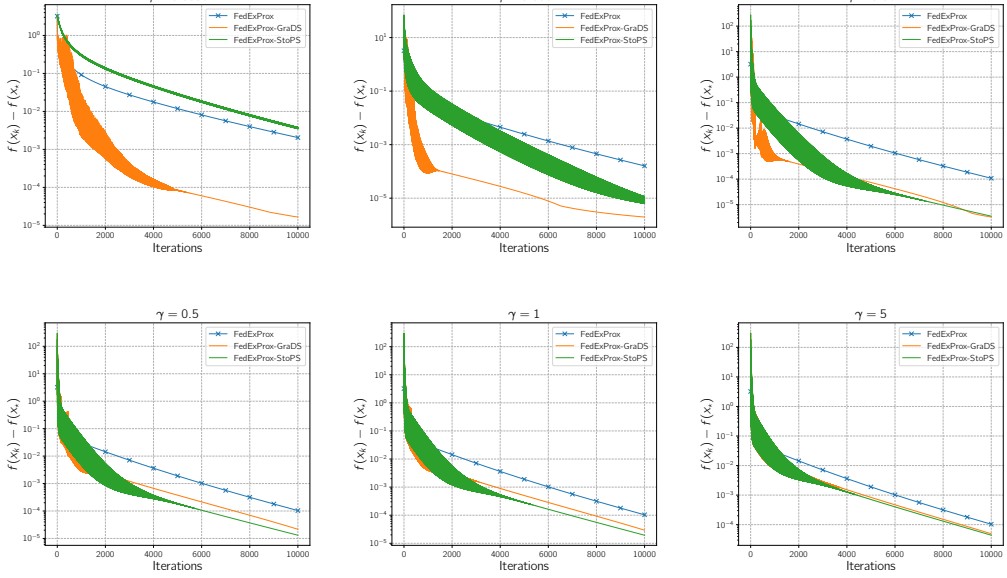

Figure 6: Comparison of `FedExProx`, `FedExProx-GraDS` and `FedExProx-StoPS` in terms of iteration complexity with different step sizes $\gamma$ chosen from $\{0.0005, 0.0005, 0.05, 0.5, 1, 5\}$ in the full participation setting.

### I.2.3 Comparison of `FedExProx` and its adaptive variants

In this section, we compare `FedExProx` and its two adaptive variants `FedExProx-GraDS` and `FedExProx-StoPS`. We first focus on the full participation case. Note that in this case, the all the algorithms are deterministic. For `FedExProx-GraDS`, as it is suggested by Theorem 2, the extrapolation parameter is given by

$$\alpha_k = \alpha_{k,G} := \frac{\frac{1}{n}\sum_{i=1}^{n}\left\|x_k - \text{prox}_{\gamma f_i}(x_k)\right\|^2}{\left\|\frac{1}{n}\sum_{i=1}^{n}\left(x_k - \text{prox}_{\gamma f_i}(x_k)\right)\right\|^2}.$$

The server can use the local iterates it received from each client to compute $\alpha_{k,G}$ directly. If, in addition, we know $L_{\max}$, we can implement a version that has a better theoretical guarantee,

$$\alpha_{k,G} := \frac{1 + \gamma L_{\max}}{\gamma L_{\max}} \cdot \frac{\frac{1}{n}\sum_{i=1}^{n}\left\|x_k - \text{prox}_{\gamma f_i}(x_k)\right\|^2}{\left\|\frac{1}{n}\sum_{i=1}^{n}\left(x_k - \text{prox}_{\gamma f_i}(x_k)\right)\right\|^2}.$$

For `FedExProx-StoPS`, we have

$$\alpha_k = \alpha_{k,S} = \frac{\frac{1}{n}\sum_{i=1}^{n}\left(M_{f_i}^{\gamma}(x_k) - \inf M_{f_i}^{\gamma}\right)}{\gamma\left\|\frac{1}{n}\sum_{i=1}^{n}\nabla M_{f_i}^{\gamma}(x_k)\right\|^2}.$$

In order to implement $\alpha_{k,S}$, the server requires each client to send the function value of its Moreau envelope at the current iterate to it, and we need to know each $\inf M_{f_i}^{\gamma}$ which, according to Lemma 5, is the same as $\inf f_i$.

From Figure 6, we can observe that in all cases when $\gamma$ is sufficiently large, `FedExProx-StoPS` is the best among the three algorithms considered, and `FedExProx-GraDS` outperforms `FedExProx`, this provides numerical evidence for the effectiveness of our proposed algorithms. In the cases when $\gamma$ is small, the convergence of `FedExProx-GraDS` seems to be better than the other two algorithms. We also plot the difference of extrapolation parameter used by the algorithms in each iteration. From Figure 7, observe that when $\gamma$ is small, $\alpha_{k,G}$ is often much larger than $\alpha_{k,S}$, resulting in better convergence of `FedExProx-GraDS` as observed in the first two plots of Figure 6. When $\gamma$

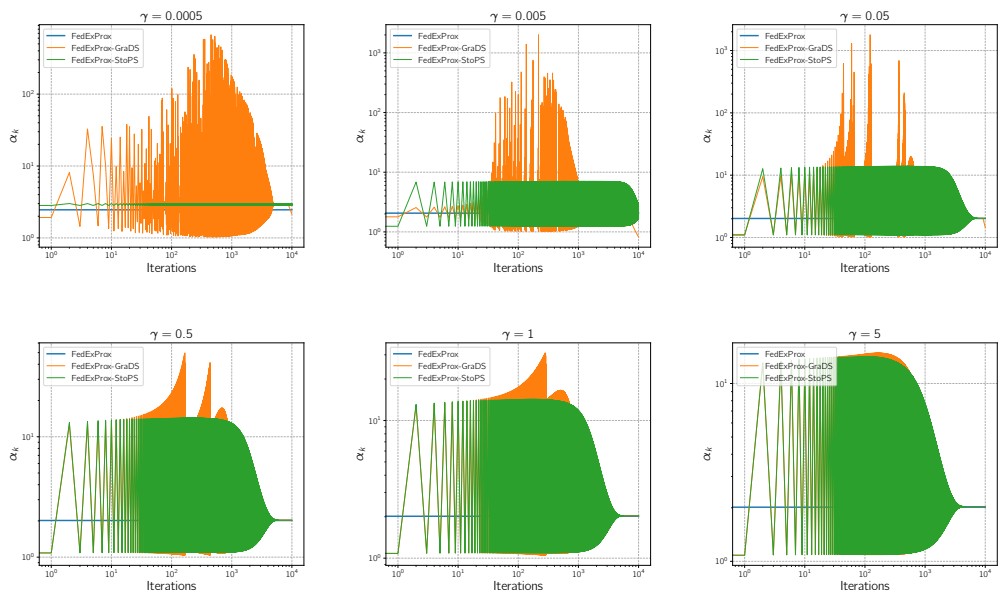

Figure 7: Comparison of the extrapolation parameter $\alpha_k$ used by `FedExProx`, `FedExProx-GraDS` and `FedExProx-StoPS` in each iteration with different step sizes $\gamma$ chosen from $\{0.0005, 0.0005, 0.05, 0.5, 1, 5\}$ in the full participation setting.

becomes larger, $\alpha_{k,G}$ and $\alpha_{k,S}$ become comparable, and their performance is also comparable, with `FedExProx-StoPS` slightly better than `FedExProx-GraDS`.

We also conduct the experiment where we take client partial participation into account. We can observe from Figure 8 that in all cases, the two adaptive variants `FedExProx-GraDS-PP` and `FedExProx-StoPS-PP` outperform `FedExProx` in iteration complexity, and between the two adaptive variants, `FedExProx-GraDS` is the better one almost all the time. However, `FedExProx-GraDS` seems to be more stable than `FedExProx-StoPS`, especially when $\gamma$ is small.

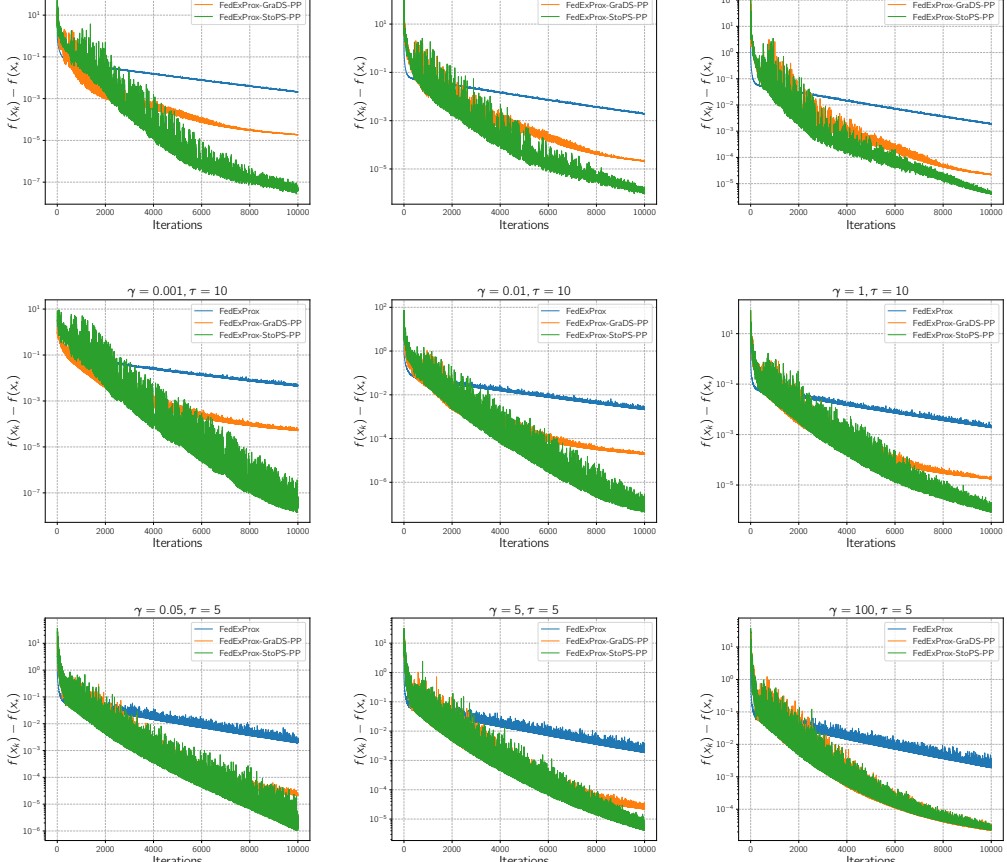

Figure 8: Comparison of `FedExProx`, `FedExProx-GraDS` and `FedExProx-StoPS` in terms of iteration complexity with different step sizes $\gamma$ in the client partial participation (PP) setting. The client minibatch size is chosen from $\{5, 10, 20\}$, for each minibatch size, a step size $\gamma \in \{0.001, 0.005, 0.1, 0.5, 1, 5, 10, 50, 100, 500\}$ is randomly selected.

