# OpenReview forum: "The Power of Extrapolation in Federated Learning"
_NeurIPS.cc/2024/Conference — NeurIPS 2024 poster_

### Official Review · Reviewer_HgaS · 2024-07-05

**Soundness:** 3
**Presentation:** 3
**Contribution:** 3
**Rating:** 6
**Confidence:** 4

**Summary:**

The paper presents a new method FedExProx, a federated learning method based on proximal splitting using extrapolation. The method combines the proximal splitting approach from FedProx with the extrapolation from FedExp. FedExProx is shown to improve upon both previous methods in multiple ways, among others faster convergence rates. Then, two adaptive methods are introduced that allow to choose the extrapolation parameter without prior knowledge of the smoothness constants of the individual Moreau envelopes and $L_max$ for both the full and partial participation scenario.

**Strengths:**

The paper introduces a new method that improves on both FedProx and FedExP in terms of iteration complexity (assuming that the functions a proxable). Further, it extends FedExP to the partial participation case and the adaptive variants do not require a local step size as opposed to FedExP.

**Weaknesses:**

**Comparison to FedExP:** FedExProx assumes that the prox problems can be solved exactly but the analysis of FedExp takes into account the number of local iterations needed. Hence I find the comparison between the convergence rates to be unfair. For smooth problems the prox problems can be approximated very cheaply and the analysis of prox problems can typically be extended to allow for some error in the prox solution. For general convex and L-smooth functions, if one chooses $1/L$ as the prox parameter then prox problem becomes strongly convex and smooth with condition number 2 hence computing an epsilon approximation takes only a logarithmic amount of steps. In your example you argue that there exists a closed form solution for the prox. But for that same example there also exists a closed-form solution for the original problem, so I am not convinced by that argument. I think that the paper should emphasize the assumption that the prox of the function is cheaply computable.

**Extrapolation:** Assuming that the prox can be solved exactly, then based on Eq. 8, I don't see the advantage of considering $\gamma$ and $\alpha_k$ separately? Couldn't one just have one step size and compute that using one of the proposed adaptive methods directly? What is gained from decoupling the prox factor from the extrapolation parameter? In FedExp it seems that this is useful because one can decouple the local from the global step size, but since you assume that the prox can computed in closed form, there is not need to adapt the prox parameter right? The only reason I see for this being useful is if one wants to choose the $\gamma$ in order to be able to solve the proxes efficiently.

I am willing to raise my score if the authors clarify these points.

**Questions:**

-

**Limitations:**

-

---

> ### Author Rebuttal · Authors · 2024-08-04
>
> We thank the reviewer for the valuable feedbacks on our paper. Here is a detailed response to the weaknesses and questions the reviewer mentioned.
> - Weakness 1: `FedExProx assumes that the prox problems can be solved exactly but the analysis of FedExp takes into account the number of local iterations needed. Hence I find the comparison between the convergence rates to be unfair. For smooth problems the prox problems can be approximated very cheaply and the analysis of prox problems can typically be extended to allow for some error in the prox solution. For general convex and L-smooth functions, if one chooses as the prox parameter then prox problem becomes strongly convex and smooth with condition number 2 hence computing an epsilon approximation takes only a logarithmic amount of steps. In your example you argue that there exists a closed form solution for the prox. But for that same example there also exists a closed-form solution for the original problem, so I am not convinced by that argument. I think that the paper should emphasize the assumption that the prox of the function is cheaply computable.`
>
> We partly agree with the reviwer on that the paper should emphasize that the proximity operator is computable. This will appear in the next version of the paper. In fact, almost every proximal algorithm needs to assume that the proximity operator is computable in some sense.
>
> In this paper, we assume that the proximity operator is solved exactly for simplicity, as our goal is to demonstrate the effectiveness of extrapolation combined with proximal algorithms in the federated learning setting. Considering inexactness and obtaining a convergence guarantee to a neighborhood is certainly possible but beyond the scope of this paper.
>
> For FedExP, the number of local training rounds plays a role in determining the local stepsize and if we take the largest local stepsize in the interpolation regime, then the convergence is independent of the local training round $\tau$. For FedExProx, however, the amount of local computation needed is hidded in $\gamma$ and often the larger $\gamma$ is, the harder it is to compute the proximity operator. However, this does not prevent us from comparing their iteration complexity, as the idea of local training aims to reduce the total number of training rounds and, consequently, the communication complexity. It is neither feasible to directly compare the total number of computations for the two algorithms nor meaningful from our perspective.
>
> ---
> - Weakness 2: `Assuming that the prox can be solved exactly, then based on Eq. 8, I don't see the advantage of considering $\gamma$ and $\alpha_k$ separately? Couldn't one just have one step size and compute that using one of the proposed adaptive methods directly? What is gained from decoupling the prox factor from the extrapolation parameter? In FedExP it seems that this is useful because one can decouple the local from the global step size, but since you assume that the prox can computed in closed form, there is not need to adapt the prox parameter $\gamma$ right? The only reason I see for this being useful is if one wants to choose the in order to be able to solve the proxes efficiently.`
>
> We did not manually separate $\alpha_k$​ and $\gamma$ to consider them individually; this separation is inherent to the algorithm. This can be seen from the original formulation in Eq (7) of Algorithm 1. The parameter $\gamma$ is the local step size associated with each client and determines the effort needed to solve the proximity operator. Often, the larger $\gamma$ is, the more challenging the local problem becomes. Note that we do not assume the proximity operator can be solved in closed form. The parameter $\alpha_k$​ is used for extrapolation. It just so happens that after the reformulation in Eq (8), their product becomes the step size for a gradient-based algorithm to minimize the average of Moreau envelopes. Note also that different $\gamma$ values correspond to different Moreau envelopes $M^{\gamma}_{f_i}$ as local objectives. As a result, $\gamma$ influences the local problems we try to solve and cannot be considered in combination with $\alpha_k$​ directly.
>
> It is fair to ask which $\gamma$ is the optimal local step size since we can find the optimal constant extrapolation parameter $\alpha_k$​ for each $\gamma$. However, this requires more information about the smoothness of the average of the Moreau envelope $L_{\gamma}$, which is usually unavailable.

---

> > ### Comment · Reviewer_HgaS · 2024-08-09
> >
> > > Weakness 1
> >
> > I don't understand why it is neither feasible nor meaningful to compare the complexity of two algorithms that are designed for precisely the same setting, i.e. smooth, convex function in the interpolation regime. I think it is fair that you want to focus on the communication complexity rather than the local iteration complexity, but in this case this should be clearly discussed in the paper.
> >
> > > Weakness 2
> >
> > What I meant with my comment was precisely that the fact that the differentiation between $\gamma$ and $\alpha_k$ only becomes relevant when taking into account the interaction between the communication complexity and the amount of local steps required. So one the one hand you simply assume that you can solve these local problem or at least you do not discuss how to solve them, but then you say that having the ability to set $\gamma$ and $\alpha_k$ separately is useful to adapt to the local problems.
> >
> > I've raised my score to 6.

---

> ### Author Response · Authors · 2024-08-10
>
> Thank you for your timely response. We appreciate your effort in reviewing our work.
>
> - Weakness 1: We agree with the reviewer and will include a discussion on our focus on communication complexity in the paper.
>
> - Weakness 2: We now better understand the reviewer's concern. Indeed, we did not include such explanations in the paper. In the next version, we will provide a discussion on solving the local problems and the role of the two parameters in this case.
>
> We will add the following discussion to the paper in its next version.
> > Each local proximity operator can be solved using different oracles. In practice, clients may use gradient descent or stochastic gradient descent to solve the local problem to a certain accuracy. The complexity of this subroutine depends on the local stepsize $\gamma$. If $\gamma$ is large, the local problem becomes harder to solve because we aim to minimize the local objective itself. Conversely, if $\gamma$ is small, the problem is easier since we do not stray far from the current iterate. As the choice of subroutine affects local computation complexity, comparing it directly with FedExP becomes complicated. Therefore, we compare the iteration complexity of the two algorithms, assuming efficient local computations are carried out by the clients.
>
> Please let us know if anything is unclear.

---

### Official Review · Reviewer_eQ2L · 2024-07-12

**Soundness:** 3
**Presentation:** 3
**Contribution:** 2
**Rating:** 6
**Confidence:** 5

**Summary:**

This paper proposes and analyzes several server-side extrapolation strategies to enhance the theoretical and empirical convergence properties of FedProx.
The authors present the convergence properties of the proposed methods for smooth convex and strongly convex problems in the interpolation regime.
Theoretical results demonstrate that the proposed methods have better dependence on the smoothness coefficient $L$ than FedExP in the general case.
Specifically, they achieve an iteration complexity of $O(L_{\gamma}(1+\gamma L_{\max})/\epsilon)$ compared to FedExP's $\mathcal{O}(L_\max/\epsilon)$. Experimental results validate the theoretical analysis, demonstrating improved performance over both FedProx and FedExP.

**Strengths:**

- A clear and solid proof is presented in the paper. I have reviewed it and believe the result is correct.

- The authors also talk about the setting where the coefficient $L_{\gamma, \tau}$ is unknow. I think these results are useful in practical applications.

**Weaknesses:**

1. Assumption 2, the interpolation regime, seems too strong. Are there any other published papers that use this assumption? If $\nabla f_i(x_*)=0$ for all $i\in [n]$, it suggests that all clients in the system have identical local datasets, which is impractical.
2. The proposed method is only designed for proximal-based algorithms, which limits its application.
3. The experiments are conducted on a small dataset with a logistic regression problem. I encourage the author to conduct experiments on more complex models.

**Questions:**

1. The main concern is the suitability of Assumption 2. Are there any reasons for this assumption?
2. There are some typo:
- Line 212, Eq.(8), $\frac{1}{n}\sum_{i\in \mathcal{S}_k}$, Should this be $\frac{1}{\vert \mathcal{S}_k\vert}$?
- Does Eq.(23) assume full participation? It does not seem to align with the partial participation described in Eq.(7) and and the setting in Theorem 1.
- Line 868, the second equation, Should it be $\chi^2$?

The author should review the rest of the paper to address these typo.

I will raise my score if the authors address my concerns.

**Limitations:**

NA.

---

> ### Author Rebuttal · Authors · 2024-08-04
>
> We thank the reviewer for the valuable feedbacks on our paper. Here is a detailed response to the weaknesses and questions the reviewer mentioned.
> - Weakness 1 & Question 1: `Assumption 2, the interpolation regime, seems too strong. Are there any other published papers that use this assumption? If for all , it suggests that all clients in the system have identical local datasets, which is impractical.`
>
> Indeed, the interpolation regime is a strong assumption. However, in deep learning scenarios, we are often in the overparameterized regime, which is stronger. We emphasize that the interpolation regime does not imply that all local clients have the same datasets, but rather that they share a common minimizer $x_\star$​. An example is the convex feasibility problem where the convex sets $\mathcal{X}_i$​ intersect. Published papers, such as [1] and [2], use the interpolation regime assumption, with theoretical justifications provided in [3].
>
> We discuss our method's performance in the non-interpolation, non-smooth, and non-convex cases in Appendix F. Specifically, without the interpolation regime assumption, our method converges to a neighborhood of the solution, and the optimal constant extrapolation parameter is reduced by a factor of 2. This paper focuses on the interpolation regime, as we were inspired by the extrapolation technique used in projection methods for convex feasibility problems and the similarity between projections and proximal operations. Prior to our method, it was unclear whether a constant extrapolation parameter would be effective.
>
> [1] A. M. Subramaniam, A. Magesh and V. V. Veeravalli, "Adaptive Step-Size Methods for Compressed SGD With Memory Feedback," _IEEE Transactions on Signal Processing_, 2024.
>
> [2] N. Ion, P. Richtárik and A. Patrascu. "Randomized projection methods for convex feasibility: Conditioning and convergence rates." _SIAM Journal on Optimization_, 2019.
>
> [3] S. Arora, S. Du, W. Hu, Z. Li and R. Wang, "Fine-Grained Analysis of Optimization and Generalization for Overparameterized Two-Layer Neural Networks" _PMLR_, 2019.
>
> ---
>
> - Weakness 2: `The proposed method is only designed for proximal-based algorithms, which limits its application.`
>
> The benefits of extrapolation also apply to gradient-based algorithms, allowing one to choose the appropriate algorithm for the specific setting. Extrapolation with gradient-based algorithms in federated settings has already been considered in [1]. In our experiments, we use the proposed algorithm FedExP as a benchmark. Our method achieves a better worst-case convergence guarantee than FedExP. Additionally, compared to gradient-based algorithms, proximal algorithms often enjoy enhanced stability.
>
> Our intuition behind this paper stems from the extrapolation technique used in the projection method to solve the convex feasibility problem and the similarity between projections and proximal operations. This is why we consider proximal algorithms in the first place.
>
> [1] J. Divyansh, S. Wang and G. Joshi, "FedExp: Speeding Up Federated Averaging via Extrapolation." _ICLR_ 2023.
>
> ---
>
> - Weakness 3: `The experiments are conducted on a small dataset with a logistic regression problem.`
>
> We thank the reviewer for the feedback, more experiments will be included in the next version of the paper.
>
> ---
> - Question 2: `The author should review the rest of the paper to address these typo.`
>
> All typos mentioned will be corrected in the next version of the paper. We will carefully check the rest of the paper.
> Specifically,
> - In line 212, Eq (8), it should be $\frac{1}{| S_k |}$ instead of $\frac{1}{n}$.
> - There is indeed a typo in Eq (23), we do not assume full participation here.
> - In line 868, yes, there is a square missing here.

---

> > ### Comment · Reviewer_eQ2L · 2024-08-13
> >
> > Thank you for addressing the majority of my concerns. Given that Assumption 2 is relevant in deep learning scenarios, I recommend that the authors validate the proposed method within neural network settings in the next version of the paper.
> >
> > Based on the responses provided, I am inclined to raise my score to 6.

---

> > > ### Author Response · Authors · 2024-08-13
> > > **Thank you!**
> > >
> > > Thank you!

---

### Official Review · Reviewer_xABN · 2024-07-26

**Soundness:** 2
**Presentation:** 2
**Contribution:** 2
**Rating:** 5
**Confidence:** 4

**Summary:**

In this paper, the authors present an enhanced version of the FedProx algorithm for federated learning, named FedExProx. Unlike the FedProx, this algorithm incorporates an extrapolation step on the server following the computation of the proximal operator on each client. The authors investigate both constant and adaptive extrapolation step-sizes and provide the corresponding convergence results. Numerical experiments demonstrate that this method surpasses FedProx in terms of convergence rate.

**Strengths:**

This paper introduces the global step on the server side to the existing FedProx algorithm for the first time, resulting in a new algorithm called FedExProx. The analysis also includes an adaptive global step size, inspired by FedExP. The paper provides clear convergence results, and the theoretical proofs for the main theorems are also presented with great clarity.

**Weaknesses:**

This paper has two main weaknesses. \
First, the benefits gained from the extrapolation step are not evident. As shown in Table 2 of this manuscript, the only noticeable difference in the convergence rate compared to the existing FedExP appears to be a difference in the constant factor. The order of parameters, such as $\tau$ (the number of participating clients or the total number of clients in full participation case) and $T$ (the total number of iterations), remains the same as in existing works. Given that this paper primarily focuses on theoretical contributions, this result is not sufficiently strong.\
Second, the assumptions appear to be too strong. Assumptions 2 (interpolation regime) and 3 (convexity) imply that all functions $f_i$ share the same optimal point. This is a very strong assumption, as it essentially indicates that there is no data heterogeneity, unlike existing works such as FedExP and FedProx, which allow for bounded data heterogeneity.

**Questions:**

In Table 2, the authors may need to include the convergence rate of FedProx for comparison, as the proposed algorithm is more similar to an enhanced version of FedProx rather than FedExP. Another suggestion is to compare the convergence rates of these algorithms in the partial participation setting, since as shown in Table 5, the proposed algorithm still performs well in this scenario.\
In line 220, Remark 2, the authors claim that data heterogeneity is successfully managed. However, as mentioned in the weaknesses section, Assumptions 2 and 3 ensure no data heterogeneity. Therefore, this claim may be incorrect.

**Limitations:**

The authors have addressed the main limitation of this paper in Section 5.1.

---

> ### Author Rebuttal · Authors · 2024-08-04
>
> We thank the reviewer for the valuable feedbacks on our paper. Here is a detailed response to the weaknesses and questions the reviewer mentioned.
> - Weakness 1: `First, the benefits gained from the extrapolation step are not evident. As shown in Table 2 of this manuscript, the only noticeable difference in the convergence rate compared to the existing FedExP appears to be a difference in the constant factor. Given that this paper primarily focuses on theoretical contributions, this result is not sufficiently strong.`
>
>
> We respectfully disagree with the reviewer. While Table 2 shows that the worst-case convergence of FedExP and FedExProx differs only by a constant factor, key differences remain. FedExP uses an adaptive step size based on gradient diversity, whereas FedExProx uses a constant step size. The adaptive version of FedExProx offers a better worst-case convergence guarantee, as it depends on $L_{\gamma}(1+\gamma L_{\max})$, which can be up to $n$ times better than $L_{\max}$ according to Lemma 7. Thus, our method has a stronger theoretical guarantee. Additionally, no convergence guarantee was provided for FedExP with a constant extrapolation parameter in the original paper and it was not clear whether or not a constant extrapolation parameter would be effective or not, so a theoretical comparison with FedExProx under these conditions was not possible. In general, due to the differences in the adaptive rules for selecting the extrapolation parameter, it is challenging to compare the convergence directly. However, our experiments in Figure 4 confirm that the iteration complexity of our method, with a properly chosen step size $\gamma$ and a constant extrapolation $\alpha$, outperforms FedExP with an adaptive extrapolation parameter.
>
> ---
> - Weakness 2: `Second, the assumptions appear to be too strong. Assumptions 2 (interpolation regime) and 3 (convexity) imply that all functions share the same optimal point. This is a very strong assumption, as it essentially indicates that there is no data heterogeneity, unlike existing works such as FedExP and FedProx, which allow for bounded data heterogeneity.`
>
> Indeed, the interpolation regime is a strong assumption. However, in deep learning scenarios, we are often in the overparameterized regime, which is even stronger. As noted in Appendix F.3, without assuming the interpolation regime, the method converges to a neighborhood and the optimal extrapolation constant is reduced by a factor of $2$. This paper focuses on the interpolation regime, as we are inspired by the extrapolation used in the projection methods of convex feasibility problems and the similarity between proximal operations and projections. We also provide discussions of our proposed algorithm in the non-smooth setting, non-convex setting and strongly convex setting in Appendix F.
>
> ---
> - Question 1: `In Table 2, the authors may need to include the convergence rate of FedProx for comparison, as the proposed algorithm is more similar to an enhanced version of FedProx rather than FedExP. Another suggestion is to compare the convergence rates of these algorithms in the partial participation setting, since as shown in Table 5, the proposed algorithm still performs well in this scenario.`
>
> Thanks for the suggestion. We will add a comparison of FedProx and FedExProx and those algorithms in the partial participation setting in the next version of the paper.
>
> ---
> - Question 2: `In line 220, Remark 2, the authors claim that data heterogeneity is successfully managed. However, as mentioned in the weaknesses section, Assumptions 2 and 3 ensure no data heterogeneity. Therefore, this claim may be incorrect.`
>
> Thanks for pointing this out, this is indeed a typo, and we will delete the last sentence in Remark 2 in the next version of the paper.

---

> ### Author Response · Authors · 2024-08-13
> **Did our response address your concerns?**
>
> Dear Reviewer xABN,
>
> Thanks again for your review. Did our rebuttal address your concerns? Please note that the other two reviewers increased their scores from 5 to 6 after rebuttal. Our average score is 5.33 now, which may still be considered borderline (the official email from NeurIPS said that scores between 4.5-5.5 are borderline). Your score may therefore have a large influence on the fate of our work.
>
> Please let us know whether the concerns were addressed. If that is the case, we would of course be happy if this could be reflected in the score. If not, please let us know what remains to be explained -- today is the last day we can do so.
>
> Thanks again!
>
> Authors

---

> > ### Comment · Reviewer_xABN · 2024-08-13
> >
> > Thank you for addressing my concerns. Regarding the replies about the first weakness, I do not fully agree with your explanation. The authors claim that this paper uses a constant step size compared to existing works such as FedExp. However, to the best of my knowledge, in cases where there is no gradient noise and no data heterogeneity, FedAvg can also use a constant learning rate. Moreover, in FedExp, the global step size \(\eta\) is required to satisfy \(\eta \le \frac{1}{\tau L}\), which still constitutes a constant step size, doesn't it?
> >
> > Based on the responses provided, I will raise my score to 5.

---

> > > ### Author Response · Authors · 2024-08-13
> > > **Thanks!!**
> > >
> > > We will respond to this question shortly.
> > >
> > > authors

---

> > > ### Author Response · Authors · 2024-08-13
> > >
> > > Thank you for your response.
> > >
> > > Regarding your concern, FedAvg indeed uses a constant learning rate $\eta_l$ for solving each local optimization problem. However, it does not employ extrapolation ($\eta_g = 1$). In its extrapolated version, FedExP, the constraint $\eta_l \leq \frac{1}{6\tau L}$​ applies to the local stepsize $\eta_l$​. The extrapolation parameter $\eta_g$​, on the other hand, is determined adaptively in each round based on gradient diversity. The original paper does not provide a theory for using a constant $\eta_g$​ in FedExP.
> > >
> > > Hope this clarifies your concern. We appreciate your time and efforts on reviewing our paper.

---

> > > > ### Comment · Reviewer_xABN · 2024-08-13
> > > >
> > > > There is a version of FedAvg that employs both a local learning rate and a global learning rate. For more details, you can refer to the paper "Achieving Linear Speedup with Partial Worker Participation in Non-IID Federated Learning."

---

> > > > > ### Author Response · Authors · 2024-08-13
> > > > >
> > > > > Thanks for the information!

---

### Author Rebuttal · Authors · 2024-08-04

We thank all the reviewers for their time and effort.

The reviewers highlighted several key strengths of our paper. Notably, we introduced an extrapolation parameter to the FedProx algorithm for the first time, developed adaptive versions that eliminate dependence on the unknown smoothness parameter, extended the algorithm to handle partial participation, and provided clear and robust proofs for our claims.

The reviewers also had questions and concerns, which we addressed in the individual responses.

---

### Decision · Program_Chairs · 2024-09-25

**Decision:**

Accept (poster)

**Comment:**

This paper introduces Extrapolated FedProx (FedExProx) to enhance the convergence of FedProx in federated learning. The authors propose three extrapolation strategies: a constant strategy and two adaptive strategies based on gradient diversity (FedExProx-GraDS) and the stochastic Polyak stepsize (FedExProx-StoPS). Theoretical insights are supported by robust numerical experiments, making this a valuable contribution to the community. All reviewers have reached a consensus that this paper should be accepted.